# Structural library and visualization of endogenously oxidized phosphatidylcholines using mass spectrometry-based techniques

Yuta Matsuoka[1], Masatomo Takahashi [2], Yuki Sugiura[3], Yoshihiro Izumi [2], Kazuhiro Nishiyama [4], Motohiro Nishida [4,5], Makoto Suematsu [3], Takeshi Bamba [2] & Ken-ichi Yamada [1✉]

Although oxidized phosphatidylcholines (oxPCs) play critical roles in numerous pathological events, the type and production sites of endogenous oxPCs remain unknown because of the lack of structural information and dedicated analytical methods. Herein, a library of 465 oxPCs is constructed using high-resolution mass spectrometry-based non-targeted analytical methods and employed to detect 70 oxPCs in mice with acetaminophen-induced acute liver failure. We show that doubly oxygenated polyunsaturated fatty acid (PUFA)-PCs (PC PUFA;O2), containing epoxy and hydroxide groups, are generated in the early phase of liver injury. Hybridization with in-vivo [18]O labeling and matrix-assisted laser desorption/ionization-tandem MS imaging reveals that PC PUFA;O2 are accumulated in cytochrome P450 2E1-expressing and glutathione-depleted hepatocytes, which are the major sites of liver injury. The developed library and visualization methodology should facilitate the characterization of specific lipid peroxidation events and enhance our understanding of their physiological and pathological significance in lipid peroxidation-related diseases.

[1] Physical Chemistry for Life Science Laboratory, Faculty of Pharmaceutical Sciences, Kyushu University, 3-1-1 Maidashi Higashi-ku, Fukuoka 812-8582, Japan. [2] Metabolomics Laboratory, Research Center for Transomics Medicine, Medical Institute of Bioregulation, Kyushu University, 3-1-1 Maidashi Higashi-ku, Fukuoka 812-8582, Japan. [3] Department of Biochemistry, Keio University School of Medicine, 35 Shinanomachi, Shinjuku-ku, Tokyo 160-8582, Japan. [4] Department of Physiology, Faculty of Pharmaceutical Sciences, Kyushu University, 3-1-1 Maidashi Higashi-ku, Fukuoka 812-8582, Japan. [5] Division of Cardiocirculatory Signaling, National Institute for Physiological Sciences and Exploratory Research Center on Life and Living Systems, National Institutes of Natural Sciences, 5-1 Higashiyama, Myodaiji-cho, Okazaki 444-8787, Japan. ✉email: kenyamada@phar.kyushu-u.ac.jp

In biological systems, glycerophospholipids (GPLs) are involved in cell membrane construction, metabolism and signal transition regulation, and other vital processes. However, GPLs contain polyunsaturated fatty acids (PUFAs); therefore, they easily form oxidized GPLs (oxGPLs) under the action of reactive oxygen species generated in intracellular organelles.

In particular, oxGPLs derived from phosphatidylcholine (PC), which is an abundant GPL in several biological tissues, such as the liver[1] and kidney[2], have been shown to be responsible for several pathological events, such as cell death and inflammation. For example, truncated oxPCs, such as 1-palmitoyl-2-(5′-oxo-valeroyl)-sn-glycero-3-phosphocholine, induce cell death driven by lipid peroxidation (LPO) (ferroptosis)[3], whereas oxidized 1-palmitoyl-2-arachidonoyl-sn-glycero-3-phosphocholine regulates the immune responses of macrophages[4] and dendritic cells[5]. These bioactivities suggest that oxPCs are associated with the pathogenesis of several oxidative stress-related diseases, such as liver and cardiovascular diseases[6,7].

The structural diversity of oxGPLs, which can be present in epoxide, hydroxide, hydroperoxide, aldehyde, or carboxylic acid[8] forms, implies that numerous oxPCs could be detected as markers of the above-mentioned LPO-related diseases. However, the number of endogenous oxPCs detected in animal disease models and clinical samples is much lower than expected, as many conventional studies on oxPCs have reported only few measurable molecular species (e.g., lipid hydroxides and peroxides) with clear structures[9,10]. Indeed, the current lipid database, viz. LIPIDMAPS (www.lipidmaps.org/), contains only 53 oxPCs, which again is ascribed to the lack of structural information[11,12]. These oxPCs are produced through complicated and multistep radical reactions[8] and have complex difficult-to-elucidate structures. Conversely, considering the difficulty of understanding the production mechanism of these oxPCs, there is no doubt that a large number of oxPCs would be detectable. In fact, Anthonymuthu et al. described that the number of identified oxidized lipids is negligible compared with the total possible number that can be generated from known lipid structures. Hence, the biggest limitation of oxidative lipidomics is this inability to identify and quantify all oxidized lipids[13].

To address the above-mentioned challenges, we here employ high-resolution mass spectrometry (HRMS)-based nontargeted analysis, which is a powerful method for the discovery of unknown molecular species. The tandem mass spectrometry (MS/MS) parameters of each compound, such as the mass-to-charge ratio ($m/z$) value and fragmentation pattern, provide valuable information on oxPC structure. Furthermore, the recent developments in HRMS and data processing methods enable the comprehensive annotation of large numbers of MS/MS peaks detected for complex samples. We use MS techniques to construct an MS/MS library containing 465 oxPCs. The established library enables the comprehensive analysis of 70 oxPCs formed during acetaminophen (APAP)-induced acute liver failure (ALF) in mice. Furthermore, to clarify the site of oxPC formation, we use in vivo $^{18}$O stable isotope labeling and matrix-assisted laser desorption/ionization-tandem MS-MS imaging (MALDI-MS/MS/MSI) for the visualization of endogenous oxPCs. Our results help elucidate the role of endogenous oxPCs in the pathogenesis of LPO-related diseases.

## Results

### Annotation of oxidized PC16:0/PUFAs via a nontargeted approach.
PCs contain a glycerophosphate backbone linked to two fatty acyl chains, one of which usually belongs to a saturated fatty acid (SFA) or a monounsaturated fatty acid (MUFA) (e.g., palmitic acid (16:0), stearic acid (18:0), and oleic acid (18:1)), and the other

belongs to a PUFA (e.g., linoleic acid (18:2), arachidonic acid (20:4), and docosahexaenoic acid (22:6))[14]. Given the high oxidative resistance of SFAs and MUFAs[8] due to the high bond dissociation energy of the C–H bond at the methylene positions[15], the structural diversity of oxPCs would depend on the oxidative modification of PUFA moieties. Therefore, we first attempted to clarify the oxidative modification patterns of major PUFAs (18:2, 20:4, and 22:6) by nontargeted analysis using three diacyl PC16:0/PUFAs, namely 1-palmitoyl-2-linoleoyl-sn-glycero-3-phosphocholine (PC16:0/18:2), 1-palmitoyl-2-arachidonoyl-sn-glycero-3-phosphocholine (PC16:0/20:4), and 1-palmitoyl-2-docosahexaenoyl-sn-glycero-3-phosphocholine (PC16:0/22:6). Next, based on the analyzed modification patterns, the substructures of other oxPCs were assessed using an in silico method (Supplementary Fig. 1).

Considering the PUFA oxidation mechanism, we used two LPO inducers, viz. 2,2′-azobis(2-methylpropionamidine) dihydrochloride (AAPH) and hemin, which lead to two major LPO processes, viz. hydrogen atom abstraction from PUFAs[8] and lipid hydroperoxide decomposition[16], respectively. PC16:0/PUFAs were oxidized by AAPH or AAPH + hemin. After the extraction of lipids from these samples, both extracts were mixed and analyzed by high-performance liquid chromatography coupled with HRMS (LC/HRMS) in negative ion mode, which is a reliable diagnostic tool for lipid structural elucidation[17]. Then, the HRMS spectra of nonoxidized and oxidized samples were acquired. To select oxPC-derived peaks from among the HRMS spectra, signals with oxidized/nonoxidized intensity ratios >2.0 were picked using a background subtraction approach. Peak detection for each data set (oxidized/nonoxidized PC16:0/18:2, oxidized/nonoxidized PC16:0/20:4, or oxidized/nonoxidized PC16:0/22:6) was performed using Compound Discoverer 3.1 software with optimized automatic filtering criteria[18] (Supplementary Fig. 2). The picked $m/z$ peaks in HRMS/MS spectra were carefully annotated de novo (i.e., manually) after the interpretation of molecular substructures by checking three typical product ions derived from the PC head group and two fatty acyl groups (Fig. 1a). For instance, Fig. 1b, c, and f show three major cases of annotation procedure used for the observed oxPCs derived from PC16:0/18:2.

When an ammonium formate-containing elution buffer was employed, PCs were detected as [M + HCOO]$^-$ ions in the negative ion mode. Under this condition, specific product ions corresponding to the two fatty acyl groups and the loss of CH$_3$ group with the product being deprotonated via loss of the formate anion were observed in HRMS/MS spectra, as reported previously[17]. For instance, the HRMS/MS spectrum of $m/z$ 680.4138 showed the characteristic product ions derived from the two fatty acyl groups ($m/z$ 255.2324 [16:0]$^-$ and $m/z$ 157.0864 [C$_8$H$_{13}$O$_3$]$^-$) and methyl group loss (620.3927; [M−CH$_3$]$^-$) (Fig. 1b). Collectively, this information allowed the precursor ion at $m/z$ 680.4138 to be annotated as PC16:0_8:1;O.

The next case exemplifies the generation of two or more product ions from oxPUFA moieties during HRMS/MS (Fig. 1c). The HRMS/MS spectrum of $m/z$ 834.5496 showed two product ions from oxidized fatty acyls ($m/z$ 311.2222 and 293.2124, eluting at 9.0–10.5 and 11.0–12.0 min, respectively) (Fig. 1c, d and Supplementary Fig. 3a, b). According to a previous study, these product ions were ascribed to the functional isomers of 18:2;O2, e.g., hydroperoxide and epoxy-hydroxide[19]. To discriminate between these isomers, we analyzed the corresponding protonated ion ($m/z$ 790.5598, [M + H]$^+$) in the positive ion mode (Fig. 1e). The HRMS/MS spectrum at 9.0–10.5 min showed a fragment ion due to H$_2$O loss ($m/z$ 772.5293), which supported the presence of epoxide and/or hydroxide derivatives (Supplementary Fig. 3c). In contrast, the spectrum at 11.0–12.0 min showed a fragment ion due to the loss of H$_2$O$_2$ ($m/z$ 756.5544), which suggested the presence of hydroperoxide derivatives (Supplementary Fig. 3d). Based on these

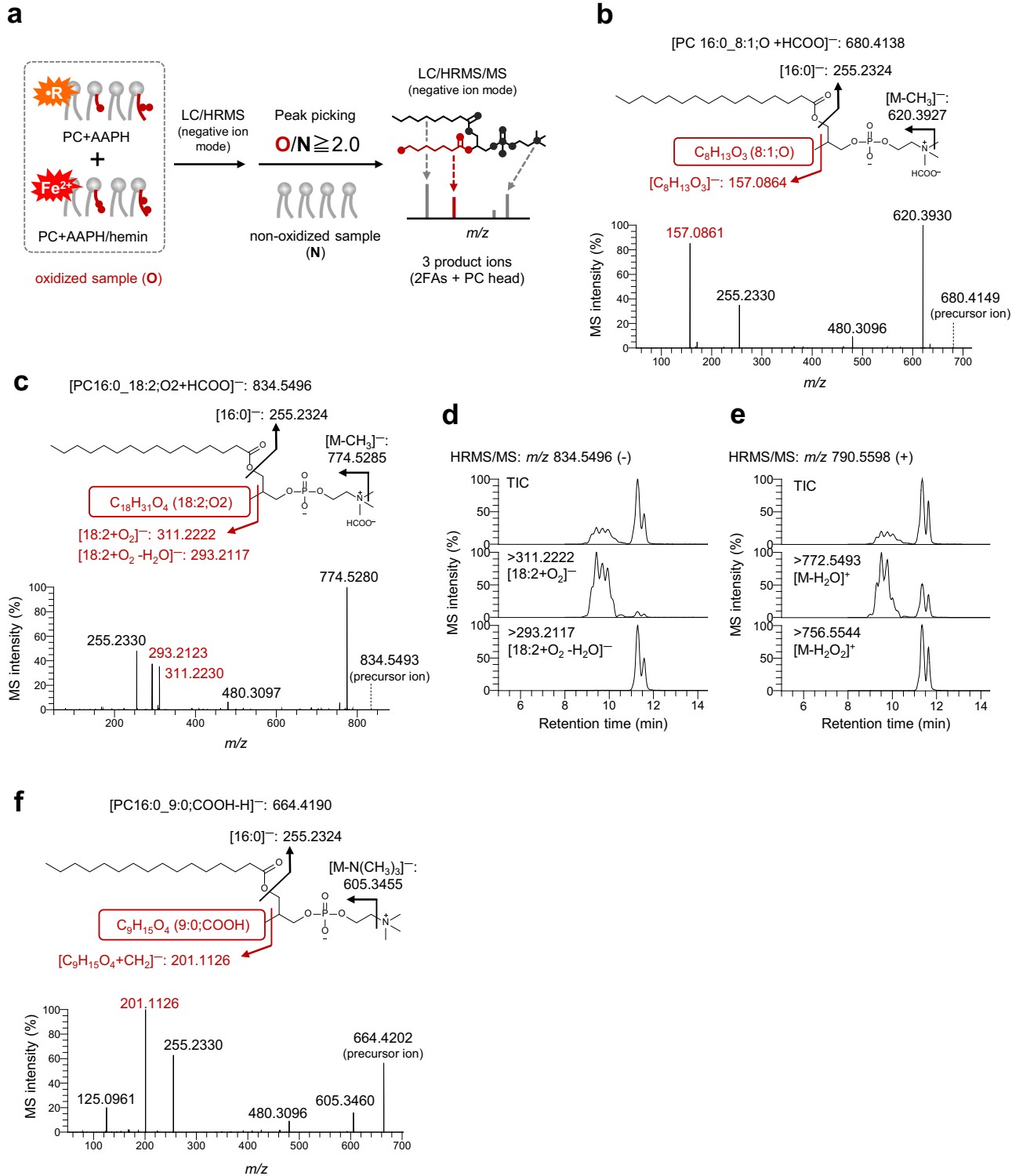

**Fig. 1 Nontargeted analysis and annotation of oxidized PC16:0/PUFAs. a** Outline of the LC/HRMS/MS-based nontargeted analysis of oxidized PCs generated by AAPH or AAPH + hemin stimulation. LC/HRMS/MS data were obtained by using data-dependent acquisition (DDA) in the negative ion mode. **b** HRMS/MS product ion spectrum of PC16:0_8:1;O ($m/z$ 680.4138, retention time ($t_R$) = 6.6 min). The structure and proposed fragmentation of the ion with $m/z$ 680.4138 are also shown. **c** Averaged HRMS/MS product ion spectrum of PC16:0_18:2;O2 ($m/z$ 834.5496, $t_R$ = 9.0–12.0 min). The structure and proposed fragmentation of the ion with $m/z$ 834.5496 are also shown. **d** Extracted ion chromatograms (EICs) recorded in negative ion mode for the precursor ion at $m/z$ 834.5496. Product ions at $m/z$ 311.2222 and 293.2117 correspond to $18:2 + O_2$ and $18:2 + O_2 - H_2O$, respectively. The total ion chromatogram (TIC) is shown. **e** EICs recorded in positive ion mode for the precursor ion at $m/z$ 790.5598. Product ions at $m/z$ 772.5493 and 756.5544 correspond to the loss of $H_2O$ and $H_2O_2$, respectively. **f** HRMS/MS product ion spectrum of PC16:0_9:0;COOH ($m/z$ 664.4190, $t_R$ = 3.0 min). The structure and proposed fragmentation of the ion with $m/z$ 664.4190 are also shown.

results, we ascribed the transitions of $834.5496 \rightarrow 311.2222$ and $834.5496 \rightarrow 293.2117$ to the fragmentation of the epoxide and/or hydroxide (PC16:0_18:2;O2) and hydroperoxide (PC16:0_18:2;OOH) forms, respectively. Thus, the functional isomers of oxPCs were individually identified through the interpretation of their HRMS/MS spectra in both negative and positive ion modes.

Some oxPCs containing ionized carboxylic acid moieties were detected as deprotonated anions. These oxPCs featured characteristic product ions derived from the loss of the $-N(CH_3)_3$ group $[M-H-N(CH_3)_3]^-$ and methylated oxidized fatty acyls $[oxFA + CH_2]^-$[20]. For example, $m/z$ 664.4190 (Fig. 1f) showed a loss of the $-N(CH_3)_3$ group ($m/z$ 605.3455) and a product ion derived from methylated oxidized fatty acyls ($m/z$ 201.1126; $[C_9H_{15}O_4 + CH_2]^-$). Hence, this precursor ion was assumed to contain a carboxyl group. Furthermore, the observation of a saturated fatty acyl-derived product ion at $m/z$ 255.2324 ($[16:0]^-$) allowed the ion at $m/z$ 664.4190 to be annotated as PC16:0_9:0;COOH.

In addition, oxPCs derived from PC16:0/20:4 and PC16:0/22:6, which should have more complicated structures, could be identified following the above-mentioned procedures (Supplementary Fig. 4). Finally, we identified 155 oxPC species derived from the three PC16:0/PUFAs (Supplementary Data 1). Surprisingly, among them, 103 oxPCs were novel, which have not been reported by previous studies, as listed in Supplementary Data 2.

Next, the structural information on oxidized PC16:0/PUFAs was used to elucidate the structures of oxidized PC18:0/PUFAs and oxidized PC18:1/PUFAs. Finally, an MS/MS library of 465 oxPC species was constructed (Supplementary Data 3).

**Effect of LPO inducers on oxPC production**. To examine the generality and comprehensiveness of our library, we employed several LPO inducers that have been used in previous studies on oxidized lipids[21–23], namely AAPH, AAPH + hemin, $CuSO_4$ + ascorbic acid (AsA), and autoxidation (Fig. 2). Although the product profiles of oxidized PC16:0_18:2 were inducer-dependent, all peaks were listed in the constructed library and could be divided into two groups according to the presence or absence of metal ions (Fig. 2a). A particularly large increase in the levels of PC hydroperoxides, such as PC16:0_18:2;OOH, was observed after stimulation by AAPH or autoxidation (Fig. 2b), whereas in the presence of metal ions, i.e., after AAPH + hemin or $CuSO_4$ + AsA stimulation, a particularly high increase was observed in the levels of mono-oxidized or di-oxidized PCs (Fig. 2c, d). In addition, we investigated the abundances of nonoxidized and oxidized PC16:0/18:2 through LC/HRMS analysis in the positive ion mode, which is a useful method for highly sensitive and semiquantification of (ox)PCs (e.g., limits of detection (LOD) and limits of quantitation of standard oxPCs measured by LC/HRMS in the positive ion mode are listed in Supplementary Data 4). To examine the quantitative potential of LC/HRMS in the positive ion mode, we analyzed the ionization responses of seven commercially available (ox)PCs with different chemical structures and their retention times (Supplementary Fig. 5a, b). LC/HRMS peak areas of (ox)PCs observed in the positive ion mode were much higher than those observed in the negative ion mode. Furthermore, the ionization efficiencies of each PC were not completely consistent, but were relatively close (Supplementary Fig. 5c, d). The LODs of oxPCs using the optimized LC/HRMS method in the positive ion mode were ~30 fmol (Supplementary Data 4), and the HRMS(/MS) operating parameters were set to achieve more than 15 data points across each (ox)PC peak. From these results, we concluded that LC/HRMS in the positive ion mode can be used for the semiquantification of (ox)PCs. Furthermore, for the semiquantification of oxPCs in the positive ion mode, we

used PC15:0/18:1-d7 as an internal standard. Thus, we observed that the proportions of oxPCs, including full-length oxPCs and truncated oxPCs, in the total amount of PCs also differed depending on the LPO inducers (Supplementary Fig. 6).

These results showed that our library was useful for a comprehensive analysis of oxPCs generated under various oxidation conditions and the oxPC formation profile was LPO inducer-dependent.

**Analysis of oxPC production in mice with APAP-induced ALF**. To validate our methodology, we applied it to an exhaustive analysis of endogenous oxPC generation in the liver of mice with APAP-induced ALF (Fig. 3a). Hepatotoxicity induced by APAP, a widely used analgesic and antipyretic agent, is the leading cause of ALF in the USA and other western countries[24]. Under the action of hepatic cytochrome P450 2E1 (CYP2E1), APAP is metabolized to $N$-acetyl-$p$-benzoquinone imine, which is usually detoxified by a glutathione (GSH)-conjugating system. APAP overdose causes GSH exhaustion, resulting in excessive oxidative stress and lethal hepatocellular injury[25]. Although recent studies have shown that LPO and its products are involved in APAP-induced hepatocellular death[26,27], the identities of oxPCs generated in the liver of mice with APAP-induced ALF remain unclear.

Plasma alanine aminotransferase (ALT) levels were remarkably elevated 4 h after APAP administration (300 mg/kg, intraperitoneal (i.p.) injection). They were attenuated by treatment with $N$-acetylcysteine (NAC; 500 mg/kg, i.p.) or mitochondria-targeted antioxidant, Mito-TEMPO (MT; 20 mg/kg, i.p.), which has been reported to suppress APAP-induced hepatotoxicity by inhibiting mitochondrial oxidative stress[28], 1 h after APAP administration (Fig. 3b). In contrast, the total GSH level in the murine liver drastically decreased 1 h after APAP treatment and recovered gradually over several hours (4–24 h). This GSH depletion was inhibited by NAC but not by MT (Fig. 3c), which meant that MT suppressed hepatocellular death by inhibiting mitochondrial oxidative stress. Next, we used our library to examine whether oxPCs were produced in APAP-treated mice and observed an increase in the levels of 70 oxPCs upon APAP treatment (Fig. 3d). This increase was completely inhibited 1 h post treatment with NAC or MT.

Interestingly, the profiles of oxPC formation in APAP-treated mice changed with time. For instance, the levels of PC monoxides, as previously detected in the serum of patients with Kawasaki disease[29] and diabetes[30], gradually increased with time up to 8 h after APAP treatment (Fig. 3e). On the contrary, the levels of some oxPCs, including those of newly identified species, remarkably increased 2 h after APAP treatment (Fig. 3f, g). Furthermore, the time profiles of oxPCs containing the same oxPUFA moiety were similar (Supplementary Fig. 7), which might indicate that SFAs in $sn-1$ position do not exert a minimal effect on oxPC formation. Here, we would like to emphasize that the production of these oxPCs increased prior to the drastic increment in the ALT level 4 h after APAP administration (Fig. 3d). These results suggested that the production profiles of the above-mentioned oxPCs were species-dependent, and some oxPCs were produced before the increase in ALT content in the APAP-treated mouse liver, which implied the importance of oxPCs for the subsequent hepatocellular death.

**Detailed structural analysis of PC PUFA;O2 generated in the early phase of APAP-induced ALF**. Next, to clarify the detailed structures of oxPCs, the levels of which increased in the early phase of APAP-induced ALF (2 h after APAP treatment), we used the positive ion mode for semiquantitative LC/HRMS analysis (Fig. 4a). The abundance of oxPCs increased with 2 h APAP

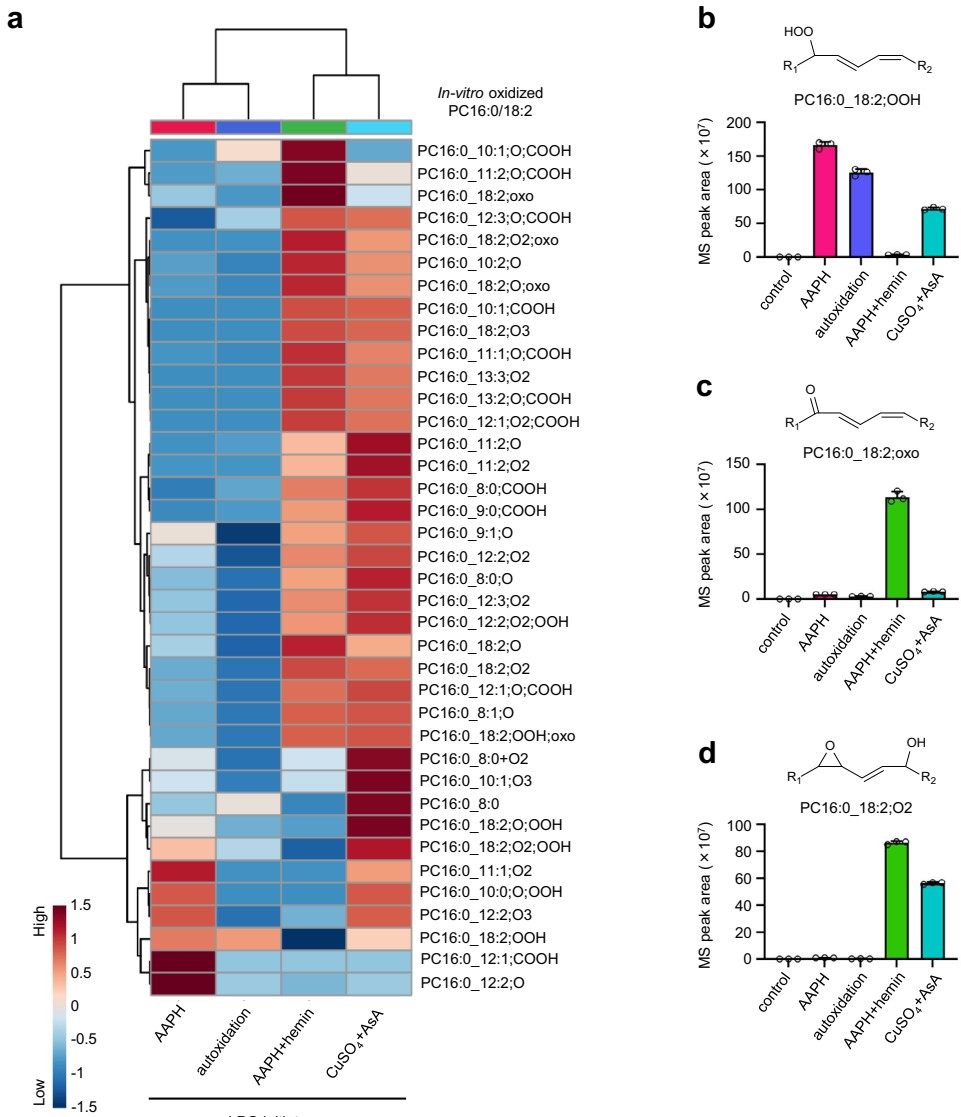

**Fig. 2 Diversity of oxidized PC16:0/18:2 formed in several in vitro peroxidation systems.** **a** Hierarchical cluster analysis of oxidized PC16:0/18:2 was generated in several peroxidation systems in vitro. The color reflects normalized MS peak area of each oxPC using $\log_{10}$ transformation and autoscaling. **b–d** MS peak areas of typical oxidized PC16:0/18:2 observed in each LPO system. **b** PC16:0_18:2;OOH, **c** PC16:0_18:2;oxo, and **d** PC16:0_18:2;O2. Data are presented as the mean + standard deviation of three repeated experiments. MS peak area of each adduct is calculated from the corresponding extracted ion chromatogram peak area. LC/HRMS/MS data were obtained by using parallel reaction monitoring (PRM) in the negative ion mode. Source data are provided as a Source Data file.

treatment (the percentages of oxPCs in vehicle- and APAP-treated mice were calculated to be $0.62 \pm 0.16\%$ and $1.81 \pm 0.30\%$, respectively), whereas that of PCs remained unchanged (Supplementary Fig. 8). Furthermore, the level of full-length oxPCs ($1.56 \pm 0.24\%$) was higher than that of truncated oxPCs ($0.26 \pm 0.07\%$) in APAP-treated mice. In particular, we found remarkable increments in the levels of doubly oxygenated PUFA-PCs (PC PUFA;O2) compared with those of other full-length oxPCs derived from each PUFA-PC (Fig. 4a, b).

The detailed structures of PC PUFA;O2, the levels of which increased 2 h after APAP treatment, were also elucidated by LC/HRMS/MS analysis in negative ion mode. As the HRMS/MS spectra of individual PC16:0_PUFA;O2 mainly featured product ions derived from [PUFA + O₂]⁻, we confirmed the existence of epoxy-hydroxides but not hydroperoxides (Supplementary Fig. 9), as shown in Fig. 1c–e. Figure 4c shows the HRMS/MS spectrum of $m/z$ 834.5496 corresponding to PC16:0_18:2;O2

extracted from the murine liver 2 h after APAP treatment. Ions at $m/z$ 139.1120, 155.1070, and 171.1020 corresponded to the product ions of $[C_9H_{15}O]^-$, $[C_9H_{15}O_2]^-$, and $[18:2-C_9H_{15}O]^-$, respectively, produced by linoleate fragmentation at the C9–C10 position. The ion with $m/z$ 211.1339 was probably produced by bond cleavage at the C12–C13 position. Furthermore, ions at $m/z$ 113.0966 and 185.1177 suggested the presence of oxygen atoms linked to C12 and C10 positions, respectively. Based on these results, two plausible chemical structures of PC16:0_18:2;O2 were suggested, featuring epoxy and hydroxy groups, but not the hydroperoxide motif, at linoleate C9–C10 or C12–C13 positions (Fig. 4c). Similarly, PC16:0_20:4;O2 and PC16:0_22:6;O2 were also concluded to contain epoxide and hydroxide groups mainly located around PUFA terminal moieties (Fig. 4d, e).

These results indicated success in the exhaustive analysis of endogenous oxPCs generated in APAP-treated mice, revealing that PC PUFA;O2 containing epoxide and/or hydroxide motifs

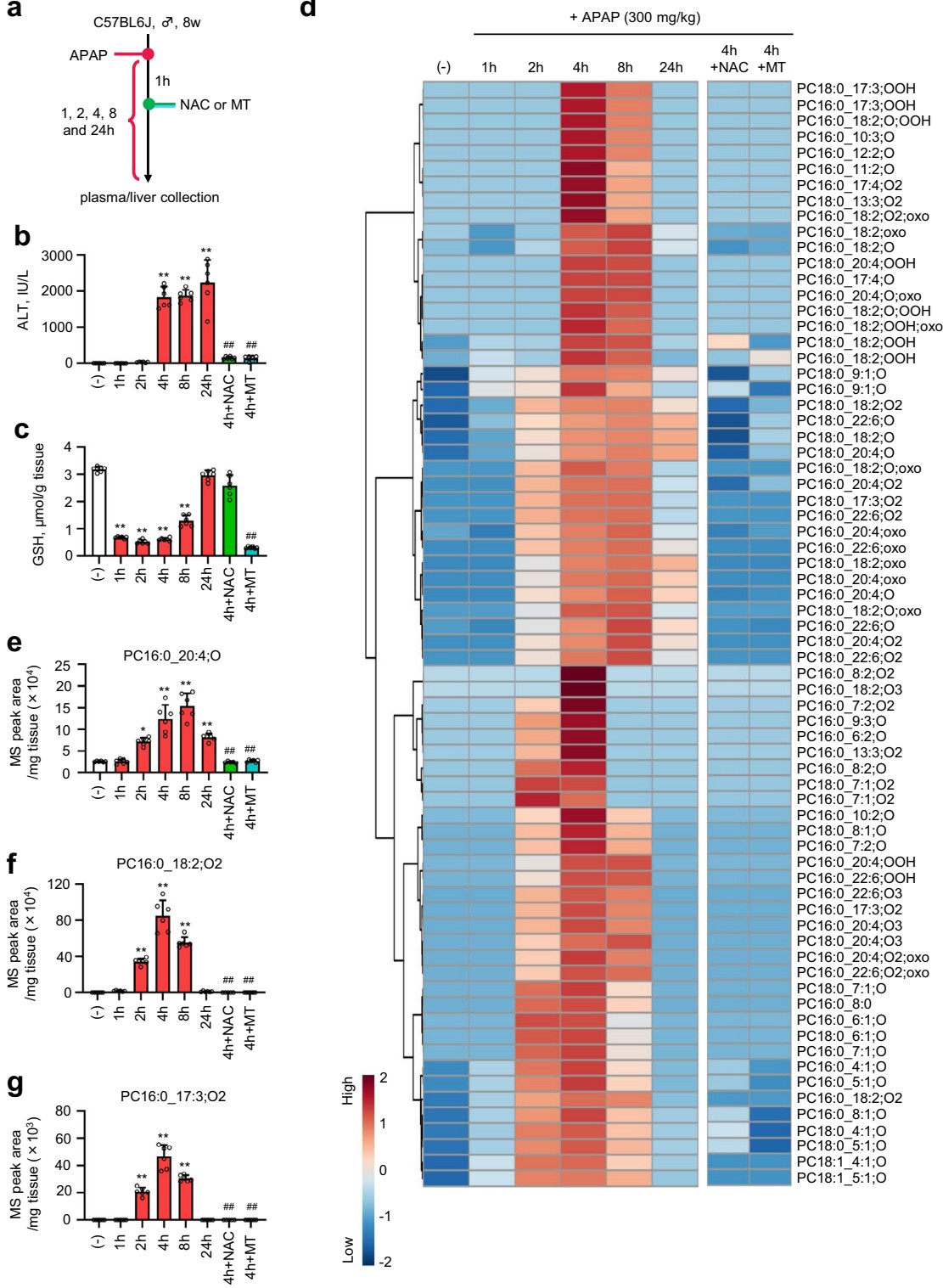

**Fig. 3 Exhaustive analysis of endogenous oxPCs in APAP-treated mice. a** Experimental setup. Plasma and liver samples were collected 1, 2, 4, 8, and 24 h after APAP (300 mg/kg in saline) administration. One hour after APAP injection, the mice intraperitoneally received NAC (500 mg/kg body weight) or MT (20 mg/kg body weight) as antioxidants for liver injury suppression. **b** Plasma ALT level. **c** Total GSH level in the liver. **d** Heat map showing time-dependent profiles of endogenous oxPCs in lipid extracts from the liver of APAP-treated mice (300 mg/kg in saline). The color reflects normalized MS peak area of each oxPC using $log_{10}$ transformation and autoscaling. **e–g** MS peak areas of typical endogenous oxPCs formed after APAP injection. PC16:0_20:4;O (**e**), PC16:0_18:2;O2 (**f**), and PC16:0_17:3;O2 (**g**). Individual MS peak areas were normalized by the wet weight of the liver tissues. LC/HRMS/MS data were obtained by using PRM in the negative ion mode. Data are presented as mean + standard deviation of experiments repeated five or six times. In **b**, **c**, $n = 6$ was used for each group. In **e–g**, $n = 5$ was used for the 4 h + NAC group, $n = 6$ was used for (−), 1, 2, 4, 8, 24, and 4 h + MT group). $P$ value was determined by the one-way ANOVA with the Tukey's multiple comparison test. *$P = 0.0005$, **$P < 0.0001$, compared with the vehicle-treated group. ##$P < 0.0001$, compared with the APAP-treated group (after 4 h). Source data are provided as a Source Data file.

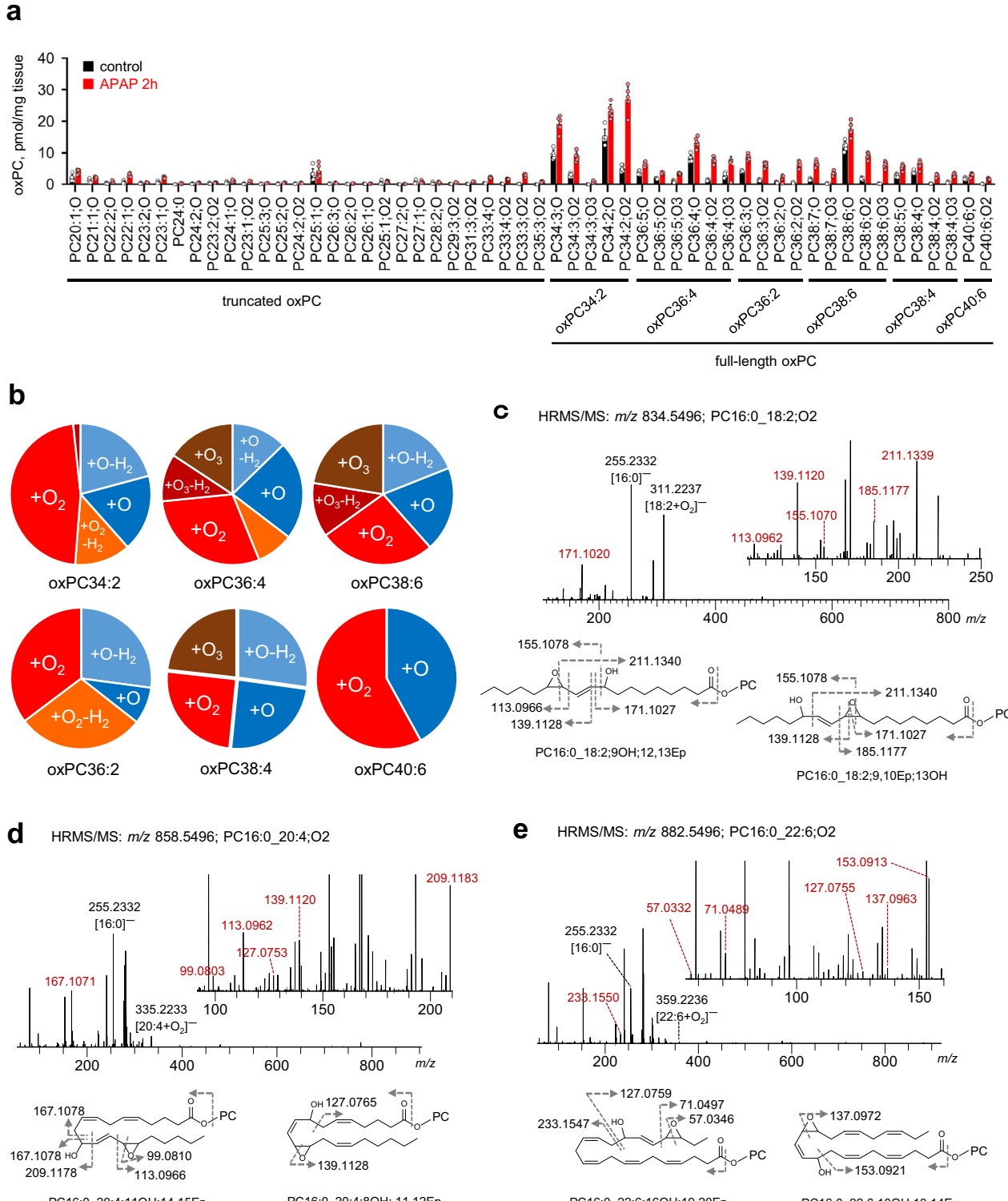

**Fig. 4 Epoxide and/or hydroxide motif-containing PC16:0_PUFA;O2 with remarkably increased levels in the early phase of APAP-induced ALF.**
**a** Semiquantified concentrations of individual oxPCs in vehicle- (black) and 2-h APAP-treated (red) groups. The peak areas from extracted ion chromatogram for each oxPCs were determined by full-scan LC/HRMS in positive ion mode. The semiquantitative values were calculated using the ratio of the MS peak area of individual oxPCs to that of PC15:0/18:1-$d_7$ (internal standard). Data are presented as the mean + standard deviation of experiments repeated six times. **b** Distribution of individual full-length oxPCs (oxPC34:2, oxPC36:4, oxPC38:6, oxPC36:2, oxPC38:4, and oxPC40:6) in the liver of 2-h APAP-treated mice. The levels of oxPCs in the APAP-treated group exceeded those in the vehicle-treated group. Pale blue = PC PUFA;O-H2, blue = PC PUFA;O, orange = PC PUFA;O2-H2, red = PC PUFA;O2, deep red = PC PUFA;O3-H2, and brown = PC PUFA;O3. **c-e** LC/HRMS/MS identification and plausible chemical structures of PC16:0_PUFA;O2 in 2-h APAP-treated samples. HRMS/MS spectra of ions with $m/z$ 834.5496 (**c**), 858.5496 (**d**), and 882.5496 (**e**) corresponding to PC16:0_18:2;O2, PC16:0_20:4;O2, and PC16:0_22:6;O2, respectively. In **c-e**, experiments were performed five times. Source data are provided as a Source Data file.

were predominantly generated in the early phase of APAP-induced liver injury.

**Analysis of oxPC formation by MALDI-MS/MS/MSI based on $^{18}O$ labeling.** We further aimed to determine the site of oxPC formation in the liver of APAP-treated mice and thus shed light on how oxPCs are actually involved in liver damage. However, the visualization of oxGPLs by MSI, which is a powerful method for visualizing biomolecules (e.g., lipids) in biological tissues, remains challenging[11,12], with only a few studies on imaging of two oxPCs containing COOH terminals in the rat spinal cord tissue reported to date[31]. One of the difficulties in oxGPL imaging is the lack of structural information on target oxGPLs. Herein, LC/HRMS/MS analysis and the developed HRMS/MS library containing 465 oxPCs were used to reveal the predominant production of PC PUFA;O2 in the early phase of APAP-induced ALF (Figs. 3 and 4). Hence, we first determined whether oxPCs generated in vitro could be detected through MALDI-MS/MS analysis. Moreover, we selected a specific ion transition to distinguish the characteristic fragment ion of oxPCs from the background noise and thus obtain sufficient signal sensitivity and selectivity.

To determine the pair of precursor and product ions employed in MALDI-MS/MS analysis, we used PC34:2;O2 oxidized by AAPH + hemin treatment in vitro. A specific product ion generated due to $H_2O$ loss from PC34:2;O2 ($m/z$ 790.5 → 772.5) was detected, in line with the LC/HRMS/MS results (Supplementary Fig. 10a). Furthermore, the same transition was detected by the MALDI-/MS/MS system even without AAPH + hemin stimulation, whereas no such transition was observed in the absence of stimulation in the LC/HRMS/MS analysis (Supplementary Fig. 10b). This phenomenon, ascribed to the artificial oxidation to PUFA-PC during sample pretreatment and/or MALDI detection, posed a serious problem for oxPC visualization by MALDI-MS/MS/MSI. To overcome this issue, we attempted to use $^{18}O_2$-containing air (79.5% $N_2$, 20% $^{18}O_2$, and 0.5% $CO_2$) during sample preparation, thereby increasing the mass of oxPC detection ions. When $^{18}O_2$ air was filled in test tubes containing the reaction solution, $^{16}O$ atoms in the oxPUFA moiety were converted to $^{18}O$ (Supplementary Fig. 11). Furthermore, an $m/z$ 794.5 → 774.5 transition, corresponding to the loss of $H_2^{18}O$ from PC34:2;$^{18}O2$, was detected by both MALDI-MS/MS and LC/HRMS/MS analyses, but was not observed in the presence of $^{16}O_2$ (Supplementary Fig. 10c, d). Thus, $^{18}O$ labeling allowed us to exclude unfavorable oxidation artifacts generated during detection and imaging of oxPCs by MALDI-MS/MS/MSI.

**Visualization of PC PUFA;O2 in the liver tissue through $^{18}O$ labeling.** Next, we investigated whether $^{18}O$ labeling can be used to image oxPCs in the livers of APAP-treated mice. The mice were exposed to a flow of $^{18}O_2$ air for 2 h after the treatment with APAP (i.p.) (Fig. 5a). After $^{18}O_2$ inhalation, PC PUFA;$^{18}O2$ were detected in the hepatic lipid-extracted samples of APAP-treated mice by LC/HRMS (Fig. 5b). The observation of characteristic fragment ions due to $H_2^{18}O$ loss from PC PUFA;$^{18}O2$ (PC34:2;$^{18}O2$, PC36:4;$^{18}O2$, and PC38:6;$^{18}O2$) (Fig. 5c) corroborated with the results of the in vitro experiments (Supplementary Fig. 10), and taken together, our findings suggested that in living animals, the inhaled $^{18}O_2$ air was consumed for fatty acid oxidation. Furthermore, under this condition, 72.2% of $^{16}O$ atoms in oxidized fatty acyls were converted to $^{18}O$ (Supplementary Fig. 12). Moreover, we confirmed that $^{18}O_2$ inhalation had little effect on the oxPC product profiles (Supplementary Fig. 13).

Lastly, we visualized oxPCs generated in the livers of APAP-treated mice by MALDI-MS/MS/MSI, which showed a clear and characteristic distribution of endogenous PC PUFA;$^{18}O2$ with limited background noise (Fig. 5d) and that the peak intensity of $^{18}O$-labeled oxPCs produced during the 120-min $^{18}O_2$ air inhalation exceeded that observed with 15-min $^{18}O_2$ air inhalation. APAP-induced liver injury is known to occur at the venous area characterized by a high expression of CYP2E1, a drug-metabolizing enzyme[32,33]. When CYP2E1 expression was evaluated in the serial liver tissue sections of APAP-treated mice analyzed by MALDI-MS/MS/MSI, the localization of PC PUFA;$^{18}O2$, such as PC34:2;$^{18}O2$, PC36:4;$^{18}O2$, and PC38:6;$^{18}O2$, matched well with the areas of CYP2E1 expression and GSH depletion (Fig. 5f).

Taken together, the constructed HRMS/MS library of oxPCs and in vivo $^{18}O$ labeling allowed us to visualize oxPCs selectively with a limited background noise and provided critical information on the oxPC formation site in the tissues of animal disease models.

## Discussion

Herein, we clarified the structures of 155 oxPCs (103 novel oxPCs) derived from three PUFA-PCs using an LC/HRMS/MS-based nontargeted approach. Until now, many researchers have explored novel oxGPLs based on the structures of well-known lipid mediators and the proposed LPO mechanisms[34,35]. However, as the precise LPO mechanism remains unclear, the prediction of all individual LPO products is challenging. In fact, the structures of some newly identified oxPCs, such as PC16:0_8:0 and PC16:0_17:3;O2, could have hardly been predicted from the previously reported mechanisms[16,36–38]. However, the above-mentioned compounds were identified in our animal disease model (Fig. 3d). These results indicate that a nontargeted approach is necessary for the exhaustive identification of diverse oxPCs. Moreover, structural information on the newly identified oxPCs is expected to facilitate the elucidation of novel LPO mechanisms.

The product profiles of oxPCs depended on the presence or absence of metal ions (Fig. 2a). In metal-free LPO systems, such as those employing AAPH stimulation or autoxidation, hydrogen atom abstraction by radical initiators resulted in an excessive accumulation of lipid hydroperoxides, as reported previously[8]. In contrast, metal ions enhanced the decomposition of lipid hydroperoxides to a wide variety of secondary oxidation products, such as lipid ketones, hydroxides, epoxides, and epoxy-hydroxides (Supplementary Fig. 14), as reported previously[16,19]. As mentioned above, the LPO product profiles depended on reaction conditions, and our developed library covered the structures of various oxPCs, regardless of their individual generation mechanisms, allowing the successful detection of 70 endogenous oxPCs formed during APAP-induced ALF (Fig. 4). The exhaustive analysis also revealed the unique kinetic profiles of endogenous oxPCs. In particular, the levels of PC PUFA;O2 predominantly increased in the early phase of liver injury (Fig. 4), and the accumulation regions matched the areas of CYP2E1 expression and GSH depletion (Fig. 5), indicating the importance of oxPCs in ALF progression. The detected species contained epoxide and hydroxide moieties possibly formed via metal ion-induced peroxidation (Figs. 2d and 4c–e). In addition, our detailed structural analysis of PC16:0_18:2;$^{18}O2$ generated in mice that had inhaled $^{18}O_2$, and were treated for 2 h with APAP validated the estimated structures (Supplementary Fig. 15). Previous studies showed that pretreatment with an iron chelator, viz. deferoxamine, inhibits the hepatotoxicity caused by APAP[26]. In addition, APAP overdose causes the translocation of iron from lysosomes to mitochondria, inducing mitochondrial oxidative stress

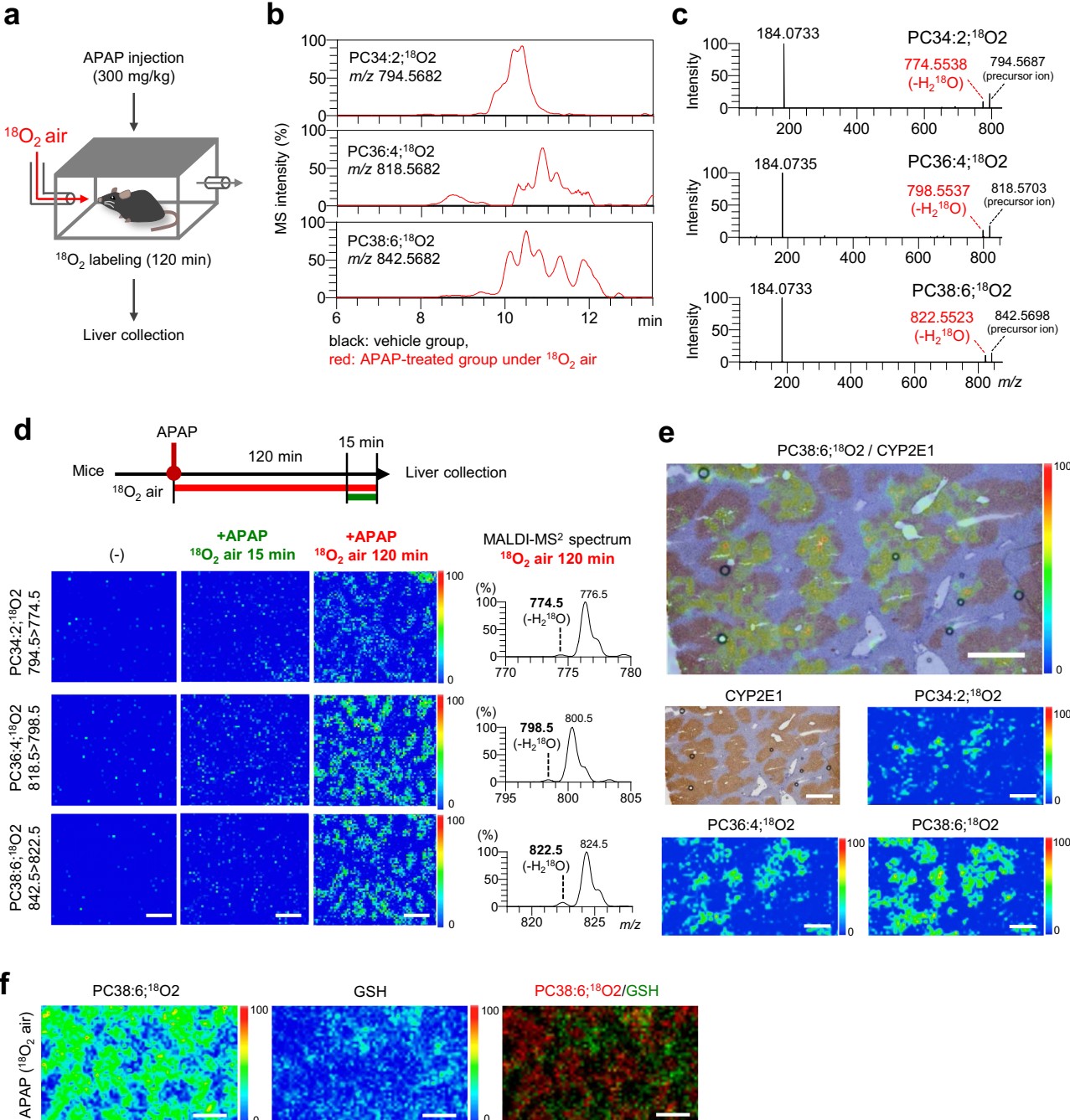

**Fig. 5 Visualization of endogenous oxPCs by a combination of MALDI-MS/MS/MSI and $^{18}O_2$ labeling. a** Experimental design for the $^{18}O_2$ labeling of endogenous oxPCs in APAP-treated mice. **b** Extracted ion chromatograms for ions with $m/z$ 794.5682, 818.5682, and 842.5682 corresponding to PC34:2;$^{18}O2$, PC36:4;$^{18}O2$, and PC38:6;$^{18}O2$, respectively. Black = vehicle-treated group, red = APAP-treated group (120 min in $^{18}O_2$ air). **c** HRMS/MS spectra of parent ions with $m/z$ 794.5682, 818.5682, and 842.5682 were observed for the APAP-treated group (120 min in $^{18}O_2$ air). **d** MALDI-MS/MS/MSI results for PC34:2;$^{18}O2$ (794.5 → 774.5), PC36:4;$^{18}O2$ (818.5 → 798.5), and PC38:6;$^{18}O2$ (842.5 → 822.5) in the mouse liver sections. Left panel = vehicle-treated group, middle panel = APAP-treated group (mice were incubated in normal air for 105 min after APAP treatment and then incubated in $^{18}O_2$ air for 15 min), and right panel = APAP-treated group (120 min in $^{18}O_2$ air). Scale bar = 1 mm. MALDI-MS/MS spectra of $^{18}O$-labeled oxPCs extracted from the liver of APAP-treated mice (120 min in $^{18}O_2$ air). **e** Distribution of $^{18}O$-labeled oxPCs and CYP2E1 expression. Hepatic CYP2E1 expression was analyzed by immunohistochemical staining. Scale bar = 500 μm. **f** MALDI-MS/MS/MSI results for PC38:6;$^{18}O2$ (842.5 → 822.5) and GSH (305.3 → 206.2). Scale bar = 1 mm. In the pseudocolor scale in **d**–**f**, the maximum intensity of the detected signal is set to "100" and the intensity 0 is set to "0".

and hepatic cell death[39]. Taken together, our results suggest that PC PUFA;O2 generated via metal-induced LPO in mitochondria are involved in APAP-induced hepatocellular death. In addition, acute inflammation after hepatocellular necrosis is a critical event in the pathogenesis of APAP-induced ALF[40]. oxGPLs can affect the inflammatory responses via recognition by macrophages[4]. When we investigated the proinflammatory effects of oxPCs on RAW 264.7 cells, the messenger RNA (mRNA) expression of

proinflammatory cytokines increased upon the addition of oxPCs obtained by metal ion-induced (AAPH + hemin-induced) LPO (Supplementary Fig. 16a–c). Furthermore, the lipid extracts from the livers of APAP-treated mice, but not normal mice, showed an increase in the expression of Il–1b mRNA (Supplementary Fig. 16d). These results indicate that the detected oxPCs might belong to the danger-associated molecular patterns released from damaged hepatocytes in APAP-induced ALF. However, further research is needed to clarify the detailed bioactivities of these species towards cell death and inflammation. In addition, the structural determination of oxPCs in vivo is an important issue, and we should address this challenge with novel technologies, such as ion mobility separation[41].

MALDI-MS/MS/MSI coupled with $^{18}$O labeling enabled the specific and sensitive visualization of endogenous oxPCs (Fig. 5), revealing that the artificial oxidation of intact PCs during MALDI/MS analysis resulted in background noise amplification (Supplementary Fig. 10). This finding is consistent with the fact that laser irradiation during MALDI/MS analysis enhances the formation of reactive oxygen species, such as hydroxyl radicals[42]. Although the possibility of artificial oxidation caused by $^{18}O_2$ cannot be ruled out completely, the specific $^{18}$O-labeling of endogenous oxPCs certainly helped eliminate this unfavorable event. The developed visualization method revealed the specific accumulation of PC PUFA;O2 in hepatocytes expressing CYP2E1 (Fig. 5e). Furthermore, we visualized other $^{18}$O-labeled oxPCs generated in APAP-treated mice, such as PC36:4;$^{18}$O and PC38:6;$^{18}$O3, by using the techniques detailed herein (Supplementary Fig. 17). The developed technique is also expected to allow the detection of not only oxidized lipids but also other oxidized species that are unstable or lowly abundant in biological systems, thereby providing novel insight into the oxidation behavior of biomolecules in vivo.

Despite the wide variety of biological GPLs, we identified and detected oxGPLs derived only from diacyl PCs. Thus, the range of target oxGPLs should be further extended to ether oxPCs and oxGPLs belonging to other lipid classes, e.g., phosphatidylethanolamines, phosphatidylserines, and phosphatidylinositols. Moreover, the visualization of oxPCs with a very low concentration in vivo remains challenging. In order to overcome this challenge, future studies on improving the sensitivity and specificity of visualization methods are warranted.

In conclusion, an HRMS/MS-based library of 465 oxPCs was developed and used for the exhaustive analysis of endogenous oxPCs generated during APAP-induced ALF. A method of visualizing endogenous oxPCs by MALDI-MS/MS/MSI and $^{18}$O labeling was established, revealing the specific accumulation of PC PUFA;O2 in CYP2E1-expressing and GSH-depleted hepatocytes.

Although oxPCs are recognized as critical molecules regulating cell death and immune response, the number of identified bioactive oxPCs remains small because of the lack of related structural information and analytical methods. Therefore, the developed library and visualization method enable the discovery of new bioactive oxPCs and shed light on their physiological and/ or pathological significance in LPO-related diseases.

## Methods

**Reagents**. PC16:0/18:2, PC16:0/20:4, and PC16:0/22:6 were purchased from Avanti Polar Lipids (Albaster, AL). AsA, AAPH, CuSO₄, and NAC were sourced from Wako Pure Chemical Industries, Ltd (Osaka, Japan); hemin was procured from Tokyo Chemical Industry Co., Ltd (Tokyo, Japan); APAP was purchased from Sigma-Aldrich (St. Louis, MO); and MT was obtained from Cayman Chemical (Ann Arbor, MI). Acetonitrile (LCMS grade, ≥99.9%), isopropanol (LCMS grade, ≥99.9%), methanol (LCMS grade, ≥99.9%), water (LCMS grade, ≥99.9%), and ammonium formate (Wako 1st grade) were purchased from Wako Pure Chemical Industries, Ltd.

**Preparation of oxidized PC16:0/PUFAs using AAPH or AAPH + hemin**. Oxidized PC16:0/PUFAs were generated from PC16:0/PUFAs by peroxidation in the presence of AAPH or AAPH + hemin, as described previously[21]. A mixture containing 500 µM PC16:0/PUFAs (PC16:0/18:2, PC16:0/20:4, and PC16:0/22:6), 0.5% ethanol, and either 50 mM AAPH or 50 mM AAPH + 10 µM hemin was incubated in phosphate-buffered saline (PBS; pH 7.4) at 37 °C. After 4 h, the mixture was extracted using the modified Bligh and Dyer method[43]. Briefly, a mixed solution of 1 mL methanol and 1 mL chloroform, containing 100 µM dibutylhydroxytoluene and ethylenediaminetetraacetic acid as antioxidants, and 100 nM PC15:0/18:1-$d_7$ as an internal standard was added to the reaction solution. After a 1 min vortex, the organic layer was collected in a glass tube. The extracted solution was dried under a stream of nitrogen gas, and the residue was dissolved in methanol (200 µL) and stored at −80 °C before injection into the high-performance liquid chromatography column.

**Peroxidation of PC16:0/18:2 by autoxidation or CuSO₄ + AsA treatment**. PC16:0/18:2 was oxidized by autoxidation or CuSO₄ + AsA treatment as described previously[22,23]. For autoxidation, a solution of 500 nmol of PC16:0/18:2 in chloroform (1 mL) was immediately dried using nitrogen gas in a glass tube. The dried samples were incubated at 37 °C for 24 h and then resuspended in methanol (200 µL). For CuSO₄ + AsA-induced peroxidation, a solution containing 500 µM PC16:0/18:2 and 500 µM CuSO₄ + 1 mM AsA was incubated in PBS (pH 7.4) at 37 °C for 72 h. After the reaction, individual reaction mixtures were subjected to extraction using the modified Bligh and Dyer method[43], and the extracts were dried under a stream of nitrogen gas.

**LC/HRMS/MS analysis of oxPCs**. LC/HRMS/MS analysis was performed using the Nexera LC system (Shimadzu Co., Kyoto, Japan) coupled with a high-performance benchtop quadrupole Orbitrap mass spectrometer (Q Exactive, Thermo Fisher Scientific, Waltham, MA) that was equipped with an electrospray ionization source. HRMS(/MS) in the negative ion mode is a reliable diagnostic method for elucidating PC structures[17]. This technique has been also applied to monitor the levels of individual oxPCs using the product ions specific to each oxPC[33]. In contrast, since the ionization efficiency of lipid molecular species including PC was mainly dependent on their polar head groups and slightly dependent on their FA side chains[44], this method cannot provide detailed information on oxPC structure, including the chemical formulae of the two fatty acyl moieties. Therefore, we used the negative ion mode in the structural analysis (Fig. 1) and monitored the individual changes in oxPCs (Figs. 2 and 3d–g), whereas the positive ion mode was employed for comparative analysis of the abundances of different oxPCs (Fig. 4a and Supplementary Figs. 6, 8, and 18). The LC parameters and equipment applied were as follows: injection volume = 10 µL, autosampler temperature = 4 °C, column = Inertsil ODS-P (2.1 × 150 mm², particle size = 3 µm, GL Sciences, Tokyo, Japan), column temperature = 50 °C, mobile phase = 5 mM ammonium formate in acetonitrile/H₂O (2:1, v/v) (A) and 5 mM ammonium formate in isopropanol/methanol (19:1, v/v) (B), flow rate = 0.4 mL/min, and gradient = 0–100% B, 0–22.5 min; 100% B, 22.5–27.5 min; and 0% B, 27.5–30.0 min. Individual oxPCs were identified by full-scan MS/data-dependent MS/MS or parallel reaction monitoring (PRM). The ionization conditions were as follows: ionization mode = negative or positive, sheath gas flow rate = 40 arbitrary units, auxiliary gas flow rate = 10 arbitrary units, spray voltage = 2000 V, capillary temperature = 265 °C, S-lens level = 50, and heater temperature = 425 °C. The experimental conditions for full-scan MS were as follows: resolving power = 70,000, automatic gain control target = 1 × 10⁶, trap fill time = 100 ms, scan range = m/z 120–1500. The experimental conditions for MS/MS and PRM were as follows: resolving power = 17,500, automatic gain control target = 1 × 10⁶, trap fill time = 80 ms, isolation width = ±0.6 Da, fixed first mass = m/z 80, normalized collision energy = 20 or 35 eV, intensity threshold of precursor ions for MS/MS analysis = 3100, apex trigger = 2–4 s, and dynamic exclusion = 2 s. The intensity threshold of precursor ions for MS/MS analysis and the dynamic exclusion were set to 1 × 10⁴ and 1 s, respectively. The inclusion list contained 465 precursor ions (m/z) of oxPCs for MS/MS analysis. LC/HRMS/MS analysis was controlled using Xcalibur 4.2.47 software (Thermo Fisher Scientific). EICs of individual oxPCs identified in this study are shown in Supplementary Fig. 19.

**Nontargeted analysis of oxidized PC16:0/PUFAs using Compound Discoverer 3.1**. LC/HRMS data for the nontargeted analysis were obtained through LC/HRMS analysis in the negative ion mode. Compound Discoverer 3.1 software (Thermo Fisher Scientific) was used for data processing, including peak alignment (node name = align retention times), peak detection (node name = detect compounds), data grouping (node name = group compounds), and background subtraction (node name = mark background compounds). Peak alignment for the three sets of LC/HRMS data (i.e., oxidized PC16:0/18:2 and nonoxidized PC16:0/18:2, oxidized PC16:0/20:4 and nonoxidized PC16:0/20:4, or oxidized PC16:0/22:6 and nonoxidized PC16:0/22:6) was performed individually, based on a nonparametric peak alignment algorithm using base peak chromatograms. The parameters for the "align retention times" node were set as follows: alignment model = adaptive curve, mass tolerance = 5 p.p.m., and maximum $t_R$ shift = 0.1 min. The extracted ion chromatogram traces for full-scan LC/HRMS data were detected using the parametric settings for mass tolerance, intensity threshold, and the isotopic pattern. The

same chemical features derived from isotopes and adducts were combined with the most abundant peak using a user-specified ion list and RT information, and the peak area was then calculated for each compound. The parameters for the "detect compounds" node were set as follows: mass tolerance = 5 p.p.m., intensity tolerance for the isotope pattern search = 50%, signal/noise ($S/N$) threshold = 3, minimum peak intensity = 10,000, adduct ions = $[M - H]^-$ and $[M + HCOO^-]^-$, minimum element counts = $C_{26}H_{48}NO_6P$, maximum element counts = $C_{60}H_{130}NO_3P$, remove singlets = true, minimum scans per peak = 3, and minimum isotopes = 1. The following parameters for the "group compounds" node were used to construct a data matrix consisting of the RT, exact mass, and peak area across the sample set: mass tolerance = 5 p.p.m. and RT tolerance = 0.1 min. The parameters for the background subtraction approach (i.e., "mark background compounds" node) applied to nonoxidized and oxidized samples were as follows: maximum peak area (sample/blank) >2.

**Semiquantitative analysis of nonoxidized and oxidized PCs.** LC/HRMS in the positive ion mode was employed for semiquantitative analysis of (ox)PCs. The semiquantitative values were calculated using the ratio of the MS peak area of each oxPC to that of PC15:0/18:1-$d_7$, which was used as an internal standard.

**Animal studies.** Seven-week-old C57BL/6J male mice were purchased from CLEA Japan (Tokyo, Japan). All animal experimentation was approved by the Committee on Ethics of Animal Experiments, Graduate School of Pharmaceutical Sciences (Kyushu University), and Keio University School of Medicine and was conducted according to the Guidelines for Animal Experiments of the Graduate School of Pharmaceutical Sciences, Kyushu University, and Keio University School of Medicine.

**Generation of APAP-induced ALF mouse model and detection of endogenously generated oxPCs.** Eight-week-old C57BL/6J male mice were used. All the mice were housed in a light-controlled room (light/dark cycle of 12 h/12 h) at 24 ± 1 °C and 60 ± 10% humidity and had free access to water and CLEA Rodent Diet CE-2 (Clea Japan, Inc., Tokyo, Japan). A single dose of APAP at 300 mg/kg body weight was intraperitoneally administered into overnight-fasted mice. The animals were anesthetized with a solution of hydrochloric acid medetomidine (Kyoritsu Seiyaku Corporation, Tokyo, Japan), midazolam (Sandoz K.K., Tokyo, Japan), and butorphanol (Meiji Seika Pharma Co., Ltd, Tokyo, Japan) 1, 2, 4, 8, or 24 h after APAP administration. To suppress LPO induction by APAP administration, either 300 mg/kg body weight of NAC or 20 mg/kg body weight of MT in saline was intraperitoneally injected 1 h after APAP administration. Mouse liver and plasma samples were collected, and the liver samples were immediately frozen in liquid nitrogen. Hepatic lipids were extracted from the liver samples according to the modified Bligh and Dyer method[43]. Briefly, 1 mL of extraction solution (methanol: chloroform:water = 5:2:2) containing 100 μM dibutylhydroxytoluene, 100 μM ethylenediaminetetraacetic acid, and 100 nM PC15:0/18:1-$d_7$ was added to a frozen tissue sample (wet weight: ~50 mg) and then the sample was homogenized using a Macro Smash homogenizer. Subsequently, the extraction solutions were sonicated on an ice bath for 5 min. After centrifugation (6000 × $g$, 10 min, 4 °C), 700 μL of the supernatant was collected and then 235 μL chloroform and 155 μL water were added to the supernatant. The organic layer was collected in a glass tube and dried under a stream of nitrogen gas; the dried residue was dissolved in methanol (200 μL) and stored at −80 °C before performing the LC/HRMS/MS experiments.

**$^{18}O_2$ labeling of oxidized PC16:0/PUFAs in vivo.** For $^{18}O_2$ labeling of endogenous oxPCs, C57BL/6J mice were exposed to $^{18}O_2$ air (79.5% $N_2$, 20% $^{18}O_2$, and 0.5% $CO_2$) for 2 h after intraperitoneal administration of APAP (300 mg/kg body weight). Subsequently, the mice were anesthetized with isoflurane and their livers were removed. Hepatic lipids were extracted following the above-mentioned procedure.

**Biochemical analysis.** Plasma ALT activity was measured using an assay kit (Wako Pure Chemical Industries, Ltd). Hepatic total GSH level was measured as described previously[45].

**MALDI-MS/MS/MSI of PC PUFA;O2.** Sample preparation for MALDI-MS/MS/MSI was performed as described previously[46]. Briefly, thin liver sections (8 μm) were prepared using a cryomicrotome (CM3050, Leica Microsystems, Tokyo, Japan) and attached onto indium tin oxide-coated glass slides (Bruker Daltonics GmbH, Leipzig, Germany). The slides were coated with a solution (50 mg/mL) of 2,5-dihydroxybenzoic acid in 80% aqueous ethanol by manual spraying with an artistic brush (Procon Boy FWA Platinum, Mr. Hobby, Tokyo, Japan).

MALDI-MS/MS and MALDI-MSI experiments were performed using a MALDI linear ion trap mass spectrometer (MALDI LTQ XL; Thermo Fisher Scientific, Bremen, Germany), equipped with a 60-Hz $N_2$ laser ($\lambda = 337$ nm). The laser energy and the raster step size were set at 32 μJ and 60 μm, respectively. Auto gain control (where the ion trap is filled with the optimum number of ions) was not used in this experiment. Mass spectra were acquired in the positive ion mode. The MS/MS transitions for PC PUFA;$^{18}O2$ imaging were as follows: $m/z$ 794.5 → 774.5 (neutral loss of $H_2^{18}O$) for PC34:2;$^{18}O2$, $m/z$ 818.5 → 798.5 for PC36:4;$^{18}O2$, and $m/z$ 842.5 → 822.5 for PC38:6;$^{18}O2$-PC, with a precursor ion isolation width of $m/z$ 1.0.

After sample analysis, ion images were reconstructed using ImageQuest v. 1.0.1 software (Thermo Fisher Scientific).

**Immunohistochemical analysis of hepatic CYP2E1 expression.** Serial sections of the liver, similar to those used for MALDI-MS/MS/MSI, were fixed in 4 wt% paraformaldehyde for 5 min at 4 °C, washed in PBS, and incubated for 1 h in a blocking solution (2 wt% bovine serum albumin and 0.2 wt% Triton X-100 in PBS). Next, the sections were incubated overnight at 4 °C in a blocking solution containing primary antibodies (anti-CYP2E1 antibody (HPA009128), Atlas Antibodies; 1:200).

After rinsing in PBS, the sections were incubated in a solution containing secondary antibodies (SignalStain Boost IHC Detection reagent (horseradish peroxidase, rabbit), Cell Signaling Technology; 1:500) for 15 min at room temperature. Antibody binding was detected using the ImmPACT DAB Peroxidase Substrate Kit (Vector Laboratories) and Mayers hematoxylin was used for counterstaining.

**Cell culture.** RAW 264.7 (ATCC, Cat# TIB-71, RRID: CVCL_0493) cells were cultured in Dulbecco's modified Eagle's medium supplemented with 10% fetal bovine serum and 1% penicillin and streptomycin.

**Statistical analysis.** Statistical analyses were conducted using GraphPad Prism (version 9.2.0, GraphPad Software, Inc.) and the data were analyzed by the one-way analysis of variance (ANOVA) with the Tukey's multiple comparison test or two-way ANOVA, followed by Sidak's comparison test. The results are expressed as mean ± standard deviation, and a level of $P < 0.05$ was considered to represent statistical significance. Hierarchical cluster analysis was carried out using the Ward method with a web-based statical tool, MetaboAnalyst 4.0[47].

**Reporting summary.** Further information on research design is available in the Nature Research Reporting Summary linked to this article.

## Data availability

All data generated in this study are provided either in the main figures, Supplementary data, or Source data file. LC/HRMS(/MS) data have been deposited in Metabolomics Workbench (https://www.metabolomicsworkbench.org/) with project ID PR001156 and can be accessed at https://doi.org/10.21228/M88T3S. Source data are provided with this paper.

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

## Acknowledgements
This work was partly supported by AMED-CREST grant (JP20gm0910013 to K.-i.Y.), JSPS KAKENHI grants (18K14884 to Y.M.; 17H03977, 18K19405, and 20H00493 to K.-i.Y.; 17H06304 to Y.I. and T.B.; and 17H06299 to T.B.), Japan. This work was also supported by the Platform Project for Supporting Drug Discovery and Life Science Research from the AMED. We appreciate the technical support from the Research Support Center of the Graduate School of Medical Sciences, Kyushu University.

## Author contributions
Y.M. designed and performed the experiments, analyzed the data, and wrote the manuscript. M.T., Y.S., Y.I., K.N., M.N., M.S., and T.B. performed the experiments and analyzed the data. K.-i.Y. designed the experiments, conceived and supervised the study overall, and wrote and edited the manuscript.

## Competing interests
The authors declare no competing interests.
