## [Peer Review File · Nature Communications]

REVIEWER COMMENTS

Reviewer #1 (Remarks to the Author):

Summary: The authors describe the development of a mass spec-based analytical strategy to detect and visualize oxidized phosphatidylcholines (oxPCs) from biological samples. Their developed method combines experimental and in silico data to construct a database of possible oxPCs. The database was validated using an acute liver failure model. The use of mass spec imaging provided the opportunity to localize oxPCs in liver tissue exposed to toxic levels of acetaminophen. This manuscript provides a starting point for identifying novel oxidized lipids that can give insight into disease progression. The concept and general approach is of high interest and suitable for Nature Communication. Yet, there are a number of issues that need to be resolved in order to justify acceptance for publication.

I recommend major revisions. See below for detailed comments.

Major issues:

- Inconsistent use of lipid nomenclature and interchangeable use of PL/GPL and oxPL/oxPC.
- Justification for focusing on PCs in the context of framing the analytical methodology towards lipids in general.
- Experimental methods as described lack detail to reproduce data. This is especially true for aligning positive and negative ion mode data, calculation of relative abundance (semi-quantitation), and mass spec imaging experiments (e.g., matrix application, MALDI acquisition parameters).
- Description of quality control measures to ensure lipid oxidation was a result of planned experiments (e.g., auto-oxidation, AAPH, APAP) and not an artifact of sample preparation (use of chloroform in lipid extraction) or sample analysis.
- The amount of lipid oxidation in the context of non-oxidized lipids was not addressed. This is important in terms of evaluating the robustness and applicability of the developed analytical method. In other words, how much oxidation needs to be present in order to detect oxPCs? What is the extent of lipid oxidation (as measured by oxPC) in the context of non-oxidized lipids? This is relevant in terms of APAP toxicity and any other disease/injury model. In effect, how sensitive is the developed method for detection of oxPCs?

Introduction:

1. The use of PL instead of GPL is an oversight. PL would include sphingomyelin (among others) but GPL would not. GPL commonly contain PUFAs whereas SM does not. A clarification of lipid nomenclature is needed. It is advised to use GPL instead of PL.
2. The use of “unbiased” should not be used for analytical techniques. At every stage of the analytical process bias introduced. For example, the choice of lipid extraction technique will bias what lipids are observed and their abundance. A more appropriate choice would be to use “discovery” or “un/non-targeted”. The authors use non-targeted in other places of the paper and it should remain consistent.
3. Intro provides background on PL and oxPL but all experiments focus on oxPC. There should be a stronger rationale and description for why experimental approach focuses exclusively on oxPC. Why are oxPCs specifically considered and what is their biological relevance? Are PCs more prone to oxidation? Is it concentration dependent?
4. Throughout the paper there is an interchangeable use of oxPL and oxPC. The authors need to distinguish between the two or justify the interchangeable use. It is too simplistic to suggest that experimental data obtained for oxPC can be readily translated to oxPL.

Results:

“Annotation of oxidized 16:0-PUFA PCs by non-targeted approach”

1. Why was PC16:0/18:2 used as the example to highlight the develop method? This structure represented the simplest of the three standards. Why not use 20:4 or 22:6 for the paper figures to show the complexity and robustness of the approach?
2. The authors describe PC as being diacyl which is certainly true yet PCs and GPL in general have a portion of their structures present in alkyl and/or vinyl ether. The presence of ether GPL is not discussed. What is the rationale for not addressing ether PCs?
3. Sentence “given the high oxidative resistance of SFA’s and MUFA’s,...” should have a reference associated with it.
4. The use of negative and positive ion modes should be explained in regards to the benefits of each. A brief description of their complementary features would suffice.
5. What is the abundance of oxPC in context of total lipid extracts? What is the extent of oxidation for standards and biological samples?
6. Why was O/N >2 used for detection of oxPC? What is the criteria for using this value?
7. The authors say that the HRMS/MS spectra were carefully annotated. Was annotation done de novo (i.e., by hand) or via the aid of software?
8. P. 7 describes “loss of methyl group in the choline moiety were observed in HRMS/MS spectra” as a major product ion. This should be clarified to say it is loss of CH₃ with product being deprotonated via loss of formate anion.

9. Along these lines (#7), all precursor ions in MS/MS spectra should be labeled (Figure 1, 4, 5). This will help clarify the loss of CH₃ and its deprotonated form. The precursor ion is the formate anion.
10. Second sentence of p. 8. "The precursor ion at m/z 834.5496..." is confusing. Clarification of this sentence would be helpful.
11. Figure 1C-E, why was EIC taken for 9-12 minutes? Why not take EIC of each peak, 9-10.5 and 11-12? What do the MS/MS spectra look like with each peak? Do the intensities of 293/311 change? I suggest adding figure (maybe SI) showing these differences.
12. Fig 1E, the positive ion mode MS/MS spectrum should be included. The discussed fragment ion of loss of H₂O should be shown.
13. It was unclear why product ions m/z 311 and 293 were classified as epoxide/hydroxide and hydroperoxide, respectively. Was this determined experimentally or from literature?
14. What was the abundance of the oxPCs containing carboxylates? The paper mentions that some oxPC were carboxylates. What was their abundance compared to other oxPC and un-oxidized PCs?
15. It is claimed that "103 new species" were identified from experiments. All species discussed to this point in the paper were identified previously in literature. This statement should be clarified as to what it means to be a "new species".
16. It is my understanding that the identified "155 species" were from sn-2 fatty acyl of 18:2. How does the analysis of 18:2 translate to more complex fatty acyls of 20:4 and 22:6. Again, the use of a more complicated sn-2 fatty acyl (e.g., 20:4 or 22:6) as an example would help highlight the identification process.
17. A major question I have is what is the percentage oxidation from these experiments? What is the amount of oxidized vs non-oxidized? A figure showing the amount of oxidation would be very helpful for evaluating the effectiveness of developed method to detect oxPC.

"Effect of LPO inducers on oxPC production"

1. No comments

"Validation of methodology: oxPC analysis in mice with APAP-induced ALF"

1. Is there literature precedence for statement that "MT suppressed hepatocellular death by inhibiting mitochondrial oxidative stress"? If so, add reference? If not, this is speculative and unless have other data it should be worded as such.
2. Figure 3 E-G, what about the 18:0 or 18:1 sn-1 species? Were monoxides only in 16:0? Was it dependent on sn-2 structure or did sn-1 contribute?

3. What were the abundances of the oxidized products compared to the un-oxidized lipids? What is the percentage of lipid peroxidation based on identified structures?

“Detailed structural analysis of PUFA+O₂-PCs generated in the early phase of APAP-induced ALF”

1. Why was positive ion mode used for determining abundance?
2. How were semi-quantitative experiments carried out? This should be explained in the methods section.
3. Was determination of peak area used for abundance? This is implied in the text. Does peak area correspond to abundance? Why not say use abundance instead of peak area?
4. Going back to an earlier item, the use of positive and negative ion modes should be described. Why use positive ion mode for abundance and negative ion mode for structure characterization? Is there a need do to both for verification and/or confirmation?
5. Did the chromatography reveal isomers for the linoleate species? Could this be used to confirm presence of hydroperoxides?
6. The use of “comprehensive analysis of endogenous oxPCs...” appears to suggest that all oxPCs have been identified which likely is not the case. I suggest revising to use exhaustive or thorough instead of comprehensive.

“Analysis of oxPC formation by MALDI/MS/MSI based on 18O labeling”

1. This section has a different tone/writing style. It is as if it was copy and pasted.
2. The use of “in test tubes” has not been used in manuscript up to this point. I suggest using in vitro.
3. The specific mention of an “ion trap instrument” appears disjointed. Why not say “a mass spectrometer with tandem mass spectrometry capability was used”? The exact mass spec can be described in methods.
4. The use of “ionic transition” should be revised. Do you mean ion transition, precursor to product ion transition?
5. “flown into test tubes” should be revised to more appropriately describe the experiment.
6. What is the conversion and/or uptake of 18O₂ in these in vitro experiments?
7. Although I agree the use of 18O₂ is a nifty experiment, I am still concerned that the lack of specificity the loss of “H₂O even H₂¹⁸O” is not unique enough to be considered a diagnostic/characteristic product ion. What is the rationale in addition to its presence/absence that the loss of H₂O is characteristic of oxPC?
8. Where is 18O₂ integrated into the oxPC? Was this determined? This would be beneficial to know.

9. $^{18}\text{O}_2$ labeling is a nice way of getting around artificial oxidation by $^{16}\text{O}_2$ from the MALDI process, but there are concerns about off-target oxidation by any labile oxygen species or ROS ($^{18}\text{O}_2$) during the MALDI process. This would be considered artificial and/or off-target oxidation, not "endogenous oxidation". It may be challenging to experimentally determine this; perhaps a description in the discussion section is appropriate.

10. Laser ablation can cause oxidation and how does use of $^{18}\text{O}_2$ compensate for this?

"Visualization of PUFA+ O_2 -PCs in liver tissue through ^{18}O labeling"

1. It was concluded that the data "suggested that in living animals, the inhaled $^{18}\text{O}_2$ was consumed for fatty acid oxidation". Is there literature precedence for this?

2. SI Figure 5 does not clearly show full conversion from $^{16}\text{O}_2$ to $^{18}\text{O}_2$. What is the conversion rate (estimated) for $^{18}\text{O}_2$ uptake from SI Fig 5?

3. What quality control measure were in place to ensure that the flow of $^{18}\text{O}_2$ -containing gas is not modifying oxidative profiles?

4. The use of "without background noise" is too absolute. It would be more appropriate to state limited background noise.

5. Were there any other oxPCs imaged?

Discussion:

1. "Furthermore, the above compounds were also observed in animal disease models". "Models" should be changed to "model". There was only one model in the manuscript.

2. The use of "unbiased" to describe analytical method should be revised.

3. The interchangeably use of oxPL and oxPC needs to be revised. All experiments were geared to oxPC. What is the connection to make conclusions applicable to oxPL in general? Can data and results be fully extended to oxPL? Does it only extend to other GPL?

4. There is still the question as to how specific and sensitive the MSI labeling experiment are. A more detailed description as to why it is specific and sensitive is needed. Are there other product ions more specific than loss of water that can be used? Why weren't MSI experiments done in the negative ion mode?

5. MALDI imaging of other oxidized GPL will pose a challenge due to inadequate sensitivity. MALDI imaging is inherently challenging for lipids that are not abundant. An added discussion on this topic is recommended.

Methods:

“Peroxidation of 16:0-18:2-PC by autoxidation or CuSO₄ + AsA”

1. Chloroform is used for auto-oxidation and lipid extraction. A rationale for why the use of chloroform in lipid extraction does not jeopardize peroxidation experiments is needed.
2. There are concerns about solvent interaction with oxidized PCs during sample prep (reference from LipidMaps: <https://www.lipidmaps.org/resources/tutorials/videos.php>)
 - a. Extractions happen after oxidation: are there any off-target oxidation during the Bligh and Dyer extraction? The procedure uses chloroform as a solvent and chloroform's oxidized byproduct (phosgene) is known to be reactive with lipids. Normally this would not matter, but it seems like the unquenched oxidation reactions immediately precedes this extraction.
 - b. Auto-oxidation experiments with chloroform: what controls are in place to ensure observed oxidation products are from auto-oxidation experiment and not solvent interaction of chloroform extraction?
 - c. Is there data to show that the oxPC were only formed from the intended reaction and not from interactions with solvents?
3. This point matters when talking about the endogenous levels of oxPCs in treated mice, how appropriate is a chloroform-based lipid extraction for analyzing oxidized lipids? Were there any special usage of antioxidants for quenching? What quality measures were in place to ensure oxidation reported was from toxicity?

“LC/HRMS/MS analysis of oxidized PUFA-PCs”

1. 10 uL injection seems high. Was the same injection amount used for positive and negative ion modes?
2. Keep consistent the use of acronyms: e.g., use of MS/MS or dd-MS/MS. Why use dd-MS/MS in method section?
3. Need to verify mass spec conditions for positive ion mode.
4. A description of method used for determining semi-quant data needs to be added. It is mentioned that an internal standard is used in animal experiments. Was an internal standard used for all experiments?
5. How are lipid IDs collated between positive and negative ion modes?

“Peak alignment and detection using Compound Discoverer 3.1”

1. Include parameters used for positive ion mode.
2. What was the rationale for ratio of oxidized to non-ox of >2?

“APAP-treated mouse model and detection of endogenously generated oxPCs”

1. Why was an LPC standard used and not a PC standard? Why not include more than one internal standard?
2. What was the extraction efficiency? And how was data normalized?

“In-vivo $^{18}O_2$ labeling of oxidized 16:0-PUFA-PCs”

1. More details on liver tissue is needed. How much tissue was used? How was tissue amount normalized (wet weight, protein, DNA)?

“MALDI-MS imaging of PUFA+O₂-PCs”

1. A more detailed description of MSI experiments is needed. The description should be sufficient enough for others to reproduce the data
 - a. MALDI MSI experimental parameters (including resolution, raster size, imaging speed/path and laser power) are needed.
 - b. Matrix application is importance, a description of this is needed.

Figures:

Figure 1:

1. Retention time should be tR not RT
2. Precursor ion for all MS/MS spectrum should be labeled

Figure 3:

1. Was there significance for E-G graphs?

Figure 4:

1. Precursor ion for all MS/MS spectrum should be labeled

Figure 5:

1. Precursor ion for all MS/MS spectrum should be labeled
2. MSI figures: what was the resolution, what is the peak unlabeled in spectrum (d) next to labeled (-H₂18O), for (f) why include a different image color background with no scale?

References:

1. Reference pages are mixed up (pages 27-28 should come after page 32)

Reviewer #2 (Remarks to the Author):

Matsuoka et al. present interesting work on the analysis of oxidized phospholipids using mass spectrometry. They used different in vitro oxidation strategies to produce a wide range of oxidation products of PUFA containing phospholipids and subsequent non-targeted MS analysis for their structural identification. Based on these findings they were able to detect oxPCs from tissue extracts of a mouse model for acute liver failure. Using heavy isotope labeling they were furthermore able to detect oxPC species directly from tissue using tandem MS MALDI-MSI.

The manuscript is well prepared and contains the appropriate references. Figures are informative and well prepared. Presented data seems very solid and based on well devised and carefully executed experiments.

I agree with the authors on the importance of the field of oxidized membrane lipids and also on the reasons that is often somewhat overlooked or neglected due to the difficulties in analysis. To this end, the authors present a valuable new analytical strategy that could be of interest for a wide readership in many fields of research.

Overall I would recommend to publish the article with some minor changes as described below:

Detailed comments:

Page 4: Authors claim that PCs are the most abundant PLs in mammals. The claim needs a suitable reference. While PCs are the most common PL, PE comes in a very close second and other classes may show high abundances in specific tissue types. The manuscript would gain from some speculation or comment on how well your findings for oxPCs would translate to oxPEs or other lipid classes and also to plasmalogens. Do you expect a strong influence of the structure of the headgroup on oxidation?

Page 5: Authors claim that PCs are most commonly found with one SFA or MUFA while the other is often a PUFA. Again this needs a suitable reference. It could be argued, that in some tissue types also PLs with two PUFA chains are not uncommon. How difficult would the analysis of an oxPL with

two PUFAs be? Also, do you have any indication if the sn position of the PUFA has an influence on oxidation pathways?

Page 8: In an alternative to MS/MS also ion mobility separation could be used to separate epoxide, hydroxide and hydroperoxide species (<https://doi.org/10.1021/acs.analchem.0c02605>). This could be included either here or maybe as part of a short outlook near the end of the manuscript.

Page9: Authors claim that 103 of the 155 identified species are “new”. Please specify what you mean with new. Are these products of new never described oxidation pathways or are these species not found in any library?

Page 10: In order to get a feeling on which oxidation pathways are prevalent in the different in vitro methods an additional S-figure analogous to figure 4a would be very helpful that compares signal intensities for the different oxidation species generated by the different methods.

Page 14 and also page 18 (This finding agreed with the fact that laser irradiation during MALDI/MS analysis enhances the formation of reactive oxygen species, such as hydroxyl radical): While the use of ^{18}O is a very clever way of differentiating between oxidation “in vivo” and during the MALDI experiment, it also significantly complicates the experimental setup. Would it also be possible to reduce oxidation during the MALDI process? The cited literature (ref 39) explains the oxidation by reactive hydroxyl groups present in the matrix. Have you tried using MALDI matrices like norharmane or 1,5 diamino-naphthalin (DAN) that do not include hydroxyl groups to reduce the unwanted oxidation side reactions during the MALDI process? Please comment on that either here or in a short outlook.

Page 21: electro spray not electron spray

Figure 5: The diagram in d suggests that the mice were in normal air 105 minutes after APAP treatment and were then transferred to ^{18}O air for 15 mins. Is that correct? If so, please add that information to the text. What color is the staining used in e? Please add the information on pixel size to the figure caption and scale bars to at least one subfigure in d and f.

Reviewer #3 (Remarks to the Author):

In this study, the authors generated a structural library for oxidized phospholipids by chemically oxidizing a group of phospholipids (PCs), then they used this to putatively identify related structures generated in a mouse model of oxidative stress, with the aim of improving structural identification of this class of lipids in model systems. In the first experiment, AAPH +/- hemin was used to chemically oxidize PC species in vitro. Next, an untargeted method was used to annotate the lipids

that formed, followed by MS/MS. This is a fairly standard approach nowadays for comparing two sample types and identifying distinguishing features.

Following this, different oxidation methods were undertaken and compared. Similar approaches have been taken by several other groups, including Maria Fedorova who using a similar method generated a software tool to manage this, called LPPTiger (<https://home.uni-leipzig.de/fedorova/software/lpptiger/>). The current study cites this paper but has not made use of this in silico tool for their analysis, instead preferring to generate their own library experimentally. It is not clear why they needed to generate a new library given this tool is openly available.

Overall, the structures found are to be expected from oxidation (since mechanisms of chemical oxidation of lipids are well known for decades), and so the level of innovation and novelty is rather reduced. Despite this, the authors should be commended for their careful approach to the actual MS analysis itself which is good quality.

The model used in mice involved treatment with acetaminophen, which is well known to be hepatotoxic in overdose through oxidative mechanisms. While it has also been long known how the hepatotoxicity occurs, less is known about the specific oxPL that are generated during this oxidative insult. While this study profiles oxPL generated in this model, the study is rather descriptive in nature and it is unclear how this information would help with design of treatments or other preventative strategies or whether the lipid oxidation is mechanistically important in this form of liver injury. This point has been discussed in a recent review which concluded that lipid peroxidation, although it occurs, is not a relevant injury mechanism in acetaminophen induced liver injury (<https://rosj.org/index.php/ros/article/view/143>, 10.1093/toxsci/kfg220)

The acetaminophen mouse experiment lipidome analysis includes putative structural identification of epoxy-hydroxy-oxPL, however this is very speculative and without any standards it is very difficult to be convinced the annotation is really proven in terms of position of oxidation and precise functional group assignment. Some ions that are labelled in the fragmentation spectra are extremely minor, and while this is to be expected in this type of experiment (a complex mix of many lipids fragmenting together in the MS will generate complex data), great care is needed with assigning any more than simply carbon chain length, rings/double bonds and number of oxygenations when there are no standards that can be compared with the lipids being analysed. The inclusion of an 18-O method is a nice innovation to improve exclusion of artefacts.

They mention that LIPID MAPS doesn't include many oxPL, however on checking this, we find that LMSD includes 259, while the in silico database includes over 44K structures.

LIPID MAPS recently updated the shorthand nomenclature, including for oxPL, and adapting to this naming convention would be recommended (<https://www.jlr.org/content/61/12/1539.long>).

Supplementary Figure 6 shows an impact of oxPC on gene expression in RAW cells using PCR. This data isn't mentioned in the results section and only in the discussion and its relevance to the study is unclear. Also, it's not novel since there are many studies showing oxPL can regulate TLR signaling in macrophages, both activation and inhibition depending on experimental design. They state that this may mean that oxPL could be considered DAMPs. However, there is an extensive literature for many years already existing that refers to oxPL as DAMPs, and this is widely acknowledged and accepted.

Responses to the Reviewer's comments

From Reviewer #1

Overall Comments; ... This manuscript provides a starting point for identifying novel oxidized lipids that can give insight into disease progression. The concept and general approach is of high interest and suitable for Nature Communication. Yet, there are a number of issues that need to be resolved in order to justify acceptance for publication.

Answer-1

Thank you for the recognition of our work and valuable suggestions. We have conducted additional experiments and revised the manuscript accordingly. Please find our itemized responses below.

Major issues:

• Inconsistent use of lipid nomenclature and interchangeable use of PL/GPL and oxPL/oxPC.

➤ Answer-1-1, 4, 6, 48 (See below)

• Justification for focusing on PCs in the context of framing the analytical methodology towards lipids in general.

➤ Answer-1-3 (See below)

• Experimental methods as described lack detail to reproduce data. This is especially true for aligning positive and negative ion mode data, calculation of relative abundance (semi-quantitation), and mass spec imaging experiments (e.g., matrix application, MALDI acquisition parameters).

➤ Answer-1-8, 10, 11, 25, 26, 31, 32, 52, 53, 55, 56, 58, 59, 60, 61, 62 (See below)

• Description of quality control measures to ensure lipid oxidation was a result of planned experiments (e.g., auto-oxidation, AAPH, APAP) and not an artifact of sample preparation (use of chloroform in lipid extraction) or sample analysis.

➤ Answer-1-51, 52 (See below)

• The amount of lipid oxidation in the context of non-oxidized lipids was not addressed. This is important in terms of evaluating the robustness and applicability of the developed analytical method. In other words, how much oxidation needs to be present in order to detect oxPCs? What is the extent of lipid oxidation (as measured by oxPC) in the context of non-oxidized lipids? This is relevant in terms of APAP toxicity and any other disease/injury model. In effect, how sensitive is the developed method for detection of oxPCs?

➤ Answer-1-9, 14, 21, 24 (See below)

Introduction:

1. The use of PL instead of GPL is an oversight. PL would include sphingomyelin (among others) but GPL would not. GPL commonly contain PUFAs whereas SM does not. A clarification of lipid nomenclature is needed. It is advised to use GPL instead of PL.

Answer-1-1

We are grateful for your helpful suggestion. As you have mentioned, sphingomyelins (SMs) generally have a lower content of PUFAs than GPLs, such as PC, phosphatidylethanolamine (PE), phosphatidylserine (PS), and phosphatidylinositol (PI).^[R1-1,2]

^[R1-1] Kerwin, J.L. et al. Identification of molecular species of glycerophospholipids and sphingomyelin using electrospray mass spectrometry. *J Lipid Res* **35**,1102-1114(1994)

^[R1-2] Puri, P. et al. A lipidomic analysis of nonalcoholic fatty liver disease. *Hepatology* **46**,1081-1090(2007)

Based on the reviewer's comments, we have revised the following sentences in the introduction section.

[Page-3, line-2]

In biological systems, **glycerophospholipids (GPLs)** are involved in cell membrane construction, metabolism and signal transition regulation, and other vital processes. However, **GPLs** contain polyunsaturated fatty acids (PUFAs); therefore, they easily form **oxidized GPLs (oxGPLs)** under the action of reactive oxygen species generated in intracellular organelles.

2. The use of “unbiased” should not be used for analytical techniques. At every stage of the analytical process bias introduced. For example, the choice of lipid extraction technique will bias what lipids are observed and their abundance. A more appropriate choice would be to use “discovery” or “un/non-targeted”. The authors use non-targeted in other places of the paper and it should remain consistent.

Answer-1-2

Thank you very much for your helpful suggestion. We completely agree with your comments. As suggested, we have corrected “unbiased identification” as “discovery” in the following sentence.

[Page-4, line-10]

To address the above-mentioned challenges, we herein employed high-resolution mass spectrometry (HRMS)-based non-targeted analysis, which is a powerful method for the **discovery** of unknown molecular species.

3. Intro provides background on PL and oxPL but all experiments focus on oxPC. There should be a stronger rationale and description for why experimental approach focuses exclusively on oxPC. Why are oxPCs specifically considered and what is there biological relevance? Are PCs more prone to oxidation? Is it concentration dependent?

Answer-1-3

We appreciate your comments. OxPCs have been accepted as important contributors to cell death and inflammation. For instance, truncated oxPCs, such as 1-palmitoyl-2-(5'-oxo-valeroyl)-sn-glycero-3-phosphocholine, induce cell death driven by

lipid peroxidation (LPO) (ferroptosis).^[R1-3] Furthermore, oxidized 1-palmitoyl-2-arachidonoyl-sn-glycero-3-phosphocholine regulates the immune responses of macrophages^[R1-4] and dendritic cells.^[R1-5] Moreover, E06 antibody, which can neutralize oxPCs, ameliorates the pathogenesis of arteriosclerosis^[R1-6] and non-alcoholic steatohepatitis.^[R1-7] However, identification of oxPCs has been limited. Indeed, the current lipid database, viz. LIPIDMAPS (www.lipidmaps.org/), contains only 53 oxPC species.

Furthermore, it is unclear whether PC is more susceptible to oxidation than other GPLs, including PE, PS, and PI. Poyton, et al.^[R1-8] have suggested that negatively charged GPLs, such as PE, show higher oxidation susceptibility than PC, due to the strong affinity to metal ions. However, considering the high abundance of PC in biological tissues^[R1-2] and bioactivities of oxPC^[R1-3-7], there is no doubt that oxPC plays critical roles in numerous pathological events.

^[R1-3] Stamenkovic, A. et al. Oxidized phosphatidylcholines trigger ferroptosis in cardiomyocytes during ischemia-reperfusion injury. *Am J Physiol Heart Circ Physiol* **320**, 1170-1184 (2021)

^[R1-4] Di Gioia, M. et al. Endogenous oxidized phospholipids reprogram cellular metabolism and boost hyperinflammation. *Nat Immunol* **21**, 42-53 (2020)

^[R1-5] Zanoni, I. et al. An endogenous caspase-11 ligand elicits interleukin-1 release from living dendritic cells. *Science* **352**, 1232-1236 (2016)

^[R1-6] Que, X. et al. Oxidized phospholipids are proinflammatory and proatherogenic in hypercholesterolaemic mice. *Nature* **558**, 301-306 (2018)

^[R1-7] Sun, Z. et al. Neutralization of oxidized phospholipids ameliorates non-alcoholic steatohepatitis. *Cell Metab* **31**, 189-206 (2020)

^[R1-8] Poyton, MF. et al. Cu²⁺ binds to phosphatidylethanolamine and increases oxidation in lipid membranes. *J. Am. Chem. Soc.* **138**, 1584-1590 (2016)

Based on the reviewer's comments, we have revised the manuscript as follows.

[Page-2, line-2]

Although oxidized phosphatidylcholines (oxPCs) play critical roles in numerous pathological events, the type and production site(s) of endogenous oxPCs remain unknown because of the lack of related structural information and analytical methods. Herein, a library of 465 oxPCs was constructed using high-resolution mass spectrometry (HRMS)-based non-targeted analytical methods and employed to detect 70 oxPCs in mice with acetaminophen-induced acute liver failure.

[Page-3, line-6]

In particular, oxGPLs derived from phosphatidylcholine (PC), which is an abundant GPL in several biological tissues such as the liver and kidney, have been shown to be responsible for several pathological events, such as cell death and inflammation. For example, truncated oxPCs, such as 1-palmitoyl-2-(5'-oxo-valeroyl)-sn-glycero-3-phosphocholine, induce cell death driven by lipid peroxidation (LPO) (ferroptosis), whereas oxidized 1-palmitoyl-2-arachidonoyl-sn-glycero-3-phosphocholine regulates the immune responses of macrophages and dendritic cells. These bioactivities suggest that oxPCs are associated with the pathogenesis of several oxidative stress-related diseases, such as liver and cardiovascular diseases.

The structural diversity of oxGPLs, which can be present in epoxide, hydroxide, hydroperoxide, aldehyde, and carboxylic acid forms, implies that numerous oxPCs could be detected as markers of the above-mentioned LPO-related diseases. However, the number of endogenous oxPCs detected in animal disease models and clinical samples is much lower than expected, as many conventional studies on oxPCs have reported only few measurable molecular species (e.g., lipid hydroxides

and peroxides) with clear structures. Indeed, the current lipid database, viz. LIPIDMAPS (www.lipidmaps.org/), contains **only 53 oxPCs**, which again, is ascribed to the lack of structural information. These **oxPCs** are produced through complicated and multistep radical reactions and have complex difficult-to-elucidate structures. Conversely, considering the difficulty of understanding the production mechanism of these **oxPCs**, there is no doubt that a large number of **oxPCs** would be detectable. In fact, Anthonymuthu, et al. described that the number of identified oxidized lipids is negligible compared with the total possible number that can be generated from known lipid structures. Hence, the biggest limitation of oxidative lipidomics is this inability to identify and quantify all oxidized lipids.

[Page-4, line-12]

The tandem mass spectrometry (MS/MS) parameters of each compound, such as the mass-to-charge ratio (m/z) value and fragmentation pattern, provide valuable information on **oxPC** structure. Furthermore, the recent developments in HRMS and data processing methods enable the comprehensive annotation of large numbers of MS/MS peaks detected for complex samples.

Herein, MS techniques were used to construct an MS/MS library containing 465 **oxPCs**. The established library enabled the comprehensive analysis of 70 **oxPCs** formed during acetaminophen (APAP)-induced acute liver failure (ALF) in mice. Furthermore, to clarify the site of **oxPC** formation, we used *in-vivo* ^{18}O stable isotope labeling and matrix-assisted laser desorption/ionization-tandem MS-MS imaging (MALDI-MS/MS/MSI) for the visualization of endogenous **oxPCs**. Thus, our results should elucidate the role of endogenous **oxPCs** in the pathogenesis of LPO-related diseases.

[Page-10, line-10]

Although recent studies have shown that LPO and its products are involved in APAP-induced hepatocellular death, the identities of **oxPCs** generated in the liver of mice with APAP-induced ALF remain unclear.

[Page-20, line-6]

Although **oxPCs** are recognized as critical molecules regulating cell death and immune response, the number of identified bioactive **oxPCs** remains small because of the lack of related structural information and analytical methods. Therefore, the developed library and visualization method enable the discovery of new bioactive **oxPCs** and shed light on their physiological and/or pathological significance in LPO-related diseases.

4. Throughout the paper there is an interchangeable use of oxPL and oxPC. The authors need to do distinguish between the two or justify the interchangeable use. It is too simplistic to suggest that experimental data obtained for oxPC can be readily translated to oxPL.

Answer-1-4

Thank you for your kind comments. We apologize for the inconsistency in usage of terminology. Based on the reviewer's comments, we have revised the introduction, results, and discussion sections. (Please see Answer-1-3)

Results:

“Annotation of oxidized 16:0-PUFA PCs by non-targeted approach”

1. Why was PC16:0/18:2 used as the example to highlight the develop method? This structure represented the simplest of the

three standards. Why not use 20:4 or 22:6 for the paper figures to show the complexity and robustness of the approach?

Answer-1-5

Thank you very much for your helpful suggestion. As suggested, we have added the following sentences in the manuscript and HRMS/MS spectra of oxPCs derived from PC16:0/20:4 and PC16:0/22:6 in **Supplementary Figure 4**.

[Page-8, line-15]

In addition, oxPCs derived from PC16:0/20:4 and PC16:0/22:6, which should have more complicated structures, could be identified following the above-mentioned procedures (**Supplementary Figure 4**). Finally, we identified 155 oxPC species derived from the three PC16:0/PUFAs (**Supplementary Table 1**).

Supplementary Figure 4. Structural analysis of oxPCs derived from PC16:0/20:4 and PC16:0/22:6. a–b. HRMS/MS product ion spectra of PC16:0_10:3;O (m/z 704.4138, $t_R = 7.0$ min) (a) and PC16:0_22:6;O₂ (m/z 882.5493, $t_R = 8.0-10.5$, 10.5–13.0 min) (b), which are derived from PC16:0/20:4 and PC16:0/22:6, respectively.

2. The authors describe PC as being diacyl which is certainly true yet PCs and GPL in general have a portion of their structures present in alkyl and/or vinyl ether. The presence of ether GPL is not discussed. What is the rationale for not addressing ether PCs?

Answer-1-6

Thank you for your valuable comments. As you have mentioned, in this research, we developed a structural library for diacyl oxPCs, but not for alkyl or vinyl ether oxPCs. In general, biological tissues contain higher content of diacyl PC than ether PCs. Indeed, we measured the amounts of PCs in the mouse liver by LC/HRMS analysis in the positive ion mode and revealed

that diacyl PC was more abundant than ether PC (**Figure R1**). Therefore, we focused on diacyl-type oxPCs in this research. However, as the construction of ether oxPC library could also be important to future studies, we have added the following sentences in the manuscript.

Figure R1. Semiquantitative analysis of diacyl and ether PCs in the mouse liver. Semiquantified concentrations of hepatic PCs in 7-week-old C57BL/6J male mice. The peak areas from extracted ion chromatogram for each oxPCs were determined by full-scan LC/HRMS in positive ion mode. The semiquantitative values were calculated using the ratio of the MS peak area of individual oxPCs to that of PC15:0/18:1-d₇ (internal standard). Data are presented as the mean + standard deviation of experiments repeated six times.

[Page-5, line-13]

Therefore, we first attempted to clarify the oxidative modification patterns of major PUFAs (18:2, 20:4, and 22:6) by non-targeted analysis using three **diacyl PC16:0/PUFAs**, namely 1-palmitoyl-2-linoleoyl-sn-glycero-3-phosphocholine (PC16:0/18:2), 1-palmitoyl-2-arachidonoyl-sn-glycero-3-phosphocholine (PC16:0/20:4), and 1-palmitoyl-2-docosahexaenoyl-sn-glycero-3-phosphocholine (PC16:0/22:6). Next, based on the analyzed modification patterns, the substructures of other oxPCs were assessed using an *in-silico* method (**Supplementary Figure 1**).

[Page-19, line-14]

Despite the wide variety of biological PLs, we identified and detected oxGPLs derived only from **diacyl PCs**. Thus, the range of target oxGPLs should be further extended to **ether oxPCs and oxGPLs belonging to other lipid classes**, e.g., phosphatidylethanolamines, phosphatidylserines, and phosphatidylinositols.

3. Sentence “given the high oxidative resistance of SFA’s and MUFA’s,...” should have a reference associated with it.

Answer-1-7

We appreciate this helpful suggestion. SFAs and MUFAs have been considered to show high oxidative resistance compared with PUFAs,^[R1-9] because of the high bond dissociation energy of the C-H bond at the methylene positions.^[R1-10]

^[R1-9] Yin, H. et al. Free radical lipid peroxidation: mechanisms and analysis. *Chem Rev* **111**, 5944-5972 (2011)

^[R1-10] Pratt, D.A. et al. Theoretical calculations of carbon-oxygen bond dissociation enthalpies of peroxy radicals formed in the autoxidation of lipids. *J Am Chem Soc* **125**, 5801-5810 (2003)

Based on the reviewer's comments, we have added the following text and references in the manuscript.

[Page-5, line-10]

Given the high oxidative resistance of SFAs and MUFAs⁸ due to the high bond dissociation energy of the C-H bond at the methylene positions,¹⁵ the structural diversity of oxPCs would depend on the oxidative modification of PUFA moieties.

4. The use of negative and positive ion modes should be explained in regards to the benefits of each. A brief description of their complementary features would suffice.

Answer-1-8

Thank you very much for your comment. HRMS/MS analysis in the negative ion mode is a reliable diagnostic method for elucidating PC structures.^[R1-11] This technique has also been applied to monitor the levels of individual oxPCs using the product ion specific to each oxPC.^[R1-12] However, since it is known that the fragmentation efficiency of various lipids varies greatly depending on the type of FA side chains.^[R1-13]

In contrast, PC shows high ionization efficiency in the positive ion mode, due to its quaternary ammonium cation with a permanently positive charge.^[R1-14] In addition, previous studies demonstrated that the ionization efficiency of lipid molecular species including phosphatidylcholine was mainly dependent on their polar head groups and slightly dependent on their FA side chains.^[R1-15-17] Therefore, HRMS in the positive ion mode can be used for the highly sensitive and relative quantification of PC; however, it cannot provide detailed information on PC structures, including the chemical formulae of the two fatty acyl moieties.

To examine the quantitative potential of HRMS in the positive ion mode, we analyzed the ionization responses of seven commercially available PCs (**Supplementary Figure 16a**). MS peak areas of PCs measured by positive ion mode were much higher than those measured by negative ion mode (**Supplementary Figure 16b**). Furthermore, the ionization efficiencies for the individual PCs were not completely consistent, but were relatively close (**Supplementary Figure 16c**). From these results, we concluded that HRMS in the positive ion mode is applicable to the semiquantification of (ox)PCs.

Supplementary Figure 16. HRMS measurements of the three standard oxPCs. **a.** Chemical structures of the PCs employed in this experiment. **b.** Each MS peak areas of 10 pmol PCs measured under either negative (black) or positive (red) ion mode. MS peak areas of PCs measured by positive ion mode were much higher than those measured by negative ion mode. **c.** Calibration curves for PC, pmol vs. MS peak area measured under the positive ion mode. Data are presented as the mean \pm standard deviation of experiments repeated three times. **d.** The summary of detection methods employed in this study. DDA; data dependent acquisition, PRM; parallel reaction monitoring.

[R1-11] Nakanishi, H. et al. Analysis of oxidized phosphatidylcholines as markers for oxidative stress, using multiple reaction monitoring with theoretically expanded data sets with reversed-phase liquid chromatography/tandem mass spectrometry. *J Chromatogr B Analyt Technol Biomed Life Sci* **877**, 1366-1374 (2009).

[R1-12] Aoyagi, R. et al. Comprehensive analyses of oxidized phospholipids using a measured MS/MS spectra library. *J Lipid Res* **58**, 2229-2237 (2017)

[R1-13] Schumann, K. et al. Quantitative fragmentation model for bottom-up shotgun lipidomics. *Anal Chem* **91**, 12085-12093 (2019)

[R1-14] Pulfer, M. and Murphy, R.C. Electrospray mass spectrometry of phospholipids. *Mass Spectrom Rev* **22**, 332-364 (2003)

[R1-15] Han, X. & Gross, RW. Electrospray ionization mass spectroscopic analysis of human erythrocyte plasma membrane phospholipids. *Proc Natl Acad Sci USA* **91**, 10635-10639 (1994)

[R1-16] Han, X. & Gross, RW. Shotgun lipidomics: electrospray ionization mass spectrometric analysis and quantitation of cellular lipidomes directly from crude extracts of biological samples. *Mass Spectrom Rev* **24**, 367-412 (2005)

[R1-17] Jung, HR. et al. High throughput quantitative molecular lipidomics. *Biochim Biophys Acta* **1811**, 925-934 (2011)

Based on the above-mentioned data, we used the negative ion mode in the structural analysis (**Figure 1**) and monitored the individual changes in oxPCs (**Figure 2, 3d-g**), whereas the positive ion mode was employed in the semiquantification of (ox)PCs (**Figure 4a** and **Supplementary Figure 5, 7**). Furthermore, for the semiquantification of oxPCs in the positive ion mode, we used PC15:0/18:1-d₇ as an internal standard. Hence, we have added the following text in the results and methods sections.

[Page-6, line-5]

PC16:0/PUFAs were oxidized by AAPH or AAPH + hemin. After the extraction of lipids from these samples, both extracts were mixed and analyzed by high-performance liquid chromatography coupled with high-resolution mass spectrometry (LC/HRMS) in negative ion mode, **which is a reliable diagnostic tool for lipid structural elucidation.**

[Page-9, line-10]

A particularly large increase in the levels of PC hydroperoxides, such as PC16:0_18:2;OOH, was observed after stimulation by AAPH or autoxidation (**Figure 2b**), whereas in the presence of metal ions, i.e., after AAPH+hemin or CuSO₄+AsA stimulation, a particularly high increase was observed in the levels of mono-oxidized or di-oxidized PCs (**Figures 2c, d**). **In addition, we investigated the abundances of non-oxidized and oxidized PC16:0/18:2 by using LC/HRMS analysis in the positive ion mode, which is a useful method for highly sensitive and relative quantification of PCs.**

[Page-22, line-5]

LC/HRMS/MS analysis was performed using the Nexera LC system (Shimadzu Co., Kyoto, Japan) coupled with a high-performance benchtop quadrupole Orbitrap mass spectrometer (Q Exactive, Thermo Fisher Scientific, Waltham, MA) that was equipped with an **electro spray ionization source. HRMS(/MS) in negative ion mode is a reliable diagnostic method for elucidating PC structures. This technique has been also applied to monitor the levels of individual oxPCs using the product ions specific to each oxPC. In contrast, since the ionization efficiency of lipid molecular species including phosphatidylcholine was mainly dependent on their polar head groups and slightly dependent on their FA side chains,⁴⁴ HRMS in positive ion mode can be used for highly sensitive and relative quantification of PCs (Supplementary Figure 16). We confirmed that the ionization efficiencies for individual oxPCs in the positive ion mode were not completely consistent, but relatively close (Supplementary Figure 16a, c). However, this method cannot provide detailed information on oxPC structure, including the chemical formulae of the two fatty acyl moieties. Therefore, we used the negative ion mode in structural analysis (Figure 1) and monitored the individual changes in oxPCs (Figure 2, 3d-g), whereas the positive ion mode was employed for comparative analysis of the abundances of different oxPCs (Figure 4a and Supplementary Figure 5, 7) (Supplementary Figure 16d).**

[Page-25, line-9]

Semiquantitative analysis of non-oxidized and oxidized PCs

LC/HRMS in the positive ion mode was employed for semiquantitative analysis for (ox)PCs. The semiquantitative values were calculated using the ratio of the MS peak area of each oxPC to that of PC15:0/18:1-d₇, which was used as an internal standard.

5. What is the abundance of oxPC in context of total lipid extracts? What is the extent of oxidation for standards and biological samples?

Answer-1-9

We appreciate your helpful suggestion. As you have mentioned, the abundance of oxPCs in samples can be vital to understand the basal levels of oxPCs in pathological states and evaluate the applicability of our developed method. Therefore, we sought to semiquantify the amounts of non-oxidized or oxidized PCs in standard and biological samples. LC/HRMS in the positive ion mode was employed for semiquantification of (ox)PCs. The semiquantified concentrations of (ox)PCs were calculated from the ratio of the MS peak area of individual oxPCs to that of PC15:0/18:1-d₇ (internal standard).

We first investigated the abundances of non-oxidized and oxidized PC16:0/18:2 generated through several LPO inducers (**Supplementary Figure 5**). The proportion of oxPCs, including carboxylic oxPCs, truncated oxPCs, and full-length oxPCs, in the total amount of PCs differed depending on the LPO inducers. (AAPH, 41.13 ± 0.48%; autoxidation, 16.40 ± 0.17%; AAPH+hemin, 69.11 ± 1.36%; and CuSO₄+AsA, 79.73 ± 0.76%).

Supplementary Figure 5. Semiquantitative analysis of oxidized PC16:0/18:2 generated in several LPO systems. a. Semiquantified concentrations of non-oxidized and oxidized PC16:0/18:2 generated in different LPO systems (black, control; magenta, AAPH; blue, autoxidation; green, AAPH+hemin; and light blue, CuSO₄+AsA). The peak areas from extracted ion chromatogram for each oxPCs were determined by full-scan LC/HRMS in positive ion mode. The semiquantitative values were calculated using the ratio of the MS peak area of individual oxPCs to that of PC15:0/18:1-d₇ (internal standard). Data are presented as the mean + standard deviation of experiments repeated three times. **b.** Proportions of non-oxidized (gray) or oxidized PCs (green, carboxylic oxPCs; orange, truncated oxPCs; red, full-length oxPCs) in the total amount of PCs.

Next, we semiquantified hepatic PCs in mice treated with APAP for 2 h (**Figure 4a** and **Supplementary Figure 7**). The amount of oxPCs markedly increased with APAP treatment, whereas that of non-oxidized PCs remain unchanged. The amounts of oxPCs in control and APAP-treated mice were calculated as 0.62 ± 0.16% and 1.81 ± 0.30%, respectively. In the case of biological samples, the levels of full-length oxPCs (control mice, 0.54 ± 0.11%; APAP-treated mice, 1.56 ± 0.24%)

were higher than that of truncated oxPCs (control mice, $0.08 \pm 0.05\%$; APAP-treated mice, $0.25 \pm 0.07\%$).

Figure 4. Epoxide and/or hydroxide motif-containing PC16:0_PUFA;O2 with remarkably increased levels in the early phase of APAP-induced ALF. a. Semiquantified concentrations of individual oxPCs in vehicle- (black) and 2-h APAP-treated (red) groups. The peak areas from extracted ion chromatogram for each oxPCs were determined by full-scan LC/HRMS in positive ion mode. The semiquantitative values were calculated using the ratio of the MS peak area of individual oxPCs to that of PC15:0/18:1-d₇ (internal standard). Data are presented as the mean + standard deviation of experiments repeated six times.

Supplementary Figure 7. Semiquantitative analysis of non-oxidized PCs in the mouse liver. Semiquantified concentrations of hepatic PCs in vehicle- (black) and 2-h APAP-treated (red) groups. The peak areas from extracted ion chromatogram for each oxPCs were determined by full-scan LC/HRMS in positive ion mode. The semiquantitative values were calculated using the ratio of the MS peak area of individual oxPCs to that of PC15:0/18:1-d₇ (internal standard). Data are presented as the mean + standard deviation of experiments repeated six times.

This information has been added to the manuscript as follows.

[Page-9, line-10]

A particularly large increase in the levels of PC hydroperoxides, such as PC16:0_18:2;OOH, was observed after stimulation by AAPH or autoxidation (**Figure 2b**), whereas in the presence of metal ions, i.e., after AAPH+hemin or CuSO₄+AsA stimulation, a particularly high increase was observed in the levels of mono-oxidized or di-oxidized PCs (**Figures 2c, d**). **In addition, we investigated the abundances of non-oxidized and oxidized PC(16:0/18:2) through LC/HRMS analysis in the**

positive ion mode, which is a useful method for highly sensitive and relative quantification of PCs. We observed that the proportions of oxPCs, including full-length oxPCs and truncated oxPCs, in the total amount of PCs also differed depending on the LPO inducers (**Supplementary Figure 5**).

[Page-12, line-2]

Next, to clarify the detailed structures of oxPCs the levels of which increased in the early phase of APAP-induced ALF (2 h after APAP treatment), we used the positive ion mode for semiquantitative LC/HRMS analysis (**Figure 4a**). The semiquantified concentrations of oxPCs increased with 2 h APAP treatment (the percentages of oxPCs in vehicle- and APAP-treated mice were calculated to be $0.62 \pm 0.16\%$ and $1.81 \pm 0.30\%$, respectively.), whereas that of PCs remained unchanged (**Supplementary Figure 7**). Furthermore, the level of full-length oxPCs ($1.56 \pm 0.24\%$) was higher than that of truncated oxPCs ($0.25 \pm 0.07\%$) in APAP-treated mice.

6. Why was $O/N > 2$ used for detection of oxPC? What is the criteria for using this value?

Answer-1-10

Thank you for your comment. In our preliminary investigation, we considered the influence of signals with oxidized sample (O)/non-oxidized sample (N) intensity ratio (values of O/N intensity ratio > 1.2, 2, 5, and 10) on comprehensive oxPCs detection in *in vitro* sample extracts using a background subtraction approach. Thus, to optimize O/N intensity ratio, we analyzed 10 reported oxPCs derived from PC16:0/20:4 (PC16:0_4:1;O, PC16:0_5:1;O, PC16:0_6:2;O, PC16:0_7:2;O, PC16:0_7:2;O₂, PC16:0_8:2;O₂, PC16:0_12:4;O₂, PC16:0_20:4;oxo, PC16:0_20:4;O₂ and PC16:0_20:4;O₃), which have been identified in previous studies (**Supplementary Table 2**), by using each O/N intensity ratio. For O/N intensity ratio 5 and 10, some of reported oxPCs which is relatively low intensity were not detected due to LC/HRMS chromatogram in non-oxidized sample with a high background noise (**Figure R2**).

On the other hand, all 10 oxPCs could be detected by O/N intensity ratio 1.2 and 2. Compared O/N intensity ratio 1.2 to 2, the number of false positive peaks (*i.e.* background noise and isobaric peaks derived from biological matrix) with O/N intensity ratio 1.2 was much more than with O/N intensity ratio 2. We concluded that value of $O/N > 2$ is the most suitable in terms of the number of false positive and negative peaks. Therefore, in this study, signals with O/N intensity ratios was set to $O/N > 2$.

Figure R2. LC/HRMS chromatogram for PC16:0_7:2;O₂ in non-oxidized and oxidized sample.

7. The authors say that the HRMS/MS spectra were carefully annotated. Was annotation done *de novo* (*i.e.*, by hand) or via the aid of software?

Answer-1-11

Thank you for your comments. All oxPCs identified in this study were carefully annotated *de novo* (i.e., manually), without the aid of software. We have clarified the same in the revised manuscript as follows.

[Page-6, line-14]

The picked m/z peaks in HRMS/MS spectra were carefully annotated *de novo* (i.e., manually) after the interpretation of molecular substructures by checking three typical product ions derived from the PC head group and two fatty acyl groups (Figure 1a).

8. P. 7 describes “loss of methyl group in the choline moiety were observed in HRMS/MS spectra” as a major product ion. This should be clarified to say it is loss of CH₃ with product being deprotonated via loss of formate anion.

Answer-1-12

Thank you for your kind suggestion. As suggested, we have corrected the following sentences.

[Page-7, line-2]

Under this condition, specific product ions corresponding to the two fatty acyl groups and the loss of CH₃ group with the product being deprotonated via loss of the formate anion were observed in HRMS/MS spectra, as reported previously.

9. Along these lines (#7), all precursor ions in MS/MS spectra should be labeled (Figure 1, 4, 5). This will help clarify the loss of CH₃ and its deprotonated form. The precursor ion is the formate anion.

Answer-1-13

Thank you for this helpful suggestion. As suggested, we have labeled the precursor ions in the HRMS/MS spectra shown in Figure 1b, c, f and Figure 5c. However, precursor ions in the HRMS/MS spectra shown in Figure 4c, d, e, could not be observed, which might be due to the use of high collision energy (35 eV) for a detailed structural analysis.

Figure 1. Non-targeted analysis and annotation of oxidized PC16:0/PUFAs.

b. HRMS/MS product ion spectrum of PC16:0_8:1;O (m/z 680.4138, retention time (t_R) = 6.6 min). The structure and proposed fragmentation of the ion with m/z 680.4138 are also shown. **c.** Averaged HRMS/MS product ion spectrum of PC16:0_18:2;O₂ (m/z 834.5496, t_R = 9.0–12.0 min). The structure and proposed fragmentation of the ion with m/z 834.5496 are also shown. **f.** HRMS/MS product ion spectrum of PC16:0_9:0;COOH (m/z 664.4190, t_R = 3.0 min). The structure and proposed fragmentation of the ion with m/z 664.4190 are also shown.

Figure 5. Visualization of endogenous oxPCs by a combination of MALDI/MS/MS/MSI and $^{18}\text{O}_2$ labeling.

c. HRMS/MS spectra of parent ions with m/z 794.5682, 818.5682, and 842.5682 observed for the APAP-treated group (120 min in $^{18}\text{O}_2$ air).

10. Second sentence of p. 8. “The precursor ion at m/z 834.5496...” is confusing. Clarification of this sentence would be helpful.

Answer-1-14

Thank you for your comment. Accordingly, we have corrected the following sentence.

[Page-7, line-10]

The HRMS/MS spectrum of m/z 834.5496 showed two product ions from oxidized fatty acyls (m/z 311.2222 and 293.2124, eluting at 9.0–10.5 and 11.0–12.0 min, respectively) (Figures 1c, d and Supplementary Figure 3a, b).

11. Figure 1C-E, why was EIC taken for 9-12 minutes? Why not take EIC of each peak, 9-10.5 and 11-12? What do the MS/MS spectra look like with each peak? Do the intensities of 293/311 change? I suggest adding figure (maybe SI) showing these differences.

Answer-1-15

We appreciate this helpful suggestion. As suggested, we have added HRMS/MS spectra of PC16:0_18:2;O2 at 9–10.5 and 11–12 min as Supplementary Figure 3a, b. HRMS/MS spectrum of PC16:0_18:2;O2 at 9–10.5 min specifically showed the product ion of m/z 311.2230, while that of PC16:0_18:2;OOH at 11–12 min showed the product ion of 293.2124.

Supplementary Figure 3. HRMS/MS spectra of PC16:0_18:2;O2 in either negative or positive ion mode at t_R 9–10.5 and 11–12 min. a–b. HRMS/MS spectrum of m/z 834.5496 acquired under negative ion mode at 9–10.5 (a) and 11–12 min (b), respectively.

[Page-7, line-10]

The HRMS/MS spectrum of m/z 834.5496 showed two product ions from oxidized fatty acyls (m/z 311.2222 and 293.2124, eluting at 9.0–10.5 and 11.0–12.0 min, respectively) (Figures 1c, d and Supplementary Figure 3a, b).

12. Fig 1E, the positive ion mode MS/MS spectrum should be included. The discussed fragment ion of loss of H₂O should be shown.

Answer-1-16

Thank you for your kind indication. As suggested, we have added HRMS/MS spectra of PC(16:0/C₁₈H₃₁O₄) obtained under the positive ion mode at 9–10.5 and 11–12 min as Supplementary Figure 3c, d.

Supplementary Figure 3. HRMS/MS spectra of PC16:0_18:2;O2 in either negative or positive ion mode at t_R 9–10.5 and 11–12 min. c–d. HRMS/MS spectrum of m/z 790.5598 acquired under the positive ion mode at 9–10.5 (c) and 11–12 min (d), respectively.

[Page-7, line-15]

The HRMS/MS spectrum at 9.0–10.5 min showed a fragment ion due to H₂O loss (m/z 772.5293), which supported the presence of epoxide and/or hydroxide derivatives (Supplementary Figure 3c). In contrast, the spectrum at 11.0–12.0 min showed a fragment ion due to the loss of H₂O₂ (m/z 756.5544), which suggested the presence of hydroperoxide derivatives (Supplementary Figure 3d).

13. It was unclear why product ions m/z 311 and 293 were classified as epoxide/hydroxide and hydroperoxide, respectively. Was this determined experimentally or from literature?

Answer-1-17

Thank you for your comments. Previous studies have demonstrated that lipid hydroperoxides show the characteristic fragmentation of hydroperoxide moieties (a neutral loss of H₂O) in MS/MS analysis, whereas no fragmentation of epoxide and/or hydroxide moieties is observed.^[R1-18-19]

[R1-18] Miotto, G. et al. Insight into the mechanism of ferroptosis inhibition by ferrostatin-1. *Redox Biol* **28**, 101328 (2020)

[R1-19] Helmer, P.O. et al. Hydroperoxylated vs dihydroxylated lipids: differentiation of isomeric cardiolipin oxidation products

by multidimensional separation techniques. *Anal Chem* **92**, 12010-12016 (2020)

Therefore, we have modified the text in the manuscript with appropriate in-text citation as follows.

[Page-7, line-10]

The HRMS/MS spectrum of m/z 834.5496 showed two product ions from oxidized fatty acyls (m/z 311.2222 and 293.2124, eluting at 9.0–10.5 and 11.0–12.0 min, respectively) (**Figures 1c, d** and **Supplementary Figure 3a, b**). **According to a previous study**, these product ions were ascribed to the functional isomers of 18:2:O₂, e.g., hydroperoxide and epoxy-hydroxide.¹⁹

14. What was the abundance of the oxPCs containing carboxylates? The paper mentions that some oxPC were carboxylates. What was their abundance compared to other oxPC and un-oxidized PCs?

Answer-1-18

Thank you very much for your comments. **Supplementary Figure 5** shows the abundances of non-oxidized and oxidized PC(16:0/18:2), including carboxylic oxPCs, generated via several LPO systems. The proportion of carboxylic oxPCs was within the range of 0.60–3.61% (AAPH, $0.60 \pm 0.01\%$; autoxidation, $0.93 \pm 0.02\%$; AAPH+hemin, $2.68 \pm 0.10\%$; and CuSO₄+AsA, $3.61 \pm 0.08\%$), and the levels of carboxylic oxPCs were particularly elevated in metal-associated LPO systems, such as the AAPH+hemin and CuSO₄+AsA treatments (Please see the response in Answer-1-9).

15. It is claimed that “103 new species” were identified from experiments. All species discussed to this point in the paper were identified previously in literature. This statement should be clarified as to what it means to be a “new species”.

Answer-1-19

We appreciate this helpful suggestion. Until now, many researchers have identified a number of oxPCs using MS/MS-based approaches (publications focusing on the structural analysis of oxPCs are listed in **Supplementary Table 2**). Contrastingly, in this study, we have determined the chemical formulae of 155 oxPC species derived from three PC(16:0/PUFAs) using non-targeted analysis (**Supplementary Table 1**). Moreover, among these, 103 oxPCs have never been reported in previous studies, as listed in **Supplementary Table 2**. Therefore, we defined these 103 oxPCs as new species, which are highlighted in **Supplementary Table 1**. We have modified the manuscript as follows.

[Page-8, line-17]

Finally, we identified 155 oxPC species derived from the three PC(16:0/PUFAs) (**Supplementary Table 1**). **Surprisingly, among them, 103 oxPCs were novel, which have never been reported in previous studies, as listed in Supplementary Table 2.**

16. It is my understanding that the identified “155 species” were from sn-2 fatty acyl of 18:2. How does the analysis of 18:2 translate to more complex fatty acyls of 20:4 and 22:6. Again, the use of a more complicated sn-2 fatty acyl (e.g., 20:4 or 22:6) as an example would help highlight the identification process.

Answer-1-20

Thank you very much for your comments. We apologize for creating any confusion. The identified 155 oxPCs were derived

from three PC16:0/PUFAs, namely PC16:0/18:2, PC16:0/20:4, and PC16:0/22:6 (**Supplementary Table 1**).

17. A major question I have is what is the percentage oxidation from these experiments? What is the amount of oxidized vs non-oxidized? A figure showing the amount of oxidation would be very helpful for evaluating the effectiveness of developed method to detect oxPC.

Answer-1-21

Thank you very much for your comments. The amounts of oxPCs generated *in vitro* were determined using semiquantitative LC/HRMS analysis in the positive ion mode (AAPH, $41.13 \pm 0.48\%$; autooxidation, $16.40 \pm 0.17\%$; AAPH+hemin, $69.11 \pm 1.36\%$; and CuSO₄+AsA, $79.73 \pm 0.76\%$). These data have been added to the revised paper. (Please see Answer-1-9)

“Effect of LPO inducers on oxPC production”

1. No comments

“Validation of methodology: oxPC analysis in mice with APAP-induced ALF”

1. Is there literature precedence for statement that “MT suppressed hepatocellular death by inhibiting mitochondrial oxidative stress”? If so, add reference? If not, this is speculative and unless have other data it should be worded as such.

Answer-1-22

Thank you very much for your comments. Mito-TEMPO (MT) has been reported to suppress APAP-induced hepatotoxicity by inhibiting mitochondrial oxidative stress.^[R1-20]

^[R1-20] Du, K., Farhood, A. & Jaeschke, H. Mitochondria-targeted antioxidant Mito-Tempo protects against acetaminophen hepatotoxicity. *Arch Toxicol* **91**, 761-773 (2017).

Thus, we have added the following text along with a reference as follows.

[Page-10, line-13]

Plasma alanine aminotransferase (ALT) levels were remarkably elevated 4 h after APAP administration (300 mg/kg, intraperitoneal injection (i.p.)). They were attenuated by treatment with N-acetylcysteine (NAC; 500 mg/kg, i.p.) or mitochondria-targeted antioxidant, Mito-TEMPO (MT; 20 mg/kg, i.p.), **which has been reported to suppress APAP-induced hepatotoxicity by inhibiting mitochondrial oxidative stress,**²⁸ 1 h after APAP administration (**Figure 3b**).

2. Figure 3 E-G, what about the 18:0 or 18:1 sn-1 species? Were monoxides only in 16:0? Was it dependent on sn-2 structure or did sn-1 contribute?

Answer-1-23

We appreciate these helpful indications. PC monoxides observed in the liver extracts contained not only 16:0, but also 18:0 (**Figure 3** and **Supplementary Figure 6**). Furthermore, time profiles of oxPCs containing the same oxPUFA moiety were similar. These results might suggest that SFAs in *sn-1* position do not exert a minimal effect on oxPC formation.

Supplementary Figure 6. Time-dependent differences in HRMS/MS peak areas obtained in the negative ion mode for PC18:0_oxPUFAs formed after APAP administration. Mouse liver samples were collected 1, 2, 4, 8, and 24 h after APAP (300 mg/kg in saline) administration. **a.** PC18:0_20:4;O, **b.** PC18:0_18:2;O2, and **c.** PC18:0_17:3;O2. LC/HRMS/MS data were obtained by using PRM in the negative ion mode. Data are presented as the mean + standard deviation of experiments repeated five or six times. ** $P < 0.01$, compared with the vehicle-treated group. ## $P < 0.01$, compared with the APAP-treated group 4 h post-treatment.

Based on the reviewer's comments, we have added the above-mentioned data and the following text in the results section.

[Page-11, line-8]

On the contrary, the levels of some oxPCs, including those of newly identified species, remarkably increased 2 h after APAP treatment (Figures 3f, g). Furthermore, time profiles of oxPCs containing the same oxPUFA moiety were similar (Supplementary Figure 6), which might indicate that SFAs in *sn-1* position do not exert a minimal effect on oxPC formation. Here, we would like to emphasize that the production of these oxPCs increased prior to the drastic increment in the ALT level 4 h after APAP administration (Figure 3d).

3. What were the abundances of the oxidized products compared to the un-oxidized lipids? What is the percentage of lipid peroxidation based on identified structures?

Answer-1-24

Thank you very much for your comments. The percentages of oxPCs in control and APAP-treated mice were calculated to be $0.62 \pm 0.16\%$ and $1.81 \pm 0.30\%$, respectively. These data have been added to the revised paper. (Please see Answer-1-9)

“Detailed structural analysis of PUFA+O2-PCs generated in the early phase of APAP-induced ALF”

1. Why was positive ion mode used for determining abundance?

2. How were semi-quantitative experiments carried out? This should be explained in the methods section.

3. Was determination of peak area used for abundance? This is implied in the text. Does peak area correspond to abundance? Why not say use abundance instead of peak area?

Answer-1-25

Thank you very much for your comments. As mentioned in Answer-1-8, since PC is permanently positively charged, HRMS analysis in the positive ion mode is a powerful method for sensitive detection and semiquantification of (ox)PCs. Therefore,

we have used LC/HRMS in positive ion mode for semi-quantification of oxPCs. The semiquantified concentrations of (ox)PCs were calculated from individual MS peak areas using the calibration curve of PC15:0/18:1-d₇ as the standard. (Please see Answer-1-8)

4. Going back to an earlier item, the use of positive and negative ion modes should be described. Why use positive ion mode for abundance and negative ion mode for structure characterization? Is there a need do to both for verification and/or confirmation?

Answer-1-26

We appreciate the helpful comments. As mentioned in Answer-1-8, HRMS(/MS) analysis in the negative ion mode is a powerful method for structural analysis,^[R1-11] because it allows obtaining information on substructures of PCs, such as PC head group and fatty acyl group.^[R1-12] In contrast, HRMS analysis in the positive ion mode can be used for highly sensitive and relative quantification of PCs; however, it cannot provide detailed structural information.

Therefore, we utilized the negative ion mode for the structural analysis (**Figure 1**) and monitored the individual changes in oxPCs (**Figure 2, 3d-g**), whereas the positive ion mode was employed for a comparative analysis of the abundances of different oxPCs (**Figure 4a** and **Supplementary Figure 5, 7**).

5. Did the chromatography reveal isomers for the linoleate species? Could this be used to confirm presence of hydroperoxides?

Answer-1-27

Thank you very much for your comments. **Supplementary Figure 11b** shows the EIC of PC16:0_18:2;O₂, and the multiple peaks were considered to be derived from the structural isomers of PC16:0_18:2;O₂. Furthermore, since these peaks were eluted at 9.0–12 min, they appeared to be a mixture of PC16:0_18:2;O₂ in epoxide, hydroxide, and hydroperoxide forms. However, we could not determine the precise structures of the individual isomers owing to the difficulty in separating each peak and a wide variety of candidate structures.

6. The use of “comprehensive analysis of endogenous oxPCs...” appears to suggest that all oxPCs have been identified which likely is not the case. I suggest revising to use exhaustive or thorough instead of comprehensive.

Answer-1-28

We appreciate this helpful suggestion. We completely agree with the reviewer’s comments, and hence, we have replaced “comprehensive analysis” with “exhaustive analysis” in the manuscript as follows.

[Page-10, line-4]

To validate our methodology, we applied it to an **exhaustive** analysis of endogenous oxPC generation in the liver of mice with APAP-induced ALF (**Figure 3a**).

[Page-13, line-8]

These results indicated success in the **exhaustive** analysis of endogenous oxPCs generated in APAP-treated mice, revealing that PUFA+O₂-PCs containing epoxide and/or hydroxide motifs were predominantly generated in the early phase of APAP-induced liver injury.

[Page-17, line-11]

The **exhaustive** analysis also revealed the unique kinetic profiles of endogenous oxPCs. In particular, the levels of PUFA+O₂-PCs predominantly increased in the early phase of liver injury (**Figure 4**), and the accumulation regions matched the areas of CYP2E1 expression and GSH depletion (**Figure 5**), indicating the importance of oxPCs in ALF progression.

[Page-20, line-2]

In conclusion, an HRMS/MS-based library of 465 oxPCs was developed and used for the **exhaustive** analysis of endogenous oxPCs generated during APAP-induced ALF.

[Page-36, line-6]

Figure 3. Exhaustive analysis of endogenous oxPCs in APAP-treated mice.

“Analysis of oxPC formation by MALDI/MS/MSI based on 18O labeling”

1. This section has a different tone/writing style. It is as if it was copy and pasted.

Answer-1-29

Thank you for your comment. We apologize for not using an appropriate tone while writing this section. Accordingly, we have carefully revised this section.

2. The use of “in test tubes” has not been used in manuscript up to this point. I suggest using in vitro.

Answer-1-30

Thank you very much for your helpful suggestion. We have replaced “test tube” with “*in vitro*” in the manuscript as follows.

[Page-14, line-2]

Hence, we first determined whether **oxPCs generated *in vitro* could be detected through MALDI-MS/MS analysis.**

[Page-14, line-5]

To determine **the pair of precursor and product ions employed in MALDI-MS/MS analysis**, we used PC34:2;O₂ oxidized by AAPH + hemin treatment *in vitro*.

3. The specific mention of an “ion trap instrument” appears disjointed. Why not say “a mass spectrometer with tandem mass spectrometry capability was used”? The exact mass spec can be described in methods.

Answer-1-31

We appreciate this helpful suggestion. We completely agree with the reviewer’s comments. Accordingly, we have corrected the following sentences in the results section and added detailed information on MALDI-MS/MS in the methods section.

[Page-14, line-2]

Hence, we first determined whether **oxPCs generated *in vitro* could be detected by through MALDI-MS/MS analysis.**

[Page-27, line-10]

MALDI-MS/MS/MSI of PC PUFA;¹⁸O₂

Sample preparation for MALDI-MS/MS/MSI was performed as described previously. Briefly, thin liver sections (8 μm) were prepared using a cryomicrotome (CM3050, Leica Microsystems, Tokyo, Japan) and attached onto indium tin oxide-coated glass slides (Bruker Daltonics GmbH, Leipzig, Germany). The slides were coated with a solution (50 mg/mL) of 2,5-dihydroxybenzoic acid in 80% aqueous ethanol by manual spraying with an artistic brush (Procon Boy FWA Platinum, Mr. Hobby, Tokyo, Japan).

MALDI-MS/MS and -MSI experiments were performed using a MALDI linear ion trap mass spectrometer (MALDI LTQ XL; Thermo Fisher Scientific, Bremen, Germany), equipped with a 60-Hz N₂ laser ($\lambda = 337$ nm). **The laser energy and the raster step size were set at 32 μJ and 60 μm, respectively. Auto gain control (where the ion trap is filled with the optimum number of ions) was not used in this experiment.** Mass spectra were acquired in the positive ion mode. The MS/MS transitions for PC PUFA;¹⁸O₂ imaging were as follows: m/z 794.5 → 774.5 (neutral loss of H₂¹⁸O) for PC34:2;¹⁸O₂, m/z 818.5 → 798.5 for PC36:4;¹⁸O₂, and m/z 842.5 → 822.5 for PC38:6;¹⁸O₂-PC, **with a precursor ion isolation width of m/z 1.0.** After sample analysis, ion images were reconstructed using ImageQuest v. 1.0.1 software (Thermo Fisher Scientific).

4. *The use of “ionic transition” should be revised. Do you mean ion transition, precursor to product ion transition?*

Answer-I-32

Thank you for your comment. We have used the term “ionic transition” to indicate the set of precursor and product ions. However, as you have mentioned, this expression can lead to misinterpretation. Thus, we have revised the following sentences.

[Page-14, line-5]

To determine **the pair of precursor and product ions employed in MALDI-MS/MS analysis**, we used PC34:2;¹⁸O₂ oxidized by AAPH + hemin treatment *in vitro*.

5. *“flown into test tubes” should be revised to more appropriately describe the experiment.*

Answer-I-33

Thank you for the helpful suggestion. As suggested, we have revised the sentence in the manuscript as follows.

[Page-14, line-14]

When ¹⁸O₂ air was filled in test tubes containing the reaction solution, ¹⁶O atoms in the oxPUFA moiety were converted to ¹⁸O (**Supplementary Figure 10**). Furthermore, an m/z 794.5 → 774.5 transition, corresponding to the loss of H₂¹⁸O from PC34:2;¹⁸O₂, was detected by both MALDI-MS/MS and LC/HRMS/MS analyses, but was not observed in the presence of ¹⁶O₂ (**Supplementary Figure 9c, d**).

6. *What is the conversion and/or uptake of ¹⁸O₂ in these in vitro experiments?*

Answer-I-34

Thank you for your comment. **Supplementary Figure 10** shows EICs of PC34:2;¹⁸O₂ and PC34:2;¹⁶O₂ generated *in vitro*. As expected, when ¹⁸O₂ air was filled in test tubes containing the reaction solution, ¹⁶O atoms in the oxPUFA moiety were converted to ¹⁸O.

Supplementary Figure 10. ¹⁸O conversion of PC34:2;O₂ generated under AAPH+hemin peroxidation system in ¹⁸O₂ air. Extracted ion chromatograms (EICs) for ions with *m/z* 790.5598 (black), 792.5640 (blue), and 794.5682 (red) corresponding to PC34:2;¹⁶O₂, PC34:2;¹⁶O¹⁸O, and PC34:2;¹⁸O₂, respectively. Left, right: PC34:2;O₂ generated under AAPH+hemin peroxidation system in ¹⁸O₂ and normal air, respectively.

As required, we have detailed the above-mentioned data in the manuscript as follows.

[Page-14, line-14]

When ¹⁸O₂ air was filled in test tubes containing the reaction solution, ¹⁶O atoms in the oxPUFA moiety were converted to ¹⁸O (Supplementary Figure 10). Furthermore, an *m/z* 794.5 → 774.5 transition, corresponding to the loss of H₂¹⁸O from PC34:2;¹⁸O₂, was detected by both MALDI-MS/MS and LC/HRMS/MS analyses, but was not observed in the presence of ¹⁶O₂ (Supplementary Figure 9c, d).

7. Although I agree the use of 18O2 is a nifty experiment, I am still concerned that the lack of specificity the loss of “H2O even H218O” is not unique enough to be considered a diagnostic/characteristic product ion. What is the rationale in addition to its presence/absence that the loss of H2O is characteristic of oxPC?

Answer-1-35

Thank you very much for your comments. Previous studies have demonstrated that HRMS/MS spectrum of full-length oxPC show a characteristic product ion derived from the neutral loss of H₂O.^[R1-21] However, as you have mentioned, there is still concern over the loss of H₂¹⁸O being specific to oxPC detection.

Since ¹⁸O₂ could be taken up only at the oxidized sites (Supplementary Figure 13), a neutral loss of H₂¹⁸O can be a specific diagnostic product ion for “oxidized” PCs. Indeed, MS peak signals derived from ¹⁸O-labeled oxPCs could not be observed for the liver sections of control mice (Figure 5d).

Furthermore, to examine the specificity of the loss of H₂¹⁸O to ¹⁸O-labeled oxPC detection, we obtained the EIC of PC34:2;¹⁸O₂ generated *in vivo* by LC/HRMS/MS in positive ion mode (Figure R3). If the loss of H₂¹⁸O was specific to ¹⁸O-labeled oxPC, any MS peaks derived from isobaric compounds should not be detected in the EIC. Considering that PC34:2;O₂ was eluted at 9–12 min (Supplementary Table 1), HRMS peaks eluted at other retention times were derived from isobaric compounds. As a result, the precursor and product ion pair of *m/z* 794.5682 → 774.5535, corresponding to a neutral loss of

H₂¹⁸O, was observed only at 9–12 min, even with a large mass tolerance (500 ppm). This result suggests that a neutral loss of H₂¹⁸O, but not H₂O, can be used for the specific detection of ¹⁸O-labeled oxPCs.

[R1-21] Reis, A., *et al.* Radical peroxidation of palmitoyl-linoleyl-glycerophosphocholine liposomes: Identification of long-chain oxidised products by liquid chromatography-tandem mass spectrometry. *J Chromatogr B* **855**, 186-199 (2007).

EIC of 794.5682 > 774.5535 (NL of H₂¹⁸O)

Figure R3. LC/HRMS/MS analysis of PC34:2;¹⁸O₂ generated *in vivo*. EICs recorded in positive ion mode for the precursor ion at *m/z* 794.5682. Product ions at *m/z* 774.5535 correspond to a neutral loss of H₂O. Mass tolerance was set to be 5 (left) and 500 (right) ppm.

8. Where is ¹⁸O₂ integrated into the oxPC? Was this determined? This would be beneficial to know.

Answer-1-36

We appreciate this helpful suggestion. To determine the position where ¹⁸O₂ was integrated into oxPCs, the detailed structure of PC PUFA;¹⁸O₂ generated *in vivo* was analyzed by LC/HRMS/MS in negative ion mode. **Supplementary Figure 13** shows the HRMS/MS spectrum of *m/z* 838.5582 corresponding to PC16:0_18:2;¹⁸O₂. The characteristic product ions at *m/z* 255.2325 and 315.2311 correspond to 16:0 and 18:2+¹⁸O₂, respectively. This result indicates that ¹⁸O₂ was integrated into the 18:2 moiety. Furthermore, ions with *m/z* 141.1171, 157.1121, 159.1164, 173.1070, and 187.1220 suggested the presence of ¹⁸O atom at C9-C10 position of linoleate. In contrast, the ions with *m/z* 115.1009 and 213.1383 showed ¹⁸O atom integration at the C12-C13 position. Moreover, these results were well consistent with the estimated structures of PC16:0_18:2;¹⁸O₂ generated under normal air condition (**Figure 4c**).

HRMS/MS: *m/z* 838.5582; PC16:0_18:2;¹⁸O₂

Supplementary Figure 13. LC/HRMS/MS identification and plausible structures of PC16:0_18:2;¹⁸O₂ generated in mice treated for 2 h with APAP and inhaling ¹⁸O₂ air. HRMS/MS spectra of the product ion at *m/z* 838.5582 corresponding to PC16:0_18:2;¹⁸O₂.

The above-mentioned data have been added to the manuscript as follows.

[Page-17, line-14]

The detected species contained epoxide and hydroxide moieties possibly formed via metal-induced peroxidation (**Figures 2d and 4c–e**). **In addition, our detailed structural analysis of PC16:0_18:2;¹⁸O₂ generated in mice that had inhaled ¹⁸O₂, and were treated for 2 h with APAP validated the estimated structures (Supplementary Figure 13).**

9. ¹⁸O₂ labeling is a nice way of getting around artificial oxidation by ¹⁶O₂ from the MALDI process, but there are concerns about off-target oxidation by any labile oxygen species or ROS (¹⁸O₂) during the MALDI process. This would be considered artificial and/or off-target oxidation, not "endogenous oxidation". It may be challenging to experimentally determine this; perhaps a description in the discussion section is appropriate.

Answer-1-37

Thank you very much for your helpful comments. Although the amount of ¹⁸O₂ in the atmosphere is extremely small (relative isotopic abundance of oxygen; ¹⁶O: 99.757%, ¹⁷O: 0.038%, and ¹⁸O: 0.205%), the possibility of artificial oxidation caused by ¹⁸O₂ cannot be ruled out completely. However, it is quite difficult to examine this phenomenon. Thus, we have added the following sentences in the discussion section.

[Page-19, line-3]

Although the possibility of artificial oxidation caused by ¹⁸O₂ cannot be ruled out completely, the specific ¹⁸O-labeling of endogenous oxPCs **certainly** helped eliminate this unfavorable event.

10. Laser ablation can cause oxidation and how does use of ¹⁸O₂ compensate for this?

Answer-1-38

Thank you very much for your comment. By inhalation of ¹⁸O₂ air, ¹⁶O₂ in the liver tissue of mice is replaced by ¹⁸O₂, and ¹⁸O-labeled oxPC is produced *in vivo*. Herein, during artificial oxidation by laser ablation, atmospheric oxygen (¹⁶O₂) is considered to modify lipids through oxidation in the liver tissue sections. Furthermore, considering the extremely small amount of ¹⁸O₂ in the atmosphere (relative isotopic abundance of oxygen; ¹⁶O: 99.757%, ¹⁷O: 0.038%, and ¹⁸O: 0.205%), the integration of ¹⁸O₂ into lipids should hardly occur. Indeed, MS peak signals derived from ¹⁸O-labeled oxPCs could not be observed for the liver sections of mice inhaling normal air (**Figure 5**). Therefore, we could selectively detect ¹⁸O-labeled oxPCs generated *in vivo*.

“Visualization of PUFA+O₂-PCs in liver tissue through ¹⁸O labeling”

1. It was concluded that the data “suggested that in living animals, the inhaled ¹⁸O₂ was consumed for fatty acid oxidation”. Is there literature precedence for this?

Answer-1-39

We appreciate your helpful suggestion. Many previous studies have suggested that dissolved molecular oxygen reacts with lipids to produce oxidized lipids.^[R1-9] However, to the best of our knowledge, this is the first study directly showing inhaled

oxygen to be immediately consumed for lipid peroxidation *in vivo*.

[R1-9] Yin, H. et al. Free radical lipid peroxidation: mechanisms and analysis. *Chem Rev* **111**, 5944-5972 (2011)

2. SI Figure 5 does not clearly show full conversion from $^{16}\text{O}_2$ to $^{18}\text{O}_2$. What is the conversion rate (estimated) for $^{18}\text{O}_2$ uptake from SI Fig 5?

Answer-I-40

Thank you very much for your comment. The proportion of ^{16}O and ^{18}O atoms integrated into PC16:0_18:2;O2 was calculated from the MS peak area of PC16:0_18:2;O2 observed in the murine liver extracts (**Supplementary Figure 11**), which was 72.2%. In contrast, considering that the reactivities of $^{16}\text{O}_2$ and $^{18}\text{O}_2$ toward lipids is almost same, the efficiency of ^{18}O in converting lipids to oxidized lipids *in vivo* is thought to be highly dependent on the conversion rate of oxygen in the biological tissue. However, it is very difficult to predict these conversion efficiencies accurately. Nevertheless, since some of the ^{16}O atoms in oxPCs were converted to ^{18}O after only 15 min of $^{18}\text{O}_2$ inhalation (**Figure 5d**), these conversion rates are considered to be relatively fast. We have added this clarification in the manuscript as follows.

[Page-15, line-11]

Furthermore, under this condition, 72.2% of ^{16}O atoms in oxidized fatty acyls were converted to ^{18}O (**Supplementary Figure 11**).

3. What quality control measure were in place to ensure that the flow of $^{18}\text{O}_2$ -containing gas is not modifying oxidative profiles?

Answer-I-41

We appreciate the helpful indication. To examine whether $^{18}\text{O}_2$ air affected oxPC profiles, the amount of oxidized PC16:0/18:2 generated in mice inhaling either normal or $^{18}\text{O}_2$ air was measured by LC/HRMS analysis in the positive ion mode (**Supplementary Figure 12**). Consequently, no significant difference was observed between the oxPC profiles, suggesting that $^{18}\text{O}_2$ inhalation did not affect oxPC formation. These data have been added to the manuscript as follows.

Supplementary Figure 12. Product profiles of oxidized PC16:0/18:2 generated in mice treated with APAP for 2 h and inhaling either normal or $^{18}\text{O}_2$ air.

[Page-15, line-11]

Furthermore, under this condition, 72.2% of ^{16}O atoms in oxidized fatty acyls were converted to ^{18}O (**Supplementary Figure 11**). Moreover, we confirmed that $^{18}\text{O}_2$ inhalation had little effect on the oxPC product profiles (**Supplementary Figure 12**).

4. The use of “without background noise” is too absolute. It would be more appropriate to state limited background noise.

Answer-1-42

Thank you very much for your insightful suggestion. We have replaced “without background noise” with “with limited background noise” in the manuscript as follows.

[Page-15, line-14]

Lastly, we visualized oxPCs generated in the livers of APAP-treated mice by MALDI-MS/MS/MSI, which showed a clear and characteristic distribution of endogenous PC PUFA;¹⁸O₂ with limited background noise.

[Page-16, line-6]

Taken together, the constructed HRMS/MS library of oxPCs and *in-vivo* ¹⁸O labeling allowed us to visualize oxPCs selectively with limited background noise and provided critical information on the oxPC formation site in the tissues of animal disease models.

5. Were there any other oxPCs imaged?

Answer-1-43

Thank you for your comment. We sought to visualize other oxPCs, such as PC36:4;¹⁸O and PC38:6;¹⁸O₃ (**Supplementary Figure 15**). Consistent with the results obtained for PC PUFA;¹⁸O₂, the above-mentioned ¹⁸O-labeled oxPCs were observed only in APAP-treated mice that inhaled ¹⁸O₂ air (**Supplementary Figure 15a**). In addition, we examined their distribution in the liver tissues and found that the localization of these oxPCs matched well with the areas of CYP2E1 expression, as well as the localization of PC(PUFA+¹⁸O₂) (**Supplementary Figure 15b**). These results have been added to the manuscript in the discussion section as follows.

Supplementary Figure 15. Visualization of relatively abundant oxPCs, such as PC36:4;¹⁸O and PC38:6;¹⁸O₃ by MALDI-MS/MS/MSI. a. PC36:4;¹⁸O and PC38:6;¹⁸O₃ are observed only in APAP-treated mice inhaling ¹⁸O₂ air. **b.** The localization of PC36:4;¹⁸O and PC38:6;¹⁸O₃ matches well with the areas of CYP2E1 expression. Scale bar = 1 mm.

[Page-19, line-7]

To the best of our knowledge, this is the first report of endogenous oxGPL visualization in an animal disease model. Furthermore, we also visualized other ¹⁸O-labeled oxPCs generated in APAP-treated mice, such as PC36:4;¹⁸O and PC38:6;¹⁸O₃, by using the techniques detailed herein (Supplementary Figure 15).

Discussion:

1. “Furthermore, the above compounds were also observed in animal disease models”. “Models” should be changed to “model”. There was only one model in the manuscript.

Answer-1-44

Thank you for your helpful suggestion. We completely agree with the reviewer’s comment. We have changed “models” to “model” as follows.

[Page-16, line-16]

However, the above-mentioned compounds were identified in our animal disease model (Figure 3d).

2. The use of “unbiased” to describe analytical method should be revised.

Answer-1-45

Thank you for your helpful suggestion. As suggested, we have corrected “unbiased non-targeted approach” to “non-targeted approach” as follows.

[Page-16, line-17]

These results indicate that a non-targeted approach is necessary for the exhaustive identification of diverse oxPCs.

3. The interchangeably use of oxPL and oxPC needs to be revised. All experiments were geared to oxPC. What is the connection to make conclusions applicable to oxPL in general? Can data and results be fully extended to oxPL? Does it only extend to other GPL?

Answer-1-46

We appreciate your helpful suggestion and completely agree with your comments. As the reviewer mentioned, this study focused on the analysis of oxPC. Therefore, the following sentences in the discussion section have been revised.

[Page-20, line-6]

Although oxPCs are recognized as critical molecules regulating cell death and immune response, the number of identified bioactive oxPCs remains small because of the lack of related structural information and analytical methods. Therefore, the developed library and visualization method enable the discovery of new bioactive oxPCs and shed light on their physiological and/or pathological significance in LPO-related diseases.

Additionally, in the future, the range of target oxPLs should be extended to oxidized lipids belonging to other lipid classes. Since lipid peroxidation mainly occurs at the PUFA moiety,^[R1-9] the oxidation modification patterns identified in this study (**Supplementary Table 1**) can provide useful information to identify and analyze oxidized lipids, other than GPLs, with PUFA structures.

^[R1-9] Yin, H. et al. Free radical lipid peroxidation: mechanisms and analysis. *Chem Rev* **111**, 5944-5972 (2011)

4. There is still the question as to how specific and sensitive the MSI labeling experiment are. A more detailed description as to why it is specific and sensitive is needed. Are there other product ions more specific than loss of water that can be used? Why weren't MSI experiments done in the negative ion mode?

Answer-1-47

Thank you very much for your comments. As mentioned in *Answer-1-35*, we showed that the precursor and product ion pair of m/z 794.5682 \rightarrow 774.5535, corresponding to a neutral loss of H₂¹⁸O, allowed extremely specific detection of PC34:2,¹⁸O₂, even under a large mass tolerance (**Figure R2**). In addition, although we sought to visualize oxPCs by MSI in negative ion mode, no MS signal deriving from oxPC was detected (data not shown), which might be due to the low sensitivity of MSI under the negative ion mode.

Accordingly, we concluded that a neutral loss of H₂¹⁸O, but not H₂O, could be a relatively specific feature for detecting

product ions of ^{18}O -labeled oxPC.

5. MALDI imaging of other oxidized GPL will pose a challenge due to inadequate sensitivity. MALDI imaging is inherently challenging for lipids that are not abundant. An added discussion on this topic is recommended.

Answer-1-48

We appreciate your helpful suggestion and completely agree with your comments. As the reviewer stated, the visualization of oxPCs with a very low concentration *in vivo* remains challenging, even with the methods developed in this study. To overcome this challenge, future studies should aim to improve the sensitivity and specificity of visualization methods. Based on the reviewer's comments, we have added the following sentences in the discussion section.

[Page-19, line-17]

Moreover, the visualization of oxPCs with a very low concentration *in vivo* remains challenging. In order to overcome this challenge, future studies on improving the sensitivity and specificity of visualization methods are warranted.

Methods:

“Peroxidation of 16:0-18:2-PC by autoxidation or CuSO₄ + AsA”

1. Chloroform is used for auto-oxidation and lipid extraction. A rationale for why the use of chloroform in lipid extraction does not jeopardize peroxidization experiments is needed.

Answer-1-49

Thank you for your comment. Firstly, autoxidation of PC16:0/18:2 was performed after removal of chloroform by a stream of N₂ gas as previously reported.^[R1-22] Therefore, the effect of chloroform on oxidation is considered as minimal. We apologize for the lack of clarity, and the text in the methods section has been revised as follows.

^[R1-22] Morgan, A.H. et al. Quantitative assays for esterified oxylipins generated by immune cells. Nat Protoc 5, 1919-1931 (2010).

[Page-21, line-15]

PC16:0/18:2 was oxidized by autoxidation or CuSO₄ + AsA as described previously. For autoxidation, a solution of 500 nmol of PC16:0/18:2 in chloroform (1 mL) was immediately dried using nitrogen gas in a glass tube. The dried samples were incubated at 37 °C for 24 h and then resuspended in methanol (200 μL).

In addition, to examine whether PC can be oxidized by chloroform, we investigated the formation of ^{18}O -labeled oxPCs in chloroform under $^{18}\text{O}_2$ air (**Figure R4**). PC16:0/18:2 was incubated in chloroform under $^{18}\text{O}_2$ air at 37 °C for 4 h, and then semiquantified by LC/HRMS in positive ion mode. AAPH-induced LPO system was used as a positive control. The levels of PC34:2; ^{18}O and PC34:2; $^{18}\text{O}_2$ were not elevated by incubation in chloroform. These results suggest that PCs are unlikely to be oxidized by chloroform during the extraction process.

Figure R4. Formation of oxidized PC16:0/18:2 by incubation in chloroform for 4 h. The reaction solution containing 500 μ M PC16:0/18:2 was incubated in chloroform at 37 °C for 4 h. After 4 h, the mixture was extracted using the modified Bligh and Dyer method. As a positive control, PC16:0/18:2 was oxidized by 50 mM AAPH in PBS.

2. There are concerns about solvent interaction with oxidized PCs during sample prep (reference from LipidMaps: <https://www.lipidmaps.org/resources/tutorials/videos.php>)

a. Extractions happen after oxidation: are there any off-target oxidation during the Bligh and Dyer extraction? The procedure uses chloroform as a solvent and chloroform's oxidized byproduct (phosgene) is known to be reactive with lipids. Normally this would not matter, but it seems like the unquenched oxidation reactions immediately precedes this extraction.

b. Auto-oxidation experiments with chloroform: what controls are in place to ensure observed oxidation products are from auto-oxidation experiment and not solvent interaction of chloroform extraction?

c. Is there data to show that the oxPC were only formed from the intended reaction and not from interactions with solvents?

3. This point matters when talking about the endogenous levels of oxPCs in treated mice, how appropriate is a chloroform-based lipid extraction for analyzing oxidized lipids? Were there any special usage of antioxidants for quenching? What quality measures were in place to ensure oxidation reported was from toxicity?

Answer-1-50

Thank you very much for your helpful comments. When lipid extraction was performed in this study, we added antioxidants, including 100 μ M of dibutylhydroxytoluene (BHT) and ethylenediaminetetraacetic acid (EDTA), to the reaction solutions to quench solvent-based lipid peroxidation. We apologize for the lack of clarity on these experimental procedures. Accordingly, we have added the following sentences in the methods section.

[Page-21, line-8]

After 4 h, the mixture was extracted using the **modified Bligh and Dyer method**. Briefly, mixed solution of 1 mL methanol and 1 mL chloroform, containing 100 μ M dibutylhydroxytoluene and ethylenediaminetetraacetic acid as antioxidants, and 100 nM PC15:0/18:1-d₇ as an internal standard was added to the reaction solution. After 1 min vortex, the organic layer was collected in a glass tube. The extracted solution was dried under a stream of nitrogen gas, and the residue was dissolved in methanol (200 μ L) and stored at -80 °C before injection into the HPLC column.

[Page-22, line-2]

After the reaction, individual reaction mixtures were subjected to extraction as per **the modified Bligh and Dyer method**, and the extracts were dried under a stream of nitrogen gas.

[Page-26, line-9]

Mouse liver and plasma samples were collected, and the liver samples were immediately frozen in liquid nitrogen. Hepatic lipids were extracted from the liver samples according to **the modified Bligh and Dyer method**. Briefly, 1 mL of extraction solution (methanol:chloroform:water = 5:2:2) containing 100 μM dibutylhydroxytoluene, 100 μM ethylenediaminetetraacetic acid, and 100 nM PC15:0/18:1-d₇ was added to a frozen tissue sample (wet weight: approx. 50 mg), and then, the sample was homogenized using a Macro Smash homogenizer. Subsequently, the extraction solutions were sonicated on ice bath for 5 min. After centrifugation (6,000 g, 10 min, 4°C), 700 μL of the supernatant was collected, and then, 235 μL chloroform and 155 μL water were added to the supernatant. The organic layer was collected in a glass tube and dried under a stream of nitrogen gas; the dried residue was dissolved in methanol (200 μL) and stored at -80 °C before performing the LC/HRMS/MS experiments.

Next, as mentioned in *Answer-1-49*, elevation in the formation of oxidized PC16:0/18:2 was not observed in chloroform at 37 °C for 4 h (**Figure R4**). Therefore, we concluded that the effect of chloroform on lipid peroxidation is minimal.

“LC/HRMS/MS analysis of oxidized PUFA-PCs”

1. 10 μL injection seems high. Was the same injection amount used for positive and negative ion modes?

Answer-1-51

Thank you for your comment. As you have mentioned, 10 μL injection amount was used for both positive and negative ion modes. To perform comprehensively quantification of the oxPCs at trace levels, large injection amount was necessary. In preliminary experiment, we evaluated injection amount using 10 μM PC16:0/9:0;9oxo standard solution, and 10 μL injection did not negatively affect to peak shape (**Figure R5**).

Figure R5. LC/HRMS chromatogram for PC16:0/9:0;9oxo in 1, 2, 5 and 10 μL injection amount.

2. *Keep consistent the use of acronyms: e.g., use of MS/MS or dd-MS/MS. Why use dd-MS/MS in method section?*

Answer-1-52

We appreciate your helpful suggestion and apologize for the inconsistency in usage of terminology. Accordingly, we have revised the following sentences in the methods section.

[Page-23, line-13]

The experimental conditions for **MS/MS** and PRM were as follows: resolving power = 17500, automatic gain control target = 1×10^6 , trap fill time = 80 ms, isolation width = ± 0.6 Da, fixed first mass = m/z 80, normalized collision energy = 20 or 35 eV, intensity threshold of precursor ions for **MS/MS** analysis = 3100, apex trigger = 2–4 s, and dynamic exclusion = 2 s. The intensity threshold of precursor ions for **MS/MS** analysis and the dynamic exclusion were set to 1×10^4 and 1 s, respectively. The inclusion list contained 465 precursor ions (m/z) of oxPCs for **MS/MS** analysis. LC/HRMS/MS analysis was controlled using Xcalibur 4.2.47 software (Thermo Fisher Scientific).

3. *Need to verify mass spec conditions for positive ion mode.*

Answer-1-53

Thank you for your helpful comments. We have added the conditions for LC/HRMS(/MS) analysis in the positive ion mode as follows.

[Page-23, line-8]

The ionization conditions were as follows: ionization mode = negative or positive, sheath gas flow rate = 40 arbitrary units, auxiliary gas flow rate = 10 arbitrary units, spray voltage = 2000 V, capillary temperature = 265 °C, S-lens level = 50, and heater temperature = 425 °C.

4. *A description of method used for determining semi-quant data needs to be added. It is mentioned that an internal standard is used in animal experiments. Was an internal standard used for all experiments?*

Answer-1-54

Thank you for your comments. As mentioned in *Answer-1-8* and *-1-9*, LC/HRMS in positive ion mode was employed for semiquantification of (ox)PCs. In addition, the abundances of individual (ox)PCs were calculated from the corresponding MS peak areas using the calibration curve of PC15:0/18:1-d₇ (an internal standard). An internal standard was used in all semiquantification experiments (**Figure 4a** and **Supplementary Figure 5, 7**).

Detailed information on the semiquantitative analysis has been added to the methods section.

(Please see the response: *Answer-1-8 and 1-9*)

5. *How are lipid IDs collated between positive and negative ion modes?*

Answer-1-55

Thank you for your comment. HRMS/MS analysis of each oxPC was performed using both positive and negative ion modes. Next, MS spectral data were obtained for both positive and negative ion modes and the lipid IDs were provided for individual

oxPCs (**Supplementary Table 1**).

“Peak alignment and detection using Compound Discoverer 3.1”

1. Include parameters used for positive ion mode.

Answer-1-56

Thank you for your comment. For non-targeted analysis of oxidized PC16:0/PUFAs, we only used LC/HRMS/MS analysis in the negative ion mode. We apologize for the lack of clarity pertaining to this aspect, and the methods section has been revised as follows.

[Page-24, line-3]

Non-targeted analysis of oxidized PC16:0/PUFAs using Compound Discoverer 3.1

LC/HRMS data for the non-targeted analysis were obtained through LC/HRMS analysis in the negative ion mode. Compound Discoverer 3.1 software (Thermo Fisher Scientific) was used for data processing, including peak alignment (node name = align retention times), peak detection (node name = detect compounds), data grouping (node name = group compounds), and background subtraction (node name = mark background compounds).

2. What was the rationale for ratio of oxidized to non-ox of >2?

Answer-1-57

Thank you for your comment. As mentioned in *Answer-1-10*, we confirmed that value of O/N > 2 is the most suitable in terms of the number of false positive and negative peaks. Therefore, in this study, signals with O/N intensity ratios was set to O/N > 2. (Please see *Answer-1-10*)

“APAP-treated mouse model and detection of endogenously generated oxPCs”

1. Why was an LPC standard used and not a PC standard? Why not include more than one internal standard?

2. What was the extraction efficiency? And how was data normalized?

Answer-1-58

We appreciate your helpful comments. As you have mentioned, although non-oxidized and oxidized PCs were the analytical targets in this study, it was not appropriate to use LPC as an internal standard. Therefore, we conducted the semiquantification of non-oxidized and oxidized PCs using PC15:0/18:1-d₇ as an internal standard (**Figure 4a** and **Supplementary Figure 5, 7**). Moreover, individual MS peak areas were normalized by wet weight of the liver tissues.

“In-vivo 18O2 labeling of oxidized 16:0-PUFA-PCs”

1. More details on liver tissue is needed. How much tissue was used? How was tissue amount normalized (wet weight, protein, DNA)?

Answer-1-59

Thank you for your comments. In this study, approx. 50 mg of mice liver tissues was used for oxPC analysis, and the tissue amount was normalized by wet weight (mg). We have added detailed information on the use of liver tissues in the methods section as follows.

[Page-26, line-8]

Mouse liver and plasma samples were collected, and the liver samples were immediately frozen in liquid nitrogen. Hepatic lipids were extracted from the liver samples according to the modified Bligh and Dyer method. Briefly, 1 mL of extraction solution (methanol:chloroform:water = 5:2:2) containing 100 μ M dibutylhydroxytoluene, 100 μ M ethylenediaminetetraacetic acid, and 100 nM PC15:0/18:1-d₇ was added to a frozen tissue sample (wet weight: approx. 50 mg), and then, the sample was homogenized using a Macro Smash homogenizer. Subsequently, the extraction solutions were sonicated on ice bath for 5 min.

“MALDI-MS imaging of PUFA+O₂-PCs”

1. A more detailed description of MSI experiments is needed. The description should be sufficient enough for others to reproduce the data

a. MALDI MSI experimental parameters (including resolution, raster size, imaging speed/path and laser power) are needed.

b. Matrix application is importance, a description of this is needed.

Answer-1-60

Thank you for your helpful suggestion. We have added further details on the MSI experiments as follows.

[Page-27, line-10]

MALDI-MS/MS/MSI of PC PUFA;O₂

Sample preparation for MALDI-MS/MS/MSI was performed as described previously. Briefly, thin liver sections (8 μ m) were prepared using a cryomicrotome (CM3050, Leica Microsystems, Tokyo, Japan) attached onto indium tin oxide-coated glass slides (Bruker Daltonics GmbH, Leipzig, Germany). The slides were coated with a solution (50 mg/mL) of 2,5-dihydroxybenzoic acid in 80% aqueous ethanol by manual spraying with an artistic brush (Procon Boy FWA Platinum, Mr. Hobby, Tokyo, Japan).

MALDI-MS/MS and -MSI experiments were performed using a MALDI linear ion trap mass spectrometer (MALDI LTQ XL; Thermo Fisher Scientific, Bremen, Germany), equipped with a 60-Hz N₂ laser ($\lambda = 337$ nm). The laser energy and the raster step size were set at 32 μ J and 60 μ m, respectively. Auto gain control (where the ion trap is filled with the optimum number of ions) was not used in this experiment. Mass spectra were acquired in the positive ion mode. The MS/MS transitions for PC PUFA;¹⁸O₂ imaging were as follows: m/z 794.5 \rightarrow 774.5 (neutral loss of H₂¹⁸O) for PC34:2;¹⁸O₂, m/z 818.5 \rightarrow 798.5 for PC36:4;¹⁸O₂, and m/z 842.5 \rightarrow 822.5 for PC38:6;¹⁸O₂-PC, with a precursor ion isolation width of m/z 1.0. After sample analysis, ion images were reconstructed using ImageQuest v. 1.0.1 software (Thermo Fisher Scientific).

Figures:

Figure 1:

1. Retention time should be tR not RT

2. Precursor ion for all MS/MS spectrum should be labeled

Figure 4:

1. Precursor ion for all MS/MS spectrum should be labeled

Figure 5:

1. Precursor ion for all MS/MS spectrum should be labeled

Answer-1-61

Thank you very much for your helpful suggestions. As suggested, we have revised these figures.

Figure 3:

1. Was there significance for E-G graphs?

Answer-1-62

Thank you for your helpful comments. The statistical analysis of data shown in **Figure 3e-g** showed significant differences. Therefore, we have added these data to **Figure 3**.

Figure 5:

2. MSI figures: what was the resolution, what is the peak unlabeled in spectrum (d) next to labeled (-H₂¹⁸O), for (f) why include a different image color background with no scale?

Answer-1-63

Thank you for your comments. Please find our pointwise responses below.

What was the resolution?

The raster step size was set to 60 μm, and the resolution of images is 600 dpi.

What is the peak unlabeled in spectrum (d) next to labeled (-H₂¹⁸O)?

MS peaks next to the labeled peaks (-H₂¹⁸O) correspond to a neutral loss of H₂¹⁶O, which might be derived from the product ions of isobaric lipids.

For (f) why include a different image color background with no scale?

We apologize for creating any confusion. As required, we have added scale bars in **Figure 5d, f**.

References:

1. Reference pages are mixed up (pages 27-28 should come after page 32)

Answer-1-64

Thank you very much for your helpful suggestion. Accordingly, we have placed the reference pages before **Figure legends**.

From Reviewer #2

Overall Comments; ...The manuscript is well prepared and contains the appropriate references. Figures are informative and well prepared. Presented data seems very solid and based on well devised and carefully executed experiments. I agree with the authors on the importance of the field of oxidized membrane lipids and also on the reasons that is often somewhat overlooked or neglected due to the difficulties in analysis. To this end, the authors present a valuable new analytical strategy that could be of interest for a wide readership in many fields of research.

Answer-2

We greatly appreciate the reviewer's recognition of our work and great suggestions. As suggested, we have performed additional experiments and revised the manuscript. Please find our itemized responses below.

Page 4: Authors claim that PCs are the most abundant PLs in mammals. The claim needs a suitable reference. While PCs are the most common PL, PE comes in a very close second and other classes may show high abundances in specific tissue types. The manuscript would gain from some speculation or comment on how well your findings for oxPCs would translate to oxPEs or other lipid classes and also to plasmalogens. Do you expect a strong influence of the structure of the headgroup on oxidation?

Answer-2-1

We appreciate the helpful suggestion. It is known that several biological tissues, such as the liver and kidney, contain more amounts of PCs than other GPL classes,^[R2-1, 2] whereas PE is the most abundant GPL in the brain.^[R2-3]

^[R2-1] Puri, P. et al. A lipidomic analysis of nonalcoholic fatty liver disease. *Hepatology* **46**,1081-1090(2007)

^[R2-2] Hoffmann, K. et al. Lipid class distribution of fatty acids including conjugated linoleic acids in healthy and cancerous parts of human kidneys. *Lipids* **40**,1057-1062(2005)

^[R2-3] Svennerholm, L. Distribution and fatty acid composition of phosphoglycerides in normal human brain. *J Lipid Res* **9**,570-579(1968)

Therefore, we have revised the following sentences and added references in the introduction section.

[Page-3, line-6]

In particular, oxGPLs derived from phosphatidylcholine (PC), which is an abundant GPL in several biological tissues such as the liver and kidney, have been shown to be responsible for several pathological events, such as cell death and inflammation. For example, truncated oxPCs, such as 1-palmitoyl-2-(5'-oxo-valeroyl)-sn-glycero-3-phosphocholine, induce cell death driven by lipid peroxidation (LPO) (ferroptosis), whereas oxidized 1-palmitoyl-2-arachidonoyl-sn-glycero-3-phosphocholine regulates the immune responses of macrophages and dendritic cells. These bioactivities suggest that oxPCs are associated with the pathogenesis of several oxidative stress-related diseases, such as liver and cardiovascular diseases.

In future studies, the range of target oxGPLs should be extended to oxidized lipids belonging to other lipid classes. Since lipid peroxidation mainly occurs at the PUFA moiety,^[R2-4] the oxidation modification patterns identified in this study (**Supplementary Table 1**) can provide useful information to identify and analyze oxidized lipids, other than GPLs, with PUFA structures.

It is unclear whether PCs are more susceptible to oxidation than other GPLs, including PE, PS, and PI. Poyton et al. have suggested that negatively charged GPLs, such as PE, show higher oxidation susceptibilities than PC, due to their strong affinity to metal ions.^[R2-5] However, considering the high abundance of PCs in biological tissues^[R2-1, 2] and bioactivities of oxPCs,^[R2-3-10] there is no doubt that oxPCs play critical roles in numerous pathological events.

^[R2-4] Yin, H. et al. Free radical lipid peroxidation: mechanisms and analysis. *Chem Rev* **111**, 5944-5972 (2011)

^[R2-5] Poyton, MF. et al. Cu²⁺ binds to phosphatidylethanolamine and increases oxidation in lipid membranes. *J. Am. Chem. Soc.* **138**, 1584-1590 (2016)

^[R2-6] Stamenkovic, A. et al. Oxidized phosphatidylcholines trigger ferroptosis in cardiomyocytes during ischemia-reperfusion injury. *Am J Physiol Heart Circ Physiol* **320**, 1170-1184 (2021)

^[R2-7] Di Gioia, M. et al. Endogenous oxidized phospholipids reprogram cellular metabolism and boost hyperinflammation. *Nat Immunol* **21**, 42-53 (2020)

^[R2-8] Zanoni, I. et al. An endogenous caspase-11 ligand elicits interleukin-1 release from living dendritic cells. *Science* **352**, 1232-1236 (2016)

^[R2-9] Que, X. et al. Oxidized phospholipids are proinflammatory and proatherogenic in hypercholesterolaemic mice. *Nature* **558**, 301-306 (2018)

^[R2-10] Sun, Z. et al. Neutralization of oxidized phospholipids ameliorates non-alcoholic steatohepatitis. *Cell Metab* **31**, 189-206 (2020)

Page 5: Authors claim that PCs are most commonly found with one SFA or MUFA while the other is often a PUFA. Again this needs a suitable reference. It could be argued, that in some tissue types also PLs with two PUFA chains are not uncommon. How difficult would the analysis of an oxPL with two PUFAs be? Also, do you have any indication if the sn position of the PUFA has an influence on oxidation pathways?

Answer-2-2

Thank you for your comments. Many previous studies have suggested that SFA and MUFAs are generally linked at the *sn*-1 position and PUFAs tend to be located at the *sn*-2 position of the glycerol backbone.^[R2-11]

^[R2-11] Menzel, D. & Olcott, H. Positional distribution of fatty acids in fish and other animal lecithins. *Biochim Biophys Acta* **84**, 133-139 (1964).

This information, along with the reference, has been added in the following sentences.

[Page-5, line-7]

PCs contain a glycerophosphate backbone linked to two fatty acyl chains, one of which usually belongs to a saturated fatty acid (SFA) or a monounsaturated fatty acid (MUFA) (e.g., palmitic acid (16:0), stearic acid (18:0), and oleic acid (18:1)) and the other belongs to a PUFA (e.g., linoleic acid (18:2), arachidonic acid (20:4), and docosahexaenoic acid (22:6)).¹⁴

Furthermore, as you have mentioned, PCs with two PUFA chains have been found in biological tissues^[R2-11,12]. However, it would be quite difficult to analyze these oxPCs because of their low abundances and complexities of structures. To address this challenge, future studies on improving the sensitivity of analytical methods are required. Also, the positional effects of

PUFAs on lipid peroxidation should be investigated in future studies. Previous research has suggested that the oxidation stabilities of PUFA-containing triacylglycerols depend on the *sn* position of PUFA. [R2-13]

[R2-12] Zacek, P. et al. Quantitation of isobaric phosphatidylcholine species in human plasma using a hybrid quadrupole linear ion-trap mass spectrometer. *J Lipid Res* **57**, 2225-2234 (2016).

[R2-13] Wijesundera, C. et al. Docosahexaenoic acid is more stable to oxidation when located at the sn-2 position of triacylglycerol compared to sn-1(3). *J Am Oil Chem Soc* **85**, 543-548 (2008).

Page 8: In an alternative to MS/MS also ion mobility separation could be used to separate epoxide, hydroxide and hydroperoxide species (<https://doi.org/10.1021/acs.analchem.0c02605>). This could be included either here or maybe as part of a short outlook near the end of the manuscript.

Answer-2-3

Thank you for providing this information. As you have mentioned, the structural determination of oxPC is challenging with oxPC analysis and ion mobility separation can be a powerful method to address this issue. Therefore, we have added the following sentences and a reference in the discussion section.

[Page-18, line-13]

However, further research is needed to clarify the detailed bioactivities of these species towards cell death and inflammation. **In addition, the structural determination of oxPCs *in vivo* is also an important issue, and we should address this challenge with novel technologies, such as ion mobility separation.**⁴¹

Page9: Authors claim that 103 of the 155 identified species are “new”. Please specify what you mean with new. Are these products of new never described oxidation pathways or are these species not found in any library?

Answer-2-4

Thank you for your comment. To date, many researchers have identified a number of oxPCs using MS/MS-based approaches (publications focusing on the structural analysis of oxPCs are listed in **Supplementary Table 2**).

In contrast, in this study, we have determined the formulae of 155 oxPC species derived from three PC(16:0/PUFAs) using non-targeted analysis (**Supplementary Table 1**). Moreover, among them, 103 oxPCs have not been reported previously (**Supplementary Table 2**). Therefore, we defined these 103 oxPCs as “new species,” as highlighted in **Supplementary Table 1**. We have added the following sentences in the revised manuscript.

[Page-8, line-17]

Finally, we identified 155 oxPC species derived from the three PC16:0/PUFAs (Supplementary Table 1**). Surprisingly, among them, 103 oxPCs were novel, which have not been reported by previous studies, as listed in **Supplementary Table 2**.**

Page 10: In order to get a feeling on which oxidation pathways are prevalent in the different in vitro methods an additional S-figure analogous to figure 4a would be very helpful that compares signal intensities for the different oxidation species generated by the different methods.

Answer-2-5

Thank you for your helpful suggestion. As required, we investigated the abundances of non-oxidized and oxidized PC16:0/18:2 generated using several LPO inducers (**Supplementary Figure 5**). The proportions of oxPCs, including carboxylic oxPCs, truncated oxPCs, and full-length oxPCs, in the total PC content differed depending on the inducers. (AAPH, $41.13 \pm 0.48\%$; autoxidation, $16.40 \pm 0.17\%$; AAPH+hemin, $69.11 \pm 1.36\%$; and CuSO_4+AsA , $79.73 \pm 0.76\%$). These results have been specified in the main manuscript as **Supplementary Figure 5** as follows.

Supplementary Figure 5. Semiquantitative analysis of oxidized PC16:0/18:2 generated in several LPO systems. a. Semiquantified concentrations of non-oxidized and oxidized PC16:0/18:2 generated in different LPO systems (black, control; magenta, AAPH; blue, autoxidation; green, AAPH+hemin; and light blue, CuSO_4+AsA). The peak areas from extracted ion chromatogram for each oxPCs were determined by full-scan LC/HRMS in positive ion mode. The semiquantitative values were calculated using the ratio of the MS peak area of individual oxPCs to that of PC15:0/18:1-d₇ (internal standard). Data are presented as the mean + standard deviation of experiments repeated three times. **b.** Proportions of non-oxidized (gray) or oxidized PCs (green, carboxylic oxPCs; orange, truncated oxPCs; red, full-length oxPCs) in the total amount of PCs.

Page 14 and also page 18 (This finding agreed with the fact that laser irradiation during MALDI/MS analysis enhances the formation of reactive oxygen species, such as hydroxyl radical): While the use of 18O is a very clever way of differentiating between oxidation “in vivo” and during the MALDI experiment, it also significantly complicates the experimental setup. Would it also be possible to reduce oxidation during the MALDI process? The cited literature (ref 39) explains the oxidation by reactive hydroxyl groups present in the matrix. Have you tried using MALDI matrices like norharmane or 1,5 diamino-naphthalin (DAN) that do not include hydroxyl groups to reduce the unwanted oxidation side reactions during the MALDI process? Please comment on that either here or in a short outlook.

Answer-2-6

We appreciate the important indication. As suggested, we examined whether the use of MALDI matrices without hydroxyl

groups reduced the unfavorable oxidation reactions during the MALDI process (**Figure R6**). 1,5 diamino-naphthalene (DAN) was used as the MALDI matrix. As in the case of the MALDI matrix DHB, the levels of the m/z 790.5 \rightarrow 772.5 transition, corresponding to PC34:2;O₂, were elevated in both control and AAPH+hemin-treated samples prepared under normal air conditions (**Figure R6a**). Conversely, the m/z 794.5 \rightarrow 774.5 transition, corresponding to PC34:2;¹⁸O₂, was detected only in AAPH+hemin-treated samples prepared under ¹⁸O₂ air conditions (**Figure R6b**). These results suggest that the unfavorable oxidation during the MALDI process occurred regardless of the type of MALDI matrix used.

a

MALDI/MS²: precursor ion; m/z 790.5

b

MALDI/MS²: precursor ion; m/z 794.5

Figure R6. MALDI/MS/MS analysis of PC34:2;O₂ and PC34:2;¹⁸O₂ generated *in vitro* using 1,5 diamino-naphthalene (DAN) as the MALDI matrix. a-b. MALDI/MS/MS spectra showing m/z 790.5 \rightarrow 772.5 (a) and m/z 794.5 \rightarrow 774.5 (b) transitions corresponding to the formation of PC34:2;O₂ and PC34:2;¹⁸O₂, respectively. As the MALDI matrix, we used 1,5 diamino-naphthalene (DAN). To prepare AAPH+hemin-treated samples, PC16:0/18:2 was oxidized using the AAPH+hemin peroxidation system in either normal or ¹⁸O₂ air.

Page 21: electro spray not electron spray

Answer-2-7

Thank you for your comment. We have corrected “electron spray” to “electro spray” as follows.

[Page-22, line-5]

LC/HRMS/MS analysis was performed using the Nexera LC system (Shimadzu Co., Kyoto, Japan) coupled with a high-performance benchtop quadrupole Orbitrap mass spectrometer (Q Exactive, Thermo Fisher Scientific, Waltham, MA) that was equipped with an **electro spray ionization** source.

Figure 5: The diagram in d suggests that the mice were in normal air 105 minutes after APAP treatment and were then transferred to ^{18}O air for 15 mins. Is that correct? If so, please add that information to the text. What color is the staining used in e? Please add the information on pixel size to the figure caption and scale bars to at least one subfigure in d and f.

Answer-2-8

Thank you for your helpful comments. As you have mentioned, the diagram in **Figure 5d** shows that the mice inhaled normal air for 105 min after APAP treatment and then subjected to $^{18}\text{O}_2$ air inhalation for 15 min. We have clarified this information in the **Figure legend** as follows.

[Page-37, line-16]

Figure 5. Visualization of endogenous oxPCs by a combination of MALDI-MS/MS/MSI and $^{18}\text{O}_2$ labeling.

a. Experimental design for the $^{18}\text{O}_2$ labeling of endogenous oxPCs in APAP-treated mice. **b.** Extracted ion chromatograms for ions with m/z 794.5682, 818.5682, and 842.5682 corresponding to PC34:2; $^{18}\text{O}_2$, PC36:4; $^{18}\text{O}_2$, and PC38:6; $^{18}\text{O}_2$, respectively. Black = vehicle-treated group, red = APAP-treated group (120 min in $^{18}\text{O}_2$ air). **c.** HRMS/MS spectra of parent ions with m/z 794.5682, 818.5682, and 842.5682 observed for the APAP-treated group (120 min in $^{18}\text{O}_2$ air). **d.** MALDI-MS/MS/MSI results for PC34:2; $^{18}\text{O}_2$ (794.5 \rightarrow 774.5), PC36:4; $^{18}\text{O}_2$ (818.5 \rightarrow 798.5), and PC38:6; $^{18}\text{O}_2$ (842.5 \rightarrow 822.5) in the mouse liver sections. Left panel = vehicle-treated group, middle panel = APAP-treated group (**mice were incubated in normal air for 105 min after APAP treatment and then incubated in $^{18}\text{O}_2$ air for 15 min**), right panel = APAP-treated group (120 min in $^{18}\text{O}_2$ air). MALDI-MS/MS spectra of ^{18}O -labeled oxPCs extracted from the liver of APAP-treated mice (120 min in $^{18}\text{O}_2$ air). **e.** Distribution of ^{18}O -labeled oxPCs and CYP2E1 expression. Hepatic CYP2E1 expression was analyzed by immunohistochemical staining. Scale bar = 500 μm . **f.** MALDI-MS/MS/MSI results for PC38:6; $^{18}\text{O}_2$ (794.5 \rightarrow 774.5) and GSH (305.3 \rightarrow 206.2).

From Reviewer #3

Overall Comments; In this study, the authors generated a structural library for oxidized phospholipids by chemically oxidizing a group of phospholipids (PCs), then they used this to putatively identify related structures generated in a mouse model of oxidative stress, with the aim of improving structural identification of this class of lipids in model systems. In the first experiment, AAPH +/- hemin was used to chemically oxidize PC species in vitro. Next, an untargeted method was used to annotate the lipids that formed, followed by MS/MS. This is a fairly standard approach nowadays for comparing two sample types and identifying distinguishing features.

Answer-3

Thank you for taking the time to review our manuscript. We appreciate all your comments and suggestions. We have performed additional experiments and revised the manuscript. Please find our itemized responses below.

Following this, different oxidation methods were undertaken and compared. Similar approaches have been taken by several other groups, including Maria Fedorova who using a similar method generated a software tool to manage this, called LPPTiger (<https://home.uni-leipzig.de/fedorova/software/lpptiger/>). The current study cites this paper but has not made use of this in silico tool for their analysis, instead preferring to generate their own library experimentally. It is not clear why they needed to generate a new library given this tool is openly available.

Answer-3-1

Thank you for your comments. As you have mentioned, many innovative approaches, including LPPTiger, have been proposed for the identification of oxidized lipids.^[R3-1,2] However, the mechanism of lipid peroxidation is very complicated, and many of such unknown mechanisms remain to be elucidated. This fact highlights the difficulty in predicting all of the oxidized lipids from previously reported reaction mechanisms.

Indeed, when we compared the libraries of oxidized PC16:0/18:2 constructed using the method developed herein and LPPTiger, there were several oxPCs, such as PC16:0_8:0 and PC16:0_13:3;O2, which were only listed in our library. In addition, the levels of these oxPCs were significantly elevated in the APAP-treated mouse liver (**Figure 3d**). These results suggest that our identification strategy using a non-targeted analysis would enable us to construct an exhaustive library for oxPCs.

^[R3-1] Ni, Z.X., Angelidou, G., Hoffmann, R. & Fedorova, M. LPPTiger software for lipidome-specific prediction and identification of oxidized phospholipids from LC-MS datasets. *Sci. Rep.* **7** (2017).

^[R3-2] Aoyagi, R. et al. Comprehensive analyses of oxidized phospholipids using a measured MS/MS spectra library. *J. Lipid Res.* **58**, 2229-2237 (2017).

Overall, the structures found are to be expected from oxidation (since mechanisms of chemical oxidation of lipids are well known for decades), and so the level of innovation and novelty is rather reduced. Despite this, the authors should be commended for their careful approach to the actual MS analysis itself which is good quality.

Answer-3-2

Thank you for your comments. In this study, we have experimentally determined the formulae of 155 oxPC species derived from three PC16:0/PUFAs using a non-targeted analysis (**Supplementary Table 1**). Moreover, among them, 103 oxPCs have

not been reported in previous studies, as listed in **Supplementary Table 2**. Therefore, we defined these 103 oxPCs as “new species,” as highlighted in **Supplementary Table 1**. We appreciate your recognition of our work pertaining to MS analysis. This information has been added to the manuscript as follows.

[Page-8, line-17]

Finally, we identified 155 oxPC species derived from three PC16:0/PUFAs (**Supplementary Table 1**). Surprisingly, among them, 103 oxPCs were novel, which have not been reported by previous studies, as listed in **Supplementary Table 2**.

The model used in mice involved treatment with acetaminophen, which is well known to be hepatotoxic in overdose through oxidative mechanisms. While it has also been long known how the hepatotoxicity occurs, less is known about the specific oxPL that are generated during this oxidative insult. While this study profiles oxPL generated in this model, the study is rather descriptive in nature and it is unclear how this information would help with design of treatments or other preventative strategies or whether the lipid oxidation is mechanistically important in this form of liver injury. This point has been discussed in a recent review which concluded that lipid peroxidation, although it occurs, is not a relevant injury mechanism in acetaminophen induced liver injury (<https://rosj.org/index.php/ros/article/view/143>, 10.1093/toxsci/kfg220)

Answer-3-3

Thank you for your comments. As you have mentioned, although we have successfully detected 70 endogenous oxPCs generated in APAP-treated mice, the role of these oxPCs in disease development remains to be clarified. The roles of lipid peroxidation in APAP-induced ALF have been controversial. Some researchers have suggested that it is not involved in the pathogenesis of ALF,^[R3-3,4] while lipid peroxidation-dependent cell death, ferroptosis, has been reported to play an important role in APAP-induced ALF.^[R3-5,6]

However, the purpose of this study was the development of a methodology for conducting oxPC analysis. Therefore, in future studies, we will apply our developed method to elucidate the mechanisms of several diseases. For instance, when our method was applied to detect oxPCs generated in carbon tetrachloride (CCl₄)-induced ALF murine model, which progresses via a mechanism different from that of APAP-induced ALF,^[R3-7] the types and amounts of oxPCs generated in CCl₄-induced ALF were different from those in APAP-induced ALF (**Figure 4a** and **Figure R7**). For example, PC34:2;O2 was the most abundant oxPC in the liver of the mice with APAP-induced ALF, while oxPCs derived from PC36:4 and PC38:6 were abundantly generated in that of the mice with CCl₄-induced ALF. These results indicate that the profiles of oxPCs generated *in vivo* differ depending on the mechanisms underlying disease models, and further exhaustive analysis using our proposed method could be a useful strategy for their elucidation.

Figure 4. Epoxide and/or hydroxide motif-containing PC16:0_PUFA;O2 with remarkably increased levels in the early phase of APAP-induced ALF. a. Semi-quantified concentrations of individual oxPCs in vehicle- (black) and 2-h APAP-treated (red) groups. The peak areas from extracted ion chromatogram for each oxPCs were determined by full-scan LC/HRMS in positive ion mode. The semi-quantitative values were calculated using the ratio of the MS peak area of individual oxPCs to that of PC15:0/18:1-d₇ (internal standard). Data are presented as the mean + standard deviation of experiments repeated six times.

Figure R7. Semi-quantitative analysis of oxidized PCs generated in carbon tetrachloride (CCl₄)-induced ALF murine model. Semi-quantified concentrations of individual oxPCs in vehicle- (black) and 4-h CCl₄-treated (blue) groups. A single dose of CCl₄ at 500 μ L/kg body weight was intraperitoneally administered into six-week-old ICR male mice, and the animals were anesthetized 4 h after CCl₄ administration. Mice liver samples were collected and immediately frozen in liquid nitrogen. Hepatic lipids were extracted from the liver samples according to the modified Bligh and Dyer method. The peak areas from extracted ion chromatogram for each oxPCs were determined by full-scan LC/HRMS in positive ion mode. The semi-quantitative values were calculated using the ratio of the MS peak area of individual oxPCs to that of PC15:0/18:1-d₇ (internal standard). Data are presented as the mean + standard deviation of experiments repeated six times.

[R3-3] Knight, TR. et al. Role of lipid peroxidation as a mechanism of liver injury after acetaminophen overdose in mice. *Toxicol. Sci.* **76**, 229-236 (2003).

[R3-4] Ramachandran, A. & Jaeschke H. A mitochondrial journey through acetaminophen hepatotoxicity. *Food Chem. Toxicol.* **140**, 111282(2020).

[R3-5] Yamada, N. et al. Ferroptosis driven by radical oxidation of n-6 polyunsaturated fatty acids mediates acetaminophen-induced acute liver failure. *Cell Death Dis.* **11**, 144 (2020).

[R3-6] Wang, M. et al. (+)-Clausenamide protects against drug-induced liver injury by inhibiting hepatocyte ferroptosis. *Cell Death Dis.* **11**, 781 (2020).

[R3-7] Recknagel, RO. Carbon tetrachloride hepatotoxicity, *Pharmacol. Rev.* **19**, 145-208(1967).

The acetaminophen mouse experiment lipidome analysis includes putative structural identification of epoxy-hydroxy-oxPL, however this is very speculative and without any standards it is very difficult to be convinced the annotation is really proven in terms of position of oxidation and precise functional group assignment.

Some ions that are labelled in the fragmentation spectra are extremely minor, and while this is to be expected in this type of experiment (a complex mix of many lipids fragmenting together in the MS will generate complex data), great care is needed with assigning any more than simply carbon chain length, rings/double bonds and number of oxygenations when there are no

standards that can be compared with the lipids being analysed.

The inclusion of an 18-O method is a nice innovation to improve exclusion of artefacts.

Answer-3-4

Thank you very much for your comments and valuable suggestions. In order to examine the validity of the estimated structure of PC16:0_PUFA;O₂, as shown in **Figure 4c-e**, the detail structure of PC16:0_PUFA;¹⁸O₂ generated in 2-h APAP-treated mice inhaling ¹⁸O₂ air was analyzed by LC/HRMS/MS in negative ion mode. If the estimated structures are appropriate, the mass of product ion increases according to the number of ¹⁸O-modified sites. **Supplementary Figure 13** shows the HRMS/MS spectrum of *m/z* 838.5582 corresponding to PC16:0_18:2;¹⁸O₂. We could observe the increase in the masses of each product ion as expected, and these results indicate the validity of the estimated structures of PC16:0_18:2;O₂ (**Figure 4c**).

Supplementary Figure 13. LC/HRMS/MS identification and plausible structures of PC16:0_18:2;¹⁸O₂ generated in mice treated for 2 h with APAP and inhaling ¹⁸O₂ air. HRMS/MS spectra of the product ion at *m/z* 838.5582 corresponding to PC16:0_18:2;¹⁸O₂.

Moreover, as the reviewer has highlighted, the product ions with extremely low peak intensities were removed from **Figure 4d-e**. These data have been specified in the manuscript as follows.

[Page-17, line-14]

The detected species contained epoxide and hydroxide moieties possibly formed via metal-ion induced peroxidation (**Figures 2d and 4c-e**). In addition, our detailed structural analysis of PC16:0_18:2;¹⁸O₂ generated in mice that had inhaled ¹⁸O₂, and were treated for 2 h with APAP validated the estimated structures (**Supplementary Figure 13**).

They mention that LIPID MAPS doesn't include many oxPL, however on checking this, we find that LMSD includes 259, while the in silico database includes over 44K structures.

Answer-3-5

Thank you for your comments. As you have mentioned, LMSD contains information on 256 oxidized lipids in total, and furthermore, less than 60 oxGPLs are listed under each GPL class (e.g., the number of oxPCs listed in LMSD is 53). We apologize for the confusion, and the sentences in the introduction section have been revised as follows.

[Page-4, line-1]

Indeed, the current lipid database, viz. LIPIDMAPS (www.lipidmaps.org/), contains **only 53 oxPCs**, which again, is ascribed

to the lack of structural information.

In addition, the database constructed herein was based on experimental data. Therefore, although the total number of listed oxPLs is smaller than that in the previously reported *in-silico* database, many novel oxPCs have been identified owing to the unique identification strategy using non-targeted analysis (Please see *Answer-3-1*, 3-2).

LIPID MAPS recently updated the shorthand nomenclature, including for oxPL, and adapting to this naming convention would be recommended (<https://www.jlr.org/content/61/12/1539.long>).

Answer-3-6

Thank you for the valuable suggestion. We have revised the nomenclature used in the manuscript text accordingly.

Supplementary Figure 6 shows an impact of oxPC on gene expression in RAW cells using PCR. This data isn't mentioned in the results section and only in the discussion and its relevance to the study is unclear. Also, it's not novel since there are many studies showing oxPL can regulate TLR signaling in macrophages, both activation and inhibition depending on experimental design. They state that this may mean that oxPL could be considered DAMPs. However, there is an extensive literature for many years already existing that refers to oxPL as DAMPs, and this is widely acknowledged and accepted.

Answer-3-7

Thank you for your comments. Based on the reviewer's comments, we further investigated the proinflammatory effects of the hepatic lipid extracts from normal or 2-h APAP-treated mice on RAW 264.7 cells (**Supplementary Figure 14d**). The mRNA expression levels of IL-1 β increased upon the addition of lipid extracts from APAP-treated mice, but not normal mice. Although further research is warranted, these results suggest that oxPLs generated by APAP treatment might induce inflammation in APAP-induced ALF. These data have been specified in the discussion section as follows.

Supplementary Figure 14. Proinflammatory effects of 16:0-20:4-PC on RAW 264.7 cells. d. RAW 264.7 cells were treated with lipid extracts from the livers of either normal (white) or 2-h APAP-treated (gray) mice. To obtain the extraction solutions, 200 mg mouse liver tissues were collected and hepatic lipids were extracted using the modified Bligh and Dyer method. The mRNA expression levels of Il-1b were analyzed at 1, 2, and 4 h post-treatment. Data are presented as the mean + standard deviation of experiments repeated three times. Significance was determined using two-way ANOVA, followed by Sidak's comparison test. * $P < 0.05$, compared with the normal group.

When we investigated the proinflammatory effects of oxPCs on RAW 264.7 cells, the mRNA expression of proinflammatory cytokines increased upon the addition of oxPCs obtained by metal ion-induced (AAPH + hemin-induced) LPO (**Supplementary Figure 14a-c**). Furthermore, the lipid extracts from the livers of APAP-treated mice, but not normal mice, showed an increase in the expression of Il-1b mRNA (**Supplementary Figure 14d**).

REVIEWER COMMENTS

Reviewer #1 (Remarks to the Author):

The authors describe the development of a mass spec-based analytical strategy to detect and visualize oxidized phosphatidylcholines (oxPCs) from biological samples. Their developed method combines experimental and in silico data to construct a database of possible oxPCs. The database was validated using an acute liver failure model. The use of mass spec imaging provided the opportunity to localize oxPCs in liver tissue exposed to toxic levels of acetaminophen. This manuscript provides a starting point for identifying novel oxidized lipids that can give insight into disease progression. The concept and general approach is of high interest and suitable for Nature Communication.

I had a number of questions and comments after the first review. The authors systematically and comprehensively addressed my questions and comments. The authors are commended for providing added explanations and data to adequately assess their manuscript. I do not have any further questions and recommend acceptance of the manuscript in its present form.

Reviewer #2 (Remarks to the Author):

The manuscript was carefully revised and new experimental data was included as requested. All of my concerns/questions were appropriately met and answered.

Reviewer #3 (Remarks to the Author):

The authors have revised in line with my comments. I have one main comment remaining based on new information provided in the revision.

One new addition is information about the method they consider as "semiquantitation" and to which they give ng values. I have concerns about this approach which uses a single internal standard

for all lipids being "quantified" since this is using reverse phase LC where lipids are eluting over a very wide time window, with a significant variation in mobile phase composition during this time. This means ionisation efficiency will vary widely with early eluting compounds likely to be detected far more sensitively than later eluting oxPC (an issue that is well known for native PLs). Obviously there are no primary standards for most of the lipids so this is not easy to solve (quantitatively) but if we cannot have some level of confidence in the values then it would be far better to simply call this A/IS and talk about relative fold change when the tissue is oxidized versus the control.

I have noticed in the Excel sheets that some lipids have extremely wide elution windows, e.g. 4-9 or 7-12 min in some cases. Presumably this means there are many peaks coming off (many different isomers of these m/z values), and it is ever more incorrect to use a single standard to generate ng values in a case like this, and call it one lipid. It maybe fine if all lipids were coming off together, or very close but this is not the case here. It would be important to see the chromatograms (examples) for the lipids being reported on in experiments.

The method used need also to have more detail, presumably this was using the headgroup ion (in pos mode) but that wasn't stated. There is no information on peak quality (S/N threshold for LOD/LOQ, points across a peak) and no example chromatograms are provided. Can we have one for each lipid so we can see the data quality, included as supplementary data. The quality of the chromatograms in Fig S10 needs improving, e.g. the lines are not clear for the data in some panels, and just look a bit like random dots that are not a clear line (middle panel, right column).

Responses to the Reviewers' comments

From Reviewer #1

The authors describe the development of a mass spec-based analytical strategy to detect and visualize oxidized phosphatidylcholines (oxPCs) from biological samples. Their developed method combines experimental and in silico data to construct a database of possible oxPCs. The database was validated using an acute liver failure model. The use of mass spec imaging provided the opportunity to localize oxPCs in liver tissue exposed to toxic levels of acetaminophen. This manuscript provides a starting point for identifying novel oxidized lipids that can give insight into disease progression. The concept and general approach is of high interest and suitable for Nature Communication.

I had a number of questions and comments after the first review. The authors systematically and comprehensively addressed my questions and comments. The authors are commended for providing added explanations and data to adequately assess their manuscript. I do not have any further questions and recommend acceptance of the manuscript in its present form.

Answer-1

We greatly appreciate the reviewer's recognition of our work.

From Reviewer #2

The manuscript was carefully revised and new experimental data was included as requested. All of my concerns/questions were appropriately met and answered.

Answer-2

Thank you very much for the recognition of our work.

From Reviewer #3

One new addition is information about the method they consider as "semiquantitation" and to which they give ng values. I have concerns about this approach which uses a single internal standard for all lipids being "quantified" since this is using reverse phase LC where lipids are eluting over a very wide time window, with a significant variation in mobile phase composition during this time. This means ionisation efficiency will vary widely with early eluting compounds likely to be detected far more sensitively than later eluting oxPC (an issue that is well known for native PLs). Obviously there are no primary standards for most of the lipids so this is not easy to solve (quantitatively) but if we cannot have some level of confidence in the values then it would be far better to simply call this A/IS and talk about relative fold change when the tissue is oxidized versus the control.

I have noticed in the Excel sheets that some lipids have extremely wide elution windows, e.g. 4-9 or 7-12 min in some cases. Presumably this means there are many peaks coming off (many different isomers of these m/z values), and it is ever more incorrect to use a single standard to generate ng values in a case like this, and call it one lipid. It maybe fine if all lipids were coming off together, or very close but this is not the case here. It would be important to see the chromatograms (examples) for the lipids being reported on in experiments.

Answer-3-1

Thank you very much for your helpful comments. In **Figure 4a**, and **Supplementary Figures 6** and **8**, based on the comments from reviewer-1 and reviewer-2, we have presented the results obtained from the semiquantification of (ox)PCs to determine the abundance of oxPCs compared to that of non-oxidized PCs (Please see Answers-1-9, 14, 21, 24 and Answers-2-5 in the 1st revision). As the reviewers mentioned, the abundance of oxPCs compared to that of non-oxidized PCs can be vital in understanding the basal levels of oxPCs in pathological states and for evaluating the applicability of our developed method. However, as reviewer-3 indicated, the quantitative potential of our developed methods is still a matter of concern, including the effects of mobile phase composition and variation of fatty acyl (FA) structures on the ionization efficiency. To examine these effects, we investigated the ionization efficiencies of seven standard (ox)PCs with different chemical structures and retention times (**Figure S5a, b**) by using LC/HRMS in the positive ion mode. The ionization efficiencies for the individual PCs were not completely consistent but were relatively close (**Figure S5c, d**). Although the influences of biological matrix and other factors on the quantitative accuracy of our method for oxPCs have not been fully evaluated, the effects of mobile phase composition and FA structures on the ionization efficiency were considered to be small under the conditions used in this study.

Modern LC/MS instruments are equipped with a heated electrospray ionization (H-ESI) source (heater temperature = 425 °C in this study); therefore, in this study, the solvent removal efficiency may have improved, and the effect on the difference in mobile phase composition ionization efficiency may have reduced. In addition, previous studies have suggested that the ionization efficiencies of lipid molecular species, including PC, are mainly dependent on their polar head groups and slightly dependent on their FA chains. ^[R1-3]

From these results, we considered that LC/HRMS in the positive ion mode can be used for the semiquantification of (ox)PCs. However, as indicated, to improve the quantitative potential of our developed methods, further studies, including the preparation of standard oxPCs and the lipid class separation of oxPCs using hydrophilic interaction liquid chromatography (HILIC) or supercritical fluid chromatography (SFC), are warranted.

[R1] Han, X. & Gross, RW. Electrospray ionization mass spectroscopic analysis of human erythrocyte plasma membrane phospholipids. *Proc Natl Acad Sci USA* **91**, 10635-10639 (1994)

[R2] Han, X. & Gross, RW. Shotgun lipidomics: electrospray ionization mass spectrometric analysis and quantitation of cellular lipidomes directly from crude extracts of biological samples. *Mass Spectrom Rev* **24**, 367-412 (2005)

[R3] Jung, HR. et al. High throughput quantitative molecular lipidomics. *Biochim Biophys Acta* **1811**, 925-934 (2011)

a

Supplementary Figure 5. LC/HRMS measurements of seven standard (ox)PCs. a. Chemical structures of the PCs employed in this experiment. **b.** HPLC gradient program and extracted ion chromatograms (EICs) of 10 pmol standard (ox)PCs measured under the positive ion mode. **c.** LC/HRMS peak areas of PC (10 pmol) measured under either negative (black) or positive (red) ion mode. LC/HRMS peak areas of PCs measured by the positive ion mode were higher than those measured by the negative ion mode. **d.** Calibration curves for PC, pmol vs. LC/HRMS peak area measured under the positive ion mode. Data are presented as the mean \pm standard deviation of experiments repeated three times.

Based on the comments from reviewer-3, we have added the following information in the results section.

[Page-9, line-15]

To examine the quantitative potential of LC/HRMS in the positive ion mode, we analyzed the ionization responses of seven commercially available (ox)PCs with different chemical structures and retention times (**Supplementary Figure 5a, b**). LC/HRMS peak areas of (ox)PCs observed in the positive ion mode were higher than those observed in the negative ion mode. Furthermore, the ionization efficiencies of each PCs were not completely consistent, but were relatively close (**Supplementary Figure 5c, d**). The LODs of oxPCs using the optimized LC/HRMS method in the positive ion mode were approximately 30 fmol (**Supplementary Table 4**), and the HRMS(/MS) operating parameters were set to achieve more than 15 data points across each (ox)PC peak. From these results, we concluded that LC/HRMS in the positive ion mode can be

used for the semiquantification of (ox)PCs. Furthermore, for the semiquantification of oxPCs in the positive ion mode, we used PC15:0/18:1-d₇ as an internal standard.

The method used need also to have more detail, presumably this was using the headgroup ion (in pos mode) but that wasn't stated. There is no information on peak quality (S/N threshold for LOD/LOQ, points across a peak) and no example chromatograms are provided. Can we have one for each lipid so we can see the data quality, included as supplementary data. The quality of the chromatograms in Fig S10 needs improving, e.g. the lines are not clear for the data in some panels, and just look a bit like random dots that are not a clear line (middle panel, right column).

Answer-3-2

We appreciate your helpful suggestions. For a detailed description of the methods employed for the measurement of oxPCs in each experiment, as the same point was raised by reviewer-1 (Please see Answers-1-8 in the 1st revision), the following information was added to the methods section.

[Page-22, line-14]

HRMS(MS) in negative ion mode is a reliable diagnostic method for elucidating PC structures.¹⁷ This technique has also been applied to monitor the levels of individual oxPCs using the product ions specific to each oxPC.³³ In contrast, since the ionization efficiency of lipid molecular species including phosphatidylcholine was mainly dependent on their polar head groups and slightly dependent on their FA side chains,⁴⁴ this method cannot provide detailed information on oxPC structure, including the chemical formulae of the two fatty acyl moieties. Therefore, we used the negative ion mode in the structural analysis (**Figure 1**) and monitored the individual changes in oxPCs (**Figure 2, 3d-g**), whereas the positive ion mode was employed for comparative analysis of the abundances of different oxPCs (**Figure 4a** and **Supplementary Figure 6, 8**) (**Supplementary Figure 17**).

Supplementary Figure 17. Summary of the detection methods employed in this study. DDA; data dependent acquisition, PRM; parallel reaction monitoring.

[Page-23, line-9]

Individual oxPCs were identified by full-scan MS/data-dependent MS/MS or parallel reaction monitoring (PRM). The

ionization conditions were as follows: ionization mode = negative or positive, sheath gas flow rate = 40 arbitrary units, auxiliary gas flow rate = 10 arbitrary units, spray voltage = 2000 V, capillary temperature = 265 °C, S-lens level = 50, and heater temperature = 425 °C. The experimental conditions for full-scan MS were as follows: resolving power = 70,000, automatic gain control target = 1×10^6 , trap fill time = 100 ms, scan range = m/z 120–1500. The experimental conditions for MS/MS and PRM were as follows: resolving power = 17,500, automatic gain control target = 1×10^6 , trap fill time = 80 ms, isolation width = ± 0.6 Da, fixed first mass = m/z 80, normalized collision energy = 20 or 35 eV, intensity threshold of precursor ions for MS/MS analysis = 3100, apex trigger = 2–4 s, and dynamic exclusion = 2 s. The intensity threshold of precursor ions for MS/MS analysis and the dynamic exclusion were set to 1×10^4 and 1 s, respectively.

In addition, the information related to LOD/LOQ and points across a peak has been added. In addition, as examples, LOD/LOQ of two standard oxPCs (PC16:0/9:0;9oxo and PC16:0/5:0;5oxo) were estimated and added to the results section. The revised text is as follows:

[Page-10, line-2]

The LODs of oxPCs using the optimized LC/HRMS in the positive ion mode were approximately 30 fmol (**Supplementary Table 4**), and the HRMS(/MS) operating parameters were set to achieve more than 15 data points across each (ox)PC peak.

[Page-9, line-11]

In addition, we investigated the abundances of non-oxidized and oxidized PC16:0/18:2 through LC/HRMS analysis in the positive ion mode, which is a useful method for highly sensitive semiquantification of (ox)PCs (for e.g., limits of detection (LOD) and limits of quantitation (LOQ) of standard oxPCs measured by LC/HRMS in the positive ion mode are listed in **Supplementary Table 4**).

Table S4 | Limits of detection (LOD) and limits of quantitation (LOQ) of two standard oxPCs measured by LC/HRMS in the positive ion mode.

Compound	LOD ^a (nmol L ⁻¹)	LOQ ^b (nmol L ⁻¹)
PC16:0/9:0;9oxo	2.9	9.6
PC16:0/5:0;5oxo	3.2	11

^a LOD was estimated based on S/N=3.

^b LOQ was determined by S/N=10.

Moreover, we have added a list of EICs of individual oxPCs identified in this study as **Figure S18**. For the LC/MS chromatograms in **Figure S11** (**Figure S10** in old version), the resolution of the data image was not sufficient, so it has been replaced with new data. We apologize for any confusion that might have been caused due to low resolution of our image. In addition, as the reviewer mentioned, the peak shapes of individual chromatograms should be improved in future studies, and it is necessary to develop novel separation systems specifically for the analysis of oxidized lipids.

[Page-24, line-3]

LC/HRMS/MS analysis was controlled using Xcalibur 4.2.47 software (Thermo Fisher Scientific). EICs of individual oxPCs identified in this study are shown in **Supplementary Figure 18**.

Figure S18

Continued Figure S18

a

Continued Figure S18

b

Continued Figure S18

b

Continued Figure S18

b

Continued Figure S18

Continued Figure S18

C

PC16:0_7:1;COOH

PC16:0_7:2;O;COOH

PC16:0_11:4;O

PC16:0_7:2;O2

PC16:0_7:1;O;COOH

PC16:0_10:4;O2

PC16:0_6:1;O3

PC16:0_7:2;O3

PC16:0_10:2;COOH

PC16:0_9:3;O

PC16:0_7:1;O3

PC16:0_10:3;O2

PC16:0_8:2;COOH

PC16:0_10:3;O

PC16:0_9:3;O3

PC16:0_8:3;O2

PC16:0_9:3;O2

PC16:0_12:4;O

Continued Figure S18

C

Continued Figure S18

C

PC16:0_13:3;O2;COOH

PC16:0_16:4;O;COOH

PC16:0_19:5;O2

PC16:0_13:4;O4

PC16:0_19:6;O

PC16:0_19:6;O3

PC16:0_17:6;O3

PC16:0_18:6;O2

PC16:0_22:6;oxo

PC16:0_16:5;O2

PC16:0_17:6;O3

PC16:0_22:6;O

PC16:0_15:5;O3

PC16:0_19:6;O2

PC16:0_22:7;O;oxo

PC16:0_18:6;O

PC16:0_18:6;O3

PC16:0_22:6;O;oxo

Continued Figure S18

C

Supplementary Figure 18. EICs of individual oxPCs derived from PC16:0/18:2 (a), PC16:0/20:4 (b) and PC16:0/22:6 (c) measured by both negative and positive ion modes.

Supplementary Figure 11. ^{18}O conversion of PC34:2;O2 generated under AAPH + hemin peroxidation system in $^{18}\text{O}_2$ air. Extracted ion chromatograms (EICs) for ions with m/z 790.5598 (black), 792.5640 (blue), and 794.5682 (red) corresponding to PC34:2; $^{16}\text{O}_2$, PC34:2; $^{16}\text{O}^{18}\text{O}$, and PC34:2; $^{18}\text{O}_2$, respectively. Left, right: PC34:2;O2 generated under AAPH + hemin peroxidation system in $^{18}\text{O}_2$ and normal air, respectively.

REVIEWERS' COMMENTS

Reviewer #3 (Remarks to the Author):

Thank you for addressing all my comments and including further information about how the lipids were analysed and semi-quantified, and the steps taken to be as rigorous as possible, recognising that this is a challenging area. The additional information is informative and should help others in the field to look in detail at this dataset as part of taking the work further. OxPL are challenging to analyse, and the authors have generally taken a pragmatic approach. The inclusion of chromatograms along with LOD and LOQ criteria is very helpful.

Responses to the Reviewers' comments

From Reviewer #3

Thank you for addressing all my comments and including further information about how the lipids were analysed and semi-quantified, and the steps taken to be as rigorous as possible, recognising that this is a challenging area. The additional information is informative and should help others in the field to look in detail at this dataset as part of taking the work further. OxPL are challenging to analyse, and the authors have generally taken a pragmatic approach. The inclusion of chromatograms along with LOD and LOQ criteria is very helpful.

Answer-1

We greatly appreciate the reviewer's recognition of our work.